# NOT HOW YOU THINK, IT'S WHAT YOU SEE: DECOUPLING PERCEPTION FROM REASONING

## ABSTRACT

The ability of Vision-Language Models (VLMs) to reason depends on a complex interplay between visual perception and abstract cognition. While it is widely recognized that perception is a significant bottleneck, systematically diagnosing how it fails and developing methods to unlock latent reasoning capabilities remains a key challenge. To address this, we introduce a cognitively-inspired framework that decomposes VLM behavior through four distinct paradigms: 1) Direct Visual Rule Learning (holistic processing), 2) Deductive Rule Learning (explicit rule extraction), 3) Componential Analysis (CA), which decouples perception by reasoning over task-agnostic textual descriptions, and 4) Interactive Componential Analysis (ICA), which introduces a feedback loop for targeted visual probing. Our framework's emphasis on task-agnostic decomposition and cognitive parallels provides a unique lens for analysis compared to prior decoupling efforts. Applying this framework across an expanded suite of benchmarks, we conduct a comprehensive evaluation on both proprietary and open-source multi-image VLMs. Our results confirm that perception is a primary bottleneck and show that our CA and ICA paradigms yield substantial performance gains, unlocking the latent reasoning abilities of powerful LLMs. Crucially, ICA demonstrates that an interactive loop can resolve fine-grained visual ambiguities that static descriptions cannot, outperforming the non-interactive CA approach. Our work provides a robust diagnostic toolkit for the community and offers concrete architectural insights, demonstrating that interactive, decoupled systems are a promising path toward more general and capable visual intelligence.

## 1 INTRODUCTION

Human cognition adeptly integrates visual perception with abstract reasoning to navigate the world (Kunda, 2020; Lake et al., 2017). While Vision-Language Models (VLMs) have made remarkable progress (Zhao et al., 2023; Radford et al., 2021), their ability to perform complex visual reasoning remains brittle. It is widely recognized that a primary failure point is the model's visual perception, but we lack systematic tools to diagnose how these perceptual systems fail and frameworks to mitigate these failures to unlock latent reasoning.

To investigate these questions, we test models on two challenging task families that probe the limits of perception and reasoning. First, Bongard Problems (BPs) (Bongard, 1968), a classic test requiring few-shot discovery of an abstract visual rule. We use natural image variants, Bongard-OW (Wu et al., 2024) and Bongard-HOI (Jiang et al., 2022). Second, Winoground (Thrush et al., 2022), which tests visio-linguistic compositional reasoning through minimally contrastive image-caption pairs.

This paper introduces an evaluation framework designed to dissect the cognitive processes of VLMs on these tasks. Our core contribution is a framework grounded in cognitive science that, unlike prior work that inventories static capabilities, uses dynamic paradigms to model problem-solving strategies:

1. **Direct Visual Rule Learning (DVRL):** Simulates holistic processing (Biederman, 1987), where the model analyzes all images simultaneously.

2. **Deductive Rule Learning (DRL):** Mimics explicit, rule-based deduction (Rips, 1994), separating rule extraction from application.

3. **Componential Analysis (CA):** Parallels analytical decomposition (Gluck et al., 2008), reasoning over structured, task-agnostic textual descriptions.

4. **Interactive Componential Analysis (ICA):** Extends CA with a feedback loop, allowing the reasoning module to actively probe the perception module for targeted details.

This framework allows for systematic analysis of VLM behavior, identifying specific processing bottlenecks. By generating comprehensive, *task-agnostic* image descriptions, our componential paradigms allow us to *disentangle perception from reasoning*. ICA enhances this by enabling a dynamic perceptual process guided by reasoning needs. This approach facilitates multi-image reasoning on single-image architectures and even allows us to evaluate text-only LLMs by providing high-quality descriptions (Section 7.2).

Applying this framework across a broadened set of benchmarks and models, we find that our CA and ICA paradigms yield substantial performance gains. These methods achieve highly competitive results on Bongard-OW, Bongard-HOI, and Winoground, primarily by pairing high-fidelity descriptions with powerful reasoning models. This success across diverse tasks highlights the robustness of decoupling perception from reasoning. Concurrently, our analysis confirms a significant *perception bottleneck*, as models' performance drastically improves when their perceptual front-end is bypassed or guided.

Our contributions are:

1. A novel, cognitively-inspired framework with four distinct paradigms (including interactive reasoning) for the diagnostic evaluation of VLMs.

2. A componential method (CA and ICA) to disentangle perception from reasoning, enabling multi-image task evaluation for diverse architectures and unlocking latent reasoning in LLMs.

3. Comprehensive empirical results on multiple benchmarks and models, confirming the perception bottleneck and demonstrating that our interactive, decoupled methods can significantly mitigate it.

4. A demonstration of the effectiveness of this approach, achieving strong, competitive performance on challenging visual reasoning tasks.

## 2 RELATED WORK

**VLM Benchmarks.** The evaluation of VLMs has rapidly evolved from foundational tasks like VQA (Antol et al., 2015) to benchmarks testing more complex reasoning. Recent efforts focus on multi-image understanding through interleaved corpora (Laurençon et al., 2024) and dedicated benchmarks such as MuirBench (Wang et al., 2024) or low-level perception tests like BLINK (Fu et al., 2024). Our work complements these by focusing on benchmarks specifically designed to probe core cognitive abilities that resist simple linguistic mediation: abstract few-shot rule discovery using natural image Bongard Problems (BPs) (Wu et al., 2024; Jiang et al., 2022) and fine-grained compositional reasoning via Winoground (Thrush et al., 2022).

**Cognitive Science Grounding.** Our framework is grounded in cognitive science perspectives on human problem-solving (Newell et al., 1972). Our paradigms model distinct cognitive strategies: *Direct Visual Rule Learning (DVRL)* mirrors rapid, holistic processing (Biederman, 1987); *Deductive Rule Learning (DRL)* reflects explicit, rule-based deduction (Rips, 1994); and our *Componential Analysis (CA)* paradigms parallel analytical decomposition, where problems are broken into constituent parts for systematic reasoning (Gluck et al., 2008).

**Decoupling Frameworks and Chain-of-Thought.** Our approach is related to a growing body of work on decoupling perception and reasoning in VLMs. This includes Multimodal Chain-of-Thought (CoT) prompting (Zhang et al., 2023; Zheng et al., 2023) and dedicated evaluation frameworks. For instance, Prism (Qiao et al., 2024) provides a valuable framework for assessing a static inventory of fine-grained VLM skills like object recognition and counting.

Our contribution is distinct in three key ways. First, our paradigms are grounded in cognitive processes (e.g., holistic vs. deductive), analyzing how a model solves a problem, not just what skills it possesses. Second, our CA paradigm deliberately uses task-agnostic descriptions, creating a

clean separation between raw perceptual capability and downstream reasoning, unlike many CoT methods that generate task-conditioned descriptions. Finally, our new Interactive CA (ICA) paradigm introduces a novel dynamic feedback loop, where the reasoning module actively probes the perception module. This moves beyond the static, one-pass evaluations common in prior work to model a more realistic, iterative reasoning process.

## 3 MODELS

We evaluated a diverse suite of VLMs, distinguishing between models based on their multi-image context capacity. State-of-the-art proprietary systems, including **GPT-4o** (OpenAI, 2024), GPT 5.1 and **Gemini 2.0** (Google, 2024), can natively process the large number of images (13) required for our DVRL and DRL paradigms on Bongard Problems. In contrast, while some contemporary open-source models like **Pixtral-12B** (Agrawal et al., 2024), **Llama-Vision-3.2** (Meta, 2024), and **LLaVA** variants (Liu et al., 2023; XTuner, 2025) accept multiple images, none currently support the large context required for a direct evaluation in these paradigms. Additionally, VLMs like Gemma 3 (Team et al., 2025), Qwen 2.5-vl (Bai et al., 2025) were also used. This technical constraint underscores the necessity of our Componential Analysis (CA) paradigm as the primary method for assessing complex, multi-image reasoning on these powerful open-source architectures. For ablation studies, we also used text-only LLMs (**Llama3** (Grattafiori et al., 2024), **Phi-4** (Abdin et al., 2024), etc.). All evaluations used few-shot prompting at zero temperature. Further details are in Appendix A.5.

## 4 DATASET AND TASK

We test our framework on a diverse suite of benchmarks chosen to probe distinct cognitive abilities. Our primary testbed is Bongard-OW (Wu et al., 2024), a 500-case subset testing few-shot abstract rule discovery on natural images (see Figure 1 for an example). Category distribution closely matches the full dataset (Appendix A.6). To assess generalization, we use Bongard-HOI (Jiang et al., 2022) (400 samples) to evaluate reasoning about human-object interactions, and Winoground (Thrush et al., 2022) (400 samples) for fine-grained compositional grounding. Together, these tasks provide a robust testbed for analyzing high-level visual reasoning, from abstraction to compositionality, across our different cognitive paradigms.

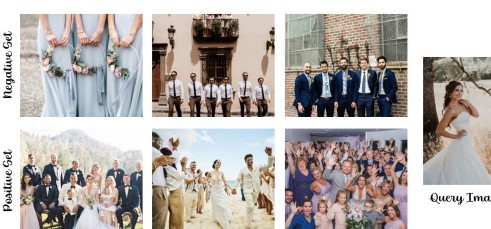

Figure 1: Example Bongard-OW task. *Bottom*: Positive examples. *Top*: Negative examples. *Right*: Query. Rule: *A group photo at a wedding reception*. Query is negative. (3 of 6 examples shown per set).

## 5 COGNITIVELY-INSPIRED EVALUATION PARADIGMS

We evaluate VLMs using four paradigms designed to probe different facets of visual reasoning and assess performance under systematically varied cognitive demands, inspired by human cognitive strategies. All paradigms require the model to output a structured response including analysis, the derived rule, query description, and classification (positive/negative). Figure 2 provides a schematic overview. Specific prompts are detailed in Appendix A.6.

### 5.1 DIRECT VISUAL RULE LEARNING (DVRL)

This paradigm assesses holistic reasoning by presenting all 13 images (6 positive, 6 negative, 1 query) simultaneously to the VLM. It demands the model integrate information across the entire set to identify the rule and classify the query in one step. This mirrors the human ability to quickly grasp the 'gist' of a visual scene or problem. Due to requiring simultaneous multi-image input, only models like Gemini 2.0 and GPT-4o were tested under this paradigm.

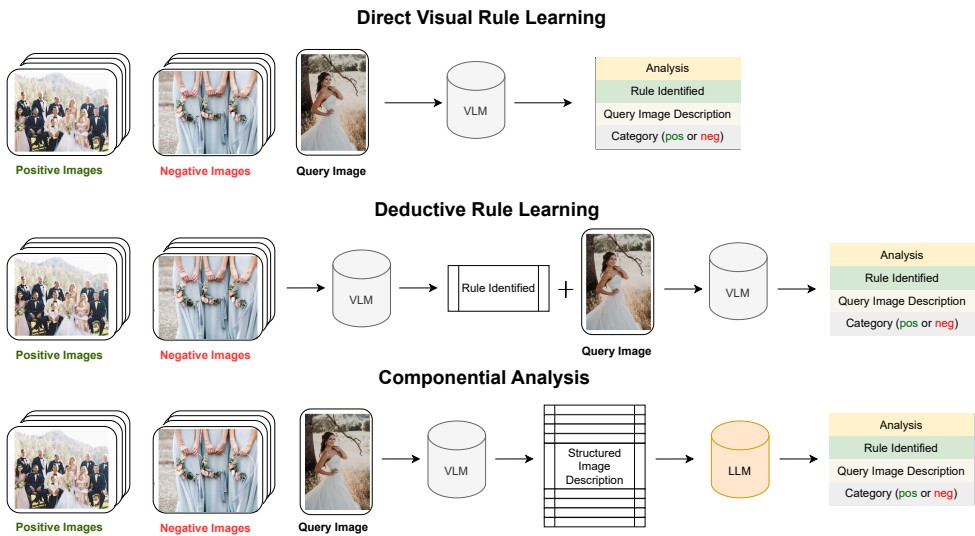

Figure 2: **Cognitively-Inspired Evaluation Paradigms. DVRL** (Direct Visual Rule Learning): Concurrent processing of all images, mimicking holistic perception. Requires multi-image input capability. **DRL** (Deductive Rule Learning): Two-stage process separating rule extraction from application, mimicking explicit deduction. **CA** (Componential Analysis): Multi-stage process involving individual image description followed by reasoning over text, mimicking analytical decomposition and enabling perception-reasoning separation.

## 5.2 DEDUCTIVE RULE LEARNING (DRL)

Mimicking deliberative, rule-based deduction, DRL involves two stages:

1. **Rule Extraction:** The VLM analyzes the 12 context images (positive/negative sets) to identify and concisely summarize (max 20 words) the distinguishing rule.
2. **Rule Application:** The VLM receives the previously generated rule summary and the query image, classifying the query based solely on the provided rule.

This separation allows examining the fidelity of both rule formation and rule application processes.

## 5.3 COMPONENTIAL ANALYSIS (CA)

Reflecting analytical problem decomposition, CA proceeds in stages based on textual representations:

1. **Image Description:** The VLM generates a detailed, structured, and ideally *task-agnostic* JSON description for each of the 13 images *individually*.
2. **Text-Based Reasoning:** A powerful LLM receives the collection of 13 JSON descriptions (labeled positive/negative/query) and performs rule extraction and query classification based *only* on this textual input.

This paradigm is crucial as it (a) allows evaluating models lacking direct multi-image input, (b) enables assessing reasoning largely independent of perceptual errors, and (c) facilitates evaluation of text-only LLMs on visual reasoning tasks.

## 5.4 INTERACTIVE COMPONENTIAL ANALYSIS (ICA)

To mitigate the limitations of static perception revealed by CA, this paradigm extends it with a dynamic, multi-step feedback loop that emulates a "look again" strategy.

1. **Initial Description:** Same as in CA, the VLM generates initial, task-agnostic descriptions for each image.

2. **Ambiguity Identification & Question Formulation:** A reasoning LLM analyzes the initial descriptions and the task goal (e.g., the Winoground captions) to identify the most critical, ambiguous visual detail needed for a confident decision. It then formulates a specific, targeted question about this detail.

3. **Focused Re-Perception:** The VLM is shown the relevant image again, but this time is asked to answer only the targeted question from the previous step.

4. **Synthesized Reasoning:** The LLM integrates the initial descriptions with the new, high-precision information from the Q&A step to make its final classification.

This interactive process allows the model to actively resolve perceptual ambiguities, moving beyond a single static "glance" to perform more robust, human-like visual verification.

| Framework | Approach | Interaction | Task-Specific | Performance |
|---|---|---|---|---|
| PRISM (Qiao et al., 2024) | Static skill tests | One-pass | Agnostic | N/A[†] |
| CoT (Zhang et al., 2024) | Task-conditioned desc. | One-pass | Yes | 64.25% (Wino.) |
| CA (Ours) | Task-agnostic desc. | Two-stage | Agnostic | **75.5%** (Wino.) |
| ICA (Ours) | Interactive feedback | Multi-turn | Agnostic | **78.0%** (Wino.) |

Table 1: Architectural comparison of evaluation frameworks. [†]PRISM evaluated on different benchmark suite. Performance shown for Winoground Text Score.

**Comparison with Existing Frameworks.** Table 1 provides a detailed comparison of our approach against related frameworks. Unlike PRISM's static skill inventory or task-conditioned CoT methods, our CA and ICA paradigms emphasize task-agnostic description generation combined with dynamic, interactive reasoning.

## 6 RESULTS AND ANALYSIS

This section details the performance of the evaluated VLMs, beginning with the primary Bongard-OW benchmark and then examining generalizability.

### 6.1 PERFORMANCE ON BONGARD-OW

Table 2 presents the core results on our 500-sample Bongard-OW subset. Under **Direct Visual Rule Learning (DVRL)**, applicable only to GPT-4o and Gemini 2.0, performance was strong but below optimal (Gemini 2.0: 82.2%, GPT-4o: 80.0%), suggesting limitations in purely holistic, simultaneous multi-image reasoning for this complex task.

Performance improved markedly under **DRL** for both models (GPT-4o: 88.0%, Gemini 2.0: 86.8%). The explicit separation of rule extraction and application stages appears beneficial, aligning with the idea that breaking down complex cognitive tasks can improve performance.

| Model | DVRL | DRL | CA |
|---|---|---|---|
| GPT-5.1[*] | 94.0 | 96.0 | 97.0 |
| GPT-4o | 80.0 | 88.0 | **92.8** |
| Gemini 2.0 | 82.2 | 86.8 | **93.6** |
| Pixtral-12B | - | - | 87.2 |
| Llama-Vision-11b | - | - | 53.4 |
| Llama-Vision-90b | - | - | 55.1 |
| Llava-7b | - | - | 66.2 |
| Llava-Llama3-8b | - | - | 53.2 |
| Gemma3:4b | - | - | 54.2 |
| Gemma3:12b | - | - | 59.2 |
| Gemma3:27b | - | - | 70.5 |
| Qwen2.5vl:32b | - | - | 50.0 |
| Prior SOTA | (GPT-4 + InstructBLIP) | | 63.8 |
| Human Average | (across samples) | | 91.0 |

Table 2: Classification accuracy (%) across evaluation paradigms on the Bongard-OW subset. Paradigms abbreviated: DVRL, DRL, CA. Dashes (-) indicate non-applicability due to model input limitations. (* smaller subset of 100 balanced samples for latest model validation.)

**Componential Analysis (CA)**, reason-
ing over textual descriptions, yielded the highest accuracies for the top models (GPT-4o: 92.8%, Gemini 2.0: 93.6%). Notably, these results establish a **new state-of-the-art (SOTA)** on the Bongard-OW Text-score benchmark, surpassing the reported human average (91.0% Wu et al. (2024)). The previous best machine performance Wu et al. (2024) involved using GPT-4 to reason over captions generated by models such as InstructBLIP, achieving a maximum accuracy of 63.8%. Our significant improvement with the CA paradigm underscores the efficacy of its comprehensive, task-agnostic description generation (Stage 1) coupled with advanced reasoning engines (Stage 2). Pixtral-12B also achieved strong CA performance (87.2%). However, a significant gap emerged with other open-source models. Models like Llama-Vision and LLaVA variants exhibited much lower CA accuracy, often with dramatic imbalances between positive and negative sample performance (e.g., Llava-Llama3-8B: 53.2% overall, heavily biased towards negative samples). This pattern strongly suggests that the bottleneck for these models is not necessarily the abstract reasoning itself, but rather the fidelity of their internal *visual perception* and subsequent translation into usable representations. (here, text descriptions).

The consistent trend of accuracy increasing from DVRL through DRL to CA for GPT-4o and Gemini 2.0 further reinforces the value of structured reasoning and, particularly, the effectiveness of the component-based textual reasoning approach for this task when perception is adequate.

## 6.2 PERFORMANCE ON BONGARD HOI

On Bongard-HOI, we evaluated GPT-4o and Gemini 2.0 across the four standard test splits (sosa, soua, uosa, uoua; N=100 each from balanced sampling). The results, shown in Table 3, largely replicated the trends observed on Bongard-OW. Performance systematically improved with increased paradigm structure (DVRL < DRL < CA) for both models. Our Componential Analysis (CA) paradigm, particularly with GPT-4o as the reasoning engine, achieved an average accuracy of 77.3% across the splits (with individual splits like sosa and soua reaching 83%), establishing a new state-of-the-art for VLM-based approaches on this benchmark. This surpasses prior SOTA Raghuraman et al. (2024) results from non-VLM specialized methods, such as the reported 76.4% average from a CLIP fine-tuned via PMF approach Raghuraman et al. (2024). Gemini 2.0 with CA also demonstrated strong competitive performance with an average of 74.5%. This consistency validates our framework's applicability and the benefit of structured evaluation across different complex natural image reasoning

| Model | | Avg |
|---|---|---|
| Gemini 2.0 | **DVRL** | 50.8 |
| | **DRL** | 61.3 |
| | **CA** | 74.5 |
| GPT-4o | **DVRL** | 68.5 |
| | **DRL** | 71.8 |
| | **CA** | 77.3 |
| Prior SOTA | PMF | 76.4 |
| Human Avg. | – | 91.4 |

Table 3: Average performance (%) on Bongard-HOI four test-splits across three paradigms.

datasets. Notably, overall model performance on HOI is lower than on OpenWorld, and a significant gap remains to the high human average scores (avg. 91.4% Jiang et al. (2022)), suggesting HOI's unique challenges in discerning subtle interaction-based rules.

## 6.3 PERFORMANCE ON WINOGROUND WITH STATIC CA

We applied our static CA paradigm to Winoground, generating task-agnostic descriptions and then using an LLM for matching. As shown in Table 4, this approach achieves new state-of-the-art results across all three metrics. Using GPT-4o as the reasoning engine yields Text: 75.5%, Image: 58.5%, and Group: 52.0%, significantly surpassing prior SOTA. This success demonstrates that our task-agnostic, decoupled strategy is highly effective not just for rule-discovery, but also for fine-grained compositional reasoning.

## 6.4 INTERACTIVE CA ON WINOGROUND: MITIGATING PERCEPTUAL GAPS

While CA is powerful, we hypothesized its static, one-pass descriptions might miss the single, subtle visual detail that differentiates Winoground pairs. To address this, we applied our Interactive CA (ICA) paradigm, allowing the reasoner to "look again" by asking a targeted question to the perception module.

The results, presented in Table 4, show a significant and consistent performance uplift over the static CA approach for both models tested.

| Model | Text Score | Image Score | Group Score |
|---|---|---|---|
| **GPT-4o + CA** | **75.50** | **58.50** | **52.00** |
| Gemini 2.0 + **CA** | 71.00 | 48.75 | 42.00 |
| **GPT-4o + ICA** | **78.00** | **62.75** | **55.25** |
| Gemini 2.0 + **ICA** | 72.50 | 55.25 | 46.75 |
| Llama3.3-70B + **CA** | 68.25 | 49.25 | 41.75 |
| Qwen2.5-32B + **CA** | 67.00 | 46.25 | 40.00 |
| Phi-4-14B + **CA** | 65.25 | 46.00 | 37.75 |
| Qwen2.5-14B + **CA** | 59.25 | 34.50 | 27.25 |
| MMICL + CoCoT (Zhang et al., 2024) | 64.25 | 52.5 | 50.75 |

Table 4: State-of-the-art performance on the **Winoground benchmark** achieved using our **Componential Analysis (CA)** paradigm. Scores reported are the standard Winoground metrics: *Text Score* (correct caption selection per image description), *Image Score* (correct image selection per caption), and *Group Score* (all selections correct per sample), averaged over 400 samples.

Critically, the largest gains are on the Image and Group scores, which are most sensitive to fine-grained visual details. For instance, Gemini's Image Score improved by a remarkable 6.5 points. This demonstrates that the interactive feedback loop is highly effective at resolving the exact visual ambiguities that Winoground is designed to test. This finding confirms that the perception bottleneck is not immutable; it can be actively mitigated by a dynamic, multi-pass reasoning process that guides perception.

In summary, across diverse reasoning tasks, our Componential Analysis paradigms consistently achieve high and often state-of-the-art performance. The success of the interactive ICA variant further highlights that dynamic, decoupled approaches—where reasoning can actively probe perception—are a powerful and promising direction for building more robust and accurate VLMs.

## 7    ABLATION STUDIES: ISOLATING PERCEPTION AND REASONING

To further investigate the interplay between visual perception, rule representation, and reasoning, we conducted targeted ablation studies. Both studies presented below serve to underscore the critical role of the initial representation derived from visual input – whether it's applying a rule *to* a perceived query image (Section 7.1) or reasoning *from* perceived context images (Section 7.2).

### 7.1    RULE APPLICATION FIDELITY

How well can models apply an abstract rule once it's formulated? To isolate rule application from rule extraction, we provided models with high-quality rule summaries (generated by GPT-4o) and the query image, tasking them solely with classification based on the given rule. This tests the model's ability to ground the symbolic rule in the visual input of the query image.

Table 5 shows performance for several open-source models under this condition. Models like Pixtral-12B demonstrate relatively strong and balanced rule application. Comparing this to Table 2, the generally higher scores here than in CA (where models generate their own descriptions) support the idea that rule application itself is less challenging for these models than the initial perception/description phase.

| Model | Acc |
|---|---|
| LLaVA-7B + **DRL** | 72.0 |
| Llama-vision-11B + **DRL** | 68.2 |
| Pixtral-12B + **DRL** | 83.8 |
| LLaVA-13B + **DRL** | 70.0 |
| LLaVA-34B + **DRL** | 74.8 |
| Llama-vision-90B + **DRL** | 74.2 |

Table 5: **Rule Application Accuracy:** Accuracy (Acc) in % when classifying query images based on externally provided rules in DRL paradigm.

## 7.2 IMPACT OF DESCRIPTION QUALITY ON REASONING

Complementing the previous ablation, we investigated how reasoning performance changes when the initial perceptual stage (description generation) is standardized using a high-fidelity source. We generated descriptions for all context and query images using GPT-4o and then used these descriptions as input to the reasoning stage (Stage 2) of the Componential Analysis paradigm for various target models, including weaker VLMs and even text-only LLMs.

The results in Table 6 were revealing. Providing high-quality descriptions dramatically improved the reasoning accuracy of VLMs that struggled when using their own descriptions. Llama-Vision-11B, for example, improved from 53.4% (Table 2) to 84.17%, and Llama-Vision-90B from 55.1% to 90.98%. This provides strong evidence that the reasoning capabilities of these models are significantly underestimated by end-to-end evaluations; their primary limitation lies in generating accurate perceptual representations. Further illustrating this sensitivity to description source quality, Table A.5 in the Appendix details a comparison using components generated by Pixtral-12B.

| Model | Acc |
|---|---|
| Gemma3:4b + **CA** | 77.35 |
| Gemma3:12b + **CA** | 87.58 |
| Llava:7B + **CA** | 80.56 |
| Llava:34B + **CA** | 81.56 |
| Llama-vision:11B + **CA** | 84.17 |
| Qwen2.5vl:3b + **CA** | 86.77 |
| Llama-vision:90B + **CA** | 90.98 |
| Gemma3:27b + **CA** | 89.78 |
| Qwen2.5vl:32b + **CA** | 92.79 |
| Deepseek-r1:14b + **CA** | 87.98 |
| Gemma2:27b + **CA** | 88.98 |
| Qwen2.5:7b + **CA** | 90.38 |
| Phi4:14b + **CA** | 91.98 |
| Qwen2.5:32b + **CA** | 92.79 |
| Qwen2.5:14b + **CA** | 92.99 |

Table 6: **Impact of High-Quality Descriptions**: Accuracy (Acc) in % using Componential Analysis (Stage 2 reasoning) with image descriptions generated externally by GPT-4o. Includes VLMs and LLMs models respectively separated by line.

Remarkably, this approach also enabled text-only LLMs to perform the visual reasoning task effectively. Models such as Phi-4 (14B) achieved 91.98% accuracy, while several Qwen models also exceeded 90%. This demonstrates that: (1) High-quality textual descriptions can serve as effective surrogates for visual input, enabling modality transfer for reasoning tasks. (2) The CA paradigm, particularly when coupled with controlled descriptive input, serves as a powerful tool for isolating and evaluating the core symbolic reasoning abilities of both VLMs and LLMs, independent of their integrated perceptual systems. These findings strongly reinforce the conclusion that improving visual perception is paramount for enhancing end-to-end visual reasoning in many current models.

## 7.3 ADDITIONAL ANALYSIS

Rule synthesis quality is validated through semantic similarity analysis and manual error categorization (Appendix A.9).

**Semantic Similarity Analysis** Experimentation during DRL (Table A.2) confirmed that derived rules generally aligned well with query descriptions, particularly for positive samples. The relatively high similarity for negative samples highlights the challenge of the dataset's near-miss counterexamples.

**Robustness to Noisy and Adversarial Samples.** We conducted a preliminary robustness evaluation A.12.4 using Gaussian noise (intensity: 0.08) applied to Bongard-OW images. Results reveal a critical advantage of our decomposed approach: **DVRL** (holistic processing) degrades severely to 31 points inaccuracy, while **CA** (componential analysis) maintains similar performance. This suggests that **our two-stage architecture provides inherent robustness**: noisy image details are abstracted into textual descriptions (Stage 1), and reasoning operates over robust semantic representations (Stage 2) rather than pixel-level features.

**Qualitative Error Analysis** Examining samples misclassified by both top models (GPT-4o, Gemini 2.0) under CA revealed recurring error patterns (details in Appendix A.10.7, Table A.8). Frequent issues involved over-generalizing rules, missing critical objects/properties present in positive

examples, focusing on spurious correlations, or failing to consistently apply derived rules. These qualitative examples underscore that even highly capable models exhibit fragility in nuanced visual detail processing and robust symbolic rule manipulation.

## 8    DISCUSSION

This research leveraged a cognitively-inspired framework to dissect the mechanisms of visual reasoning in VLMs. By evaluating performance across paradigms mirroring human cognitive strategies (holistic-*DVRL*, deductive-*DRL*, analytical-*CA*), we moved beyond aggregate scores to probe how these models process complex information. A central finding emerges: a critical perception bottleneck limits many contemporary VLMs, masking strong downstream reasoning capabilities. While advanced models (GPT-4o, Gemini 2.0) excel when reasoning over high-fidelity textual descriptions, many open-source models falter, pointing to failures in reliably extracting relevant visual information.

The success of the CA paradigm is particularly insightful. Its strength lies in effectively decoupling perception from reasoning via task-agnostic descriptions. Unlike context-dependent CoT approaches, CA first builds a comprehensive, independent textual "world model" of each image. This allows powerful LLMs to apply their sophisticated reasoning abilities on a clean, symbolic representation. The resulting robust performance across different reasoning types (abstraction in BPs, compositionality in Winoground) suggests that modular architectures—featuring specialized perception modules that output rich symbolic data for general reasoning engines—are a highly promising direction.

Our CA paradigm prioritizes **task-agnosticism**: Stage 1 (perception) receives only images and domain-general schema (Scene, Objects, Activities) *without knowledge of the reasoning task*. While this schema reflects natural-image structure, it does not encode task-specific information (rule domains, caption pairs, HOI labels). Detailed discussion of task-agnosticism vs. domain-general schemas appears in Appendix A.8.

The choice of hierarchical JSON format is motivated by empirical evidence that structured representations significantly enhance reasoning performance. Our ablation study (Appendix A.10.6, Table A.7) demonstrates that structured prompting improves DVRL accuracy from 61.6% to 80.0% (+18.4 points), confirming that LLMs reason more effectively with organized, hierarchical information. The JSON schema provides: (1) explicit entity identification and tracking across images, (2) hierarchical organization mirroring analytical decomposition (Gluck et al., 2008), and (3) systematic attribute binding critical for compositional reasoning tasks like Winoground.

The success of our Interactive CA (ICA) paradigm takes this insight a crucial step further. While static CA bypasses the bottleneck, ICA demonstrates how to actively mitigate it. On a fine-grained task like Winoground, where a single detail is critical, the ability for the reasoner to formulate a targeted question and prompt a "second look" from the perception module led to significant performance gains (Table 4). This result is pivotal: it suggests the perception bottleneck is not an immutable property but often a failure of one-pass processing. By modeling a more realistic, iterative process of verification—akin to human visual scanning—ICA shows that the connection between perception and reasoning modules should not be a one-way street, but a dynamic, bidirectional dialogue.

While CA demonstrates that static decoupling unlocks reasoning, Interactive CA (ICA) proves that bidirectional feedback resolves fine-grained visual ambiguities single-pass descriptions miss (Table 4). ICA achieves significant improvements on Winoground, suggesting the perception bottleneck is not immutable but reflects insufficient focus. Architectural implications and future extensions are discussed in Appendix A.13.

Our ablation studies provide strong converging evidence. Isolating reasoning by providing high-quality external descriptions (Section 7.2) resulted in dramatic performance improvements for bottlenecked models and enabled high performance even from text-only LLMs. This clearly demonstrates that a model's latent reasoning capability often far exceeds what its end-to-end performance suggests.

In conclusion, our cognitively-inspired framework serves a dual purpose. It is both a valuable diagnostic tool for pinpointing the prevalent perception bottleneck and a proof-of-concept for a more powerful architectural class. The success of our componential, and especially our interactive, paradigms reveals the significant latent reasoning potential within today's models. More importantly,

it offers a clear path forward: building modular, interactive systems where reasoning can dynamically guide perception is a key step towards more robust and general visual intelligence.

**Limitations** Our work has several important limitations that define avenues for future research.

The primary limitation of our componential paradigms (CA and ICA) (expanded in Appendix A.12 is their reliance on language as an intermediate representation. Their effectiveness is likely highest on tasks where critical visual properties are readily verbalizable. For challenges that hinge on non-verbalizable or geometric reasoning—such as the fine-grained correspondence tasks in benchmarks like BLINK (Fu et al., 2024)—the utility of a purely text-mediated approach may be reduced. While our interactive ICA shows that a dynamic dialogue can resolve ambiguities missed by static descriptions, its scope is still bounded by what can be effectively queried and described in text.

Second, while our study covers abstract, compositional, and interactive reasoning, the framework's applicability to other complex visual domains, such as scientific chart interpretation or mathematical reasoning, requires further investigation.

Finally, we acknowledge several practical scope limitations. This work did not conduct a systematic analysis of prompt sensitivity, a known factor in VLM performance. A deeper investigation into the computational costs and latency trade-offs of our multi-stage paradigms, especially the interactive ICA, is also warranted for practical application.

## 9 CONCLUSION

This paper introduced a cognitively-inspired framework to dissect the perception-reasoning interface in VLMs. Through four distinct paradigms—including our novel Interactive Componential Analysis (ICA)—we systematically analyzed VLM problem-solving strategies, revealing two key insights. First, our diagnostic approach confirms that a critical perception bottleneck limits many contemporary VLMs, masking significant latent reasoning abilities.

Second, and more importantly, we demonstrate a powerful architectural solution. Our componential paradigms, which decouple perception from reasoning via task-agnostic textual descriptions, achieve highly competitive performance across diverse benchmarks testing abstraction (Bongard-OW), interaction (Bongard-HOI), and compositionality (Winoground). The success of the interactive ICA paradigm, which allows the reasoning module to actively probe and guide perception, is particularly significant. It shows that the perception bottleneck is not an immutable barrier but can be dynamically mitigated.

Ultimately, our work suggests that the path toward more robust visual intelligence lies not just in scaling monolithic models, but in developing modular, interactive architectures. By providing both a diagnostic toolkit and a proof-of-concept for this interactive approach, we offer a blueprint for a new class of systems capable of more deliberate, verifiable, and human-like visual reasoning.

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

# APPENDIX

## A.1 BROADER RELEVANCE

This study offers insights with broader implications for developing more robust and human-like AI systems. Our cognitively-inspired evaluation paradigms provide valuable tools for assessing and understanding the strengths and limitations of Vision-Language Models (VLMs) on complex visual reasoning tasks. The insights gained extend beyond Bongard problems, contributing to the development of VLMs capable of advanced reasoning in real-world applications. Our key finding regarding the visual processing bottleneck in many models has significant implications for future research aimed at bridging the performance gap and unlocking the full potential of accessible models. The demonstration of high performance by advanced VLMs underscores the potential for sophisticated visual understanding, reinforcing the importance of architectures integrating robust perception and reasoning. Finally, our comparative evaluation contributes to discussions about AI accessibility and transparency, identifying specific areas for improvement and paving the way for more reliable AI.

## A.2 ATTENTION AND MEMORY IN VISUAL REASONING

While our study primarily focuses on the interplay between perception and reasoning, the roles of attention and memory are also implicitly present in our paradigms. The DVRL paradigm likely engages VLM "visual attention" mechanisms (Bahdanau et al., 2016) to identify salient features across the image set, akin to human holistic processing (Biederman, 1987; Li et al., 2002). DRL relies on the model's ability to "memorize" the extracted rule, involving processes related to working memory (Baddeley, 2012) and internal representation storage (Squire, 1992). Although not directly measured, their involvement is inherent. Future work could explore these aspects more explicitly, perhaps via attention map analysis (Vaswani et al., 2017) or probing memory representations (Vaishnav and Serre, 2023).

## A.3 REPRODUCIBILITY DETAILS

This section consolidates all information necessary to replicate our experimental results.

### A.3.1 MODEL VERSIONS

1. **GPT-4o**: gpt-4-turbo-2024-04-09 (OpenAI API, accessed Sept 2024)
2. **Gemini 2.0**: gemini-2.0-flash-exp (Google AI Studio, Dec 2024 release)
3. **Pixtral-12B**: mistralai/pixtral-12b-2409 (HuggingFace)
4. **Llama-Vision**: meta-llama/Llama-3.2-11B-Vision-Instruct, meta-llama/Llama-3.2-90B-Vision-Instruct
5. **LLaVA**: liuhaotian/llava-v1.5-7b, liuhaotian/llava-v1.5-13b, liuhaotian/llava-v1.6-34b
6. **Text-only LLMs**: Phi-4-14B (microsoft/phi-4), Qwen2.5 (7B/14B/32B), DeepSeek-R1-14B, Gemma2-27B

### A.3.2 HARDWARE CONFIGURATION

- **Proprietary models** (GPT-4o, Gemini 2.0): Cloud-based API calls, no local GPU
- **Open-source VLMs** (Llama-Vision, Pixtral, LLaVA): NVIDIA RTX 6000 Ada (48GB VRAM)
- **Text-only LLMs**: NVIDIA RTX 3090 (24GB VRAM) for smaller models (<32B parameters)

### A.3.3 HYPERPARAMETERS

All evaluations used fixed settings for consistency:

- **Temperature**: 0 (deterministic decoding for reproducibility)
- **Max tokens**: 2000 (image descriptions), 500 (reasoning outputs)
- **Batch size**: 1 (sequential processing)
- **Image resolution**: API defaults (GPT-4o, Gemini) or 1024px max (Ollama-served models)
- **Retry logic**: 3 attempts for API timeout/rate-limit errors

### A.3.4 DATASET SUBSETS

**Bongard-OW**: First 250 samples from full dataset, generating 2 test cases each (1 positive query, 1 negative query) = 500 balanced cases.

**Bongard-HOI**: 100 balanced samples per split (50 pos/50 neg queries) from each of 4 standard test splits (SOSA, SOUA, UOSA, UOUA) = 400 total cases.

**Winoground**: All 400 samples from standard release.

### A.3.5 EVALUATION PIPELINE

1. Load dataset subset with specified sample IDs
2. For each sample, apply paradigm-specific prompts (Appendix A.6)
3. Extract classification decision via regex matching (`cat_0`, `cat_1`, or `positive/negative`)
4. Compute accuracy as $\frac{\text{\# correct classifications}}{\text{total samples}}$
5. For Winoground, compute Text/Image/Group scores per Equations 1–3

### A.3.6 CODE AVAILABILITY

Evaluation scripts, prompts, and dataset splits will be released upon acceptance.

## A.4 DATASET DETAILS

### A.4.1 BONGARD OPENWORLD DATASET

We utilize a subset of 500 test cases from the Bongard OpenWorld dataset (Wu et al., 2024). The full dataset contains 1001 samples, each with 7 positive and 7 negative real-world images distinguished by a "commonsense" rule. Our evaluation set was created by taking the first 250 samples and generating two test cases from each (one positive query, one negative query), resulting in 500 balanced test cases. Specific sample IDs used will be released.

#### A.4.1.1 COMMONSENSE VALUE DISTRIBUTION: SUBSET VS. FULL DATASET

Table A.1 shows the distribution in our subset. Category (visual concept) '0' is predominant.

To validate the representativeness of our 500-case subset, Table A.1 compares the category distribution in our evaluation subset against the full Bongard-OW dataset (1001 samples). The distributions are nearly identical, confirming that our subset maintains the original dataset's category (visual concept) balance.

**Implications for Validity:**

- The identical distributions confirm that our subset is not biased toward specific rule types
- Results generalize to the full dataset (no sampling artifacts)
- Category-level performance (Table A.7) shows consistent accuracy across all 10 categories, including minority classes (Meta Class: 100%, Relationship: 100%)

**Cross-Dataset Validation:** Our findings generalize beyond this subset, as evidenced by consistent paradigm trends on Bongard-HOI (entirely different dataset, 400 samples) and Winoground (400 samples, different task modality). The 29-point improvement over prior SOTA ($63.8\% \rightarrow 92.8\%$) establishes robust evidence regardless of subset size.

| ID | Category | Count | Our % | Orig. % | Example |
|---|---|---|---|---|---|
| 0 | Anything else | 365 | 73.0 | 73.4 | Strawberry leaves. A perched mantis... |
| 1 | HOI | 15 | 3.0 | 2.7 | Persons riding bicycles. People holding flag... |
| 2 | Taste/Nutrition/Food | 14 | 2.8 | 2.5 | Grilled steaks. Chocolate pudding... |
| 3 | Color/Material/Shape | 36 | 7.2 | 5.7 | Steel beams. Ceramic bowl. Leather bags... |
| 4 | Functionality/Status | 12 | 2.4 | 3.0 | Birds soaring. Balloon floating... |
| 5 | And/Or/Not | 10 | 2.0 | 1.5 | Room without people. Wedding photos... |
| 6 | Factual Knowledge | 10 | 2.0 | 2.8 | Capital of US. Egyptian pyramids... |
| 7 | Meta Class | 4 | 0.8 | 2.0 | Canine animals. Marine animals... |
| 8 | Relationship | 8 | 1.6 | 1.9 | Coral reefs. Feather falls... |
| 9 | Unusual observations | 26 | 5.2 | 4.7 | Clear lake bottom. Moonlight reflected... |
| | **Total** | **500** | **100.0** | **100.0** | |

Table A.1: Commonsense category (visual concept) distribution in our Bongard-OW subset vs. original dataset. Our sampling closely matches the original distribution, with notable differences in Meta Class (0.8% vs 2.0%) and Factual Knowledge (2.0% vs 2.8%).

### A.4.2 BONGARD-HOI DATASET

To assess generalizability on natural images with a different reasoning focus (human-object interactions), we used the Bongard-HOI dataset (Jiang et al., 2022). We evaluated performance on its four standard test splits, defined by object/action novelty:

- sosa: seen object, seen action
- soua: seen object, unseen action
- uosa: unseen object, seen action
- uoua: unseen object, unseen action

The original splits vary significantly in size and balance (e.g., sosa: 200 pos/200 neg queries; soua: 2236 pos/1348 neg; uosa: 660 pos/660 neg; uoua: 695 pos/695 neg). For consistent cross-split evaluation in this work, we created balanced subsets by sampling 100 test cases from each of the four splits, ensuring an equal distribution of 50 positive and 50 negative query images per split. This resulted in a total evaluation set of 400 samples for Bongard-HOI (100 per split), used for the results reported in Table 3.

### A.4.3 WINOGROUND DATASET

To test performance on fine-grained visio-linguistic compositional reasoning, we utilized the Winoground dataset (Thrush et al., 2022). This dataset comprises 400 samples specifically designed to challenge compositional understanding. Each sample contains a pair of minimally contrastive images $(I_0, I_1)$ and a corresponding pair of minimally contrastive captions $(C_0, C_1)$, requiring models to correctly match image $I_0$ to caption $C_0$ and image $I_1$ to caption $C_1$. We used all 400 samples provided in the standard dataset release for our Winoground evaluations reported in Section 6.3 and Table 4.

### A.4.4 DATASET AVAILABILITY

Bongard OpenWorld: https://rujiewu.github.io/Bongard-OW.github.io/.
Bongard-HOI: https://github.com/NVlabs/Bongard-HOI/blob/master/assets/dataset.md.
Winoground: https://huggingface.co/datasets/facebook/winoground

Details on the specific subsets and samples used in our evaluations will be released upon publication.

## A.5 MODEL AND EXPERIMENT DETAILS

### A.5.1 MODEL DETAILS

**VLMs:** GPT-4o; Gemini 2.0; Pixtral-12B; Llama-Vision-3.2 (11B, 90B); LLaVA (Llama-2 based; 7B, 13B, 34B); LLaVA-Llama3-8B. **Text-Only LLMs (for Ablation 7.2):** Phi-4 (14B) (Abdin et al., 2024); Qwen2.5 (7B, 14B, 32B) (Yang et al., 2024); Deepseek-r1 (32B, 70B) (Guo et al., 2025); Gemma2 (27B) (Team et al., 2024).

### A.5.2 EXPERIMENT CONFIGURATION

- **Access:** APIs for closed models; Ollama for open models.
- **Input:** Base64 images in prompts (see Appendix A.6).
- **Image Handling:** API defaults or max 1024px (Ollama). Multi-image calls for DVRL where supported.
- **Decoding:** Temperature 0.
- **Fine-tuning:** None.
- **Hardware:** NVIDIA GPUs (2080Ti, 3090, 6000 Ada).

### A.5.3 EVALUATION METRICS

- **Classification Accuracy:** Primary metric (% correct).
- **Semantic Similarity:** Cosine similarity of OpenAI embeddings ('text-embedding-3-large') between descriptions/rules. Inspired by (Risch et al., 2021).

### A.5.4 WINOGROUND SCORE CALCULATION USING COMPONENTIAL ANALYSIS

This section defines the calculation of Winoground (Thrush et al., 2022) scores (`text_score`, `image_score`, `group_score`) within our Componential Analysis (CA) paradigm (Section 6.2).

In the standard Winoground task, a sample $i$ consists of two images $I_{0,i}, I_{1,i}$ and two captions $C_{0,i}, C_{1,i}$, where $(C_{0,i}, I_{0,i})$ and $(C_{1,i}, I_{1,i})$ are the ground truth correct pairs. Models are typically evaluated based on a scoring function $s(C, I)$ indicating the match between a caption and an image.

In our CA paradigm, the reasoning model (Stage 2) does not access images $I_{0,i}, I_{1,i}$ directly. Instead, it operates on textual image descriptions $D_{0,i}, D_{1,i}$ generated in Stage 1. The model is prompted to make explicit choices about the best match between descriptions and captions. Let $Choice_C(D_k, \{C_0, C_1\})$ denote the caption ($C_0$ or $C_1$) chosen by the model as the best match for description $D_k$. Similarly, let $Choice_D(C_k, \{D_0, D_1\})$ denote the description ($D_0$ or $D_1$) chosen for caption $C_k$.

The scores for each sample $i$ in the dataset $W$ (where $N = |W| = 400$) are calculated as follows:

**1. Text Score** ($f_{CA}$)**:** This measures if the correct caption is selected for each image description. We use an indicator function $\mathbb{I}[\cdot]$ which is 1 if the condition inside is true, and 0 otherwise.

$$f_{CA}(i) = \mathbb{I}\left[\begin{array}{c} Choice_C(D_{0,i}, \{C_{0,i}, C_{1,i}\}) = C_{0,i} \\ and \\ Choice_C(D_{1,i}, \{C_{0,i}, C_{1,i}\}) = C_{1,i} \end{array}\right] \quad (1)$$

This score is 1 only if the model correctly identifies the caption for *both* description $D_{0,i}$ and description $D_{1,i}$.

**2. Image Score** ($g_{CA}$)**:** This measures if the correct image description is selected for each caption.

$$g_{CA}(i) = \mathbb{I}\left[\begin{array}{c} Choice_D(C_{0,i}, \{D_{0,i}, D_{1,i}\}) = D_{0,i} \\ and \\ Choice_D(C_{1,i}, \{D_{0,i}, D_{1,i}\}) = D_{1,i} \end{array}\right] \quad (2)$$

This score is 1 only if the model correctly identifies the description for *both* caption $C_{0,i}$ and caption $C_{1,i}$.

**3. Group Score ($h_{CA}$):** This requires all associations within the sample to be correct.

$$h_{CA}(i) = f_{CA}(i) \wedge g_{CA}(i) \tag{3}$$

Equivalently, $h_{CA}(i) = 1$ if and only if $f_{CA}(i) = 1$ and $g_{CA}(i) = 1$.

## A.6 MODEL PROMPTS

### A.6.1 DIRECT VISUAL RULE LEARNING

The prompt used for the Direct Visual Rule Learning paradigm is designed to elicit a holistic analysis of the provided images, encouraging the model to identify a distinguishing rule and apply it to the query image. The prompt emphasizes the distinction between positive ($cat\_2$) and negative ($cat\_1$) examples and guides the model to provide a structured output containing its analysis, the identified rule, details about the query image, and the final classification.

```python
def visual_concept_test_prompt(m, n):
    """
    Generates a visual analysis prompt.

    Args:
        m (int): Number of positive samples.
        n (int): Number of negative samples.

    Returns:
        str: The formatted prompt string.
    """
    return f"""
    You are provided with {m + n + 1} images: the first {m} samples are
        `cat_2`, the next {n} samples are `cat_1`, and the last image is
        the `query image`.
    Analyze the common characteristics or patterns found in the `cat_2`
        samples (positive samples: following 1 common rule) that
        distinctly separate them from the `cat_1` samples (negative
        samples: it might not follow any possible rule).
    Your task is to:

    1. Determine the rule or criterion that distinguishes the `cat_2`
        samples from the `cat_1` ones.
    2. Analyse the `query image` (last image).
    3. Provide your conclusion for the `query image` if it can be
        categorized as either `cat_1` or `cat_2` based on the analysis
        and the rule.

    Ensure that the output is clear, well-formatted, and free of
        unnecessary explanations.
    Omit the ``` tags at the beginning and end of the page. The format
        of your output should be as follows:

    - **Analysis**: (Your analysis here)
    - **Rule**: (The distinguishing rule here)
    - **Query Image**: (Query image details)
    - **Conclusion**: (cat_1 or cat_2)
    """
```

### A.6.2 DEDUCTIVE RULE LEARNING

The Deductive Rule Learning paradigm employs a two-stage prompting strategy. The first stage focuses on rule extraction from positive and negative examples, while the second stage applies the extracted rule to classify a query image. The prompts for each stage are detailed below.

### A.6.2.1 First-Stage Prompt (Rule Extraction)

This prompt guides the model to identify and summarize a distinguishing rule based on provided positive and negative examples. It emphasizes conciseness in the rule summary.

```python
def visual_concept_prompt(m, n):
try:
    if m < 0 or n < 0:
        raise ValueError(f"Invalid input: m and n must be
            non-negative. Received m={m}, n={n}.")

    if m > 0 and n > 0:
        prompt = f"""
            You are provided with {m + n} images: the first {m}
                samples are cat_2, the next {n} samples are cat_1.
                Analyze the common characteristics or patterns found
                in the cat_2 samples (positive samples: following 1
                common rule) that distinctly separate them from the
                cat_1 samples (negative samples: it might not follow
                any possible rule).
            Your task is to provide the rules that defines cat_2
                samples. At the end, write "summary" of the rule
                identified in less than 20 words.
            Ensure that the output is clear, well-formatted, and
                free of unnecessary explanations. Omit the ''' tags
                at the beginning and end of the page.
            """
    if n == 0:
        prompt = f"""
            You are provided with {m} images: {m} samples are cat_2.
                Analyze the common characteristics or patterns found
                in the cat_2 samples (positive samples: following 1
                common rule) that distinctly separate them from
                negative samples which might not follow any possible
                rule.
            Your task is to provide the rules that defines cat_2
                samples. At the end, write "summary" of the rule
                identified in less than 20 words.
            Ensure that the output is clear, well-formatted, and
                free of unnecessary explanations. Omit the ''' tags
                at the beginning and end of the page.
            """
    return prompt

except ValueError as e:
    print(f"Error: {e}")
    raise
```

### A.6.2.2 Second-Stage Prompt (Rule Application)

This prompt presents the previously extracted rule summary and a query image, prompting the model to classify the image based on the rule. It reinforces the Bongard problem context and requests a structured output.

```python
# Define the visual analysis prompt
def visual_concept_test_prompt(m, n, summary):
    return f"""
    We are working with Bongard dataset where there are {m} image in the
        cat_2 and {n} images in the cat_1. Summary of the common
        characteristics or patterns found in the cat_2 samples (positive
        samples: following 1 common rule) that distinctly separate them
        from the cat_1 samples (negative samples: it might not follow
        any possible rule) is as follows: \n {summary}.
```

```
Your task is to ponder over the rule and provide your conclusion for
    the 'query image' if it can be categorized as either "cat_1" or
    "cat_2".

Ensure that the output is clear, well-formatted, and free of
    unnecessary explanations.
Omit the ''' tags at the beginning and end of the page. The format
    of your output should be as follows:

- **Analysis**: (Your analysis here)
- **Rule**: (The distinguishing rule here)
- **Query Image**: (Query image details)
- **Conclusion**: (cat_1 or cat_2)
"""
```

### A.6.3    COMPONENTIAL ANALYSIS

The Componential Analysis paradigm also uses a two-stage prompting strategy. The first stage generates detailed image descriptions, while the second stage derives a rule from these descriptions and applies it to a query image. The specific prompts for each stage are presented below.

#### A.6.3.1    FIRST-STAGE PROMPT (IMAGE DESCRIPTION GENERATION)

This prompt instructs the model to generate a comprehensive, hierarchical description of a given image in JSON format. It guides the model to cover various aspects of the image, from scene and objects to activities and contextual elements, facilitating detailed comparative analysis in the subsequent stage.

```
# Define the visual analysis prompt
def visual_concept_prompt():
    """
    Generates a visual analysis prompt.

    Args:

    Returns:
        str: The formatted prompt string.
    """
    return """
            Carefully examine the provided image and identify all
                possible visual elements, organizing them into a
                detailed hierarchical structure. Start with broad
                categories and progress to more specific subcategories.
                This should cover everything visible in the image,
                ensuring no detail is overlooked. Structure your
                findings in a JSON format to enable easy comparison and
                synthesis of data from other images. This will help
                discern patterns, contexts, and rules valuable for
                identifying or understanding query images.

            Your hierarchy might encompass the following elements:

            1. **Scene/Environment**: Description of the overall setting
                depicted, such as urban, natural, indoor, or outdoor
                scenes.
            2. **Objects**: Define distinct items or entities present in
                the scene.
            - **Living Beings**: Animals, humans, or other biological
                entities.
                - Species or classification (e.g., dog, bird, human).
                - Characteristics (e.g., color, posture, movement).
            - **Inanimate Objects**: Both synthetic and natural elements.
                - Categories (e.g., vehicle, building, trees).
```

```
                            - Properties (e.g., color, size, material, shape).
                        3. **Activities**: Observable actions or interactions
                            involving any objects or beings.
                        - Specific descriptions of actions (e.g., walking, flying).
                        - Participants involved in these actions.
                        4. **Contextual Elements**: Environmental conditions and
                            time markers, such as time of day or weather.
                        - Detailed characteristics (e.g., cloudy, night, winter).
                        5. **Visual Patterns**: Prominent colors, textures, and
                            patterns that are visually significant.
                        6. **Emotional Undertones**: Any emotional presence or
                            expressions evident in the image.
                        7. **Textual Information**: Any visible text within the
                            image, including what it says and its visual style.
                        8. **Summary**: A concise narrative summarizing the overall
                            content and context of the image.

                        Ensure that every aspect from the image is represented under
                            these categories. The information should be presented in
                            the following JSON format:

                        {
                        "Scene": {
                            "Description": "..."
                        },
                        "Objects": {
                            "Living Beings": [...],
                            "Inanimate Objects": [...]
                        },
                        "Activities": [...],
                        "Contextual Elements": {
                            "Time of Day": "...",
                            "Weather": "..."
                        },
                        "Visual Patterns": {
                            "Dominant Colors": [...],
                            "Textures": [...]
                        },
                        "Emotional Undertones": "..."
                        "Textual Information": "..."
                        "Summary": "..."
                        }
                        Ensure that the JSON output is clear, well-formatted, and
                            free of unnecessary explanations. Omit the ```json tags
                            at the beginning and end of the page.
                        """
```

### A.6.3.2 SECOND-STAGE PROMPT (RULE DERIVATION INSTRUCTION)

This prompt guides the model to analyze the JSON descriptions generated in the first stage, derive a distinguishing rule, and apply it to classify a query image. It emphasizes the use of the provided JSON format and requests a structured output.

```
def user_eval_prompt(all_image_specs, m, n):
    return f"""
        We are working with the Bongard dataset, which contains {m}
            images in cat_2 (positive samples) and {n} images in cat_1
            (negative samples). These categories are defined as follows:
        - Cat_2: Positive samples that follow a single common rule.
        - Cat_1: Negative samples that may not follow any specific rule.

        The image descriptions for the positive samples, negative
            samples, and the test image are provided in JSON format.
```

```
            Analyze the common patterns or characteristics in the cat_2
            samples that distinguish them from cat_1 samples.

        Your task is to:
        1. Derive the rule that defines the cat_2 samples.
        2. Apply this rule to categorize the test image.

        Here are the image descriptions:

        ### Positive Samples (cat_2):
        {all_image_specs[:m]}

        ### Negative Samples (cat_1):
        {all_image_specs[m:m+n]}

        ### Test Image:
        {all_image_specs[-1]}

        Provide your output in the following format:

        - **Analysis**: (Your analysis here)
        - **Rule**: (The distinguishing rule here)
        - **Test Image**: (Test image details)
        - **Conclusion**: (cat_1 or cat_2)
        """
```

### A.6.4 EXAMPLE JSON DESCRIPTIONS FROM CA STAGE 1

To illustrate the task-agnostic, hierarchical descriptions generated in CA Stage 1, we provide three representative examples from Bongard-OW evaluations:

**Example 1: Positive Sample**

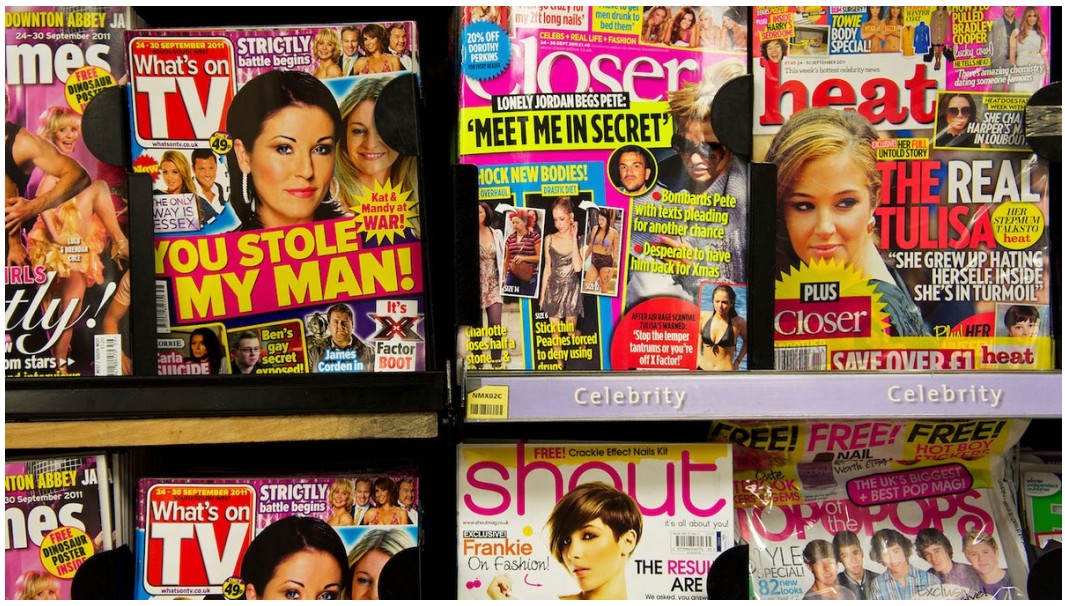

Figure A.1: **Positive Image Example.**

```
{
    "Scene": {
        "Description": "A display of various celebrity and entertainment
            magazines on a shelf."
```

```
        },
        "Objects": {
            "Living Beings": [
                {
                    "Classification": "Human",
                    "Characteristics": "Images of people on magazine covers"
                }
            ],
            "Inanimate Objects": [
                {
                    "Category": "Magazines",
                    "Properties": [
                        "Colorful covers",
                        "Various titles and headlines"
                    ]
                },
                {
                    "Category": "Shelf",
                    "Properties": [
                        "Wooden or metal",
                        "Labeled 'Celebrity'"
                    ]
                }
            ]
        },
        "Activities": [
            "Display of magazines for sale"
        ],
        "Contextual Elements": {
            "Time of Day": "Indeterminate",
            "Weather": "Indoors"
        },
        "Visual Patterns": {
            "Dominant Colors": [
                "Red",
                "Yellow",
                "Pink",
                "Black"
            ],
            "Textures": [
                "Glossy magazine covers"
            ]
        },
        "Emotional Undertones": "Sensational and dramatic headlines
            suggesting gossip and intrigue.",
        "Textual Information": "Various magazine titles and headlines such
            as 'What's on TV', 'Closer', 'heat', 'shout', and 'Top of the
            Pops'.",
        "Summary": "The image depicts a shelf filled with various celebrity
            and entertainment magazines, each featuring bold and colorful
            headlines and images of people, suggesting a focus on gossip and
            entertainment news.",
}
```

**Example 2: Negative Sample**

```
{
    "Scene": {
        "Description": "Magazine cover featuring a person with text and
            additional images."
    },
    "Objects": {
        "Living Beings": [
            {
                "Classification": "Human",
```

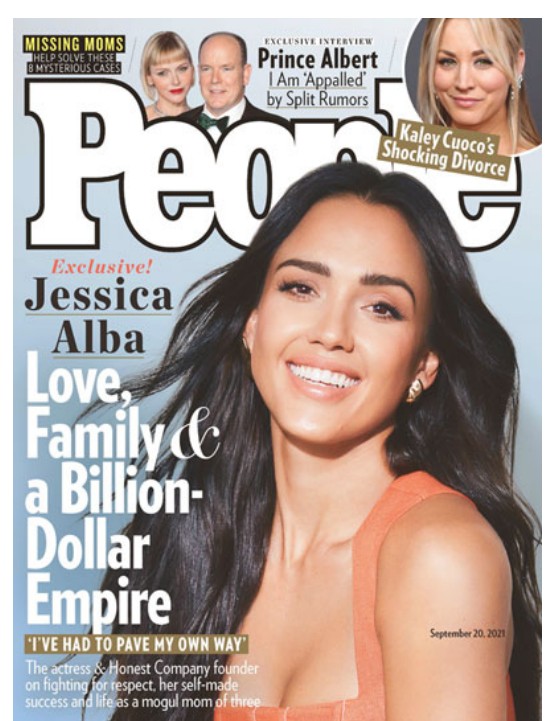

Figure A.2: **Negative Image Example.**

```
            "Characteristics": {
                "Hair Color": "Dark",
                "Posture": "Smiling"
            }
        }
    ],
    "Inanimate Objects": [
        {
            "Category": "Text",
            "Properties": {
                "Color": "White, Black, Yellow",
                "Size": "Various",
                "Style": "Bold, Italic"
            }
        },
        {
            "Category": "Magazine Title",
            "Properties": {
                "Text": "People",
                "Color": "White",
                "Size": "Large"
            }
        }
    ]
},
"Activities": [],
"Contextual Elements": {
    "Time of Day": "N/A",
    "Weather": "N/A"
},
"Visual Patterns": {
    "Dominant Colors": [
        "White",
        "Black",
```

```
            "Yellow",
            "Orange"
        ],
        "Textures": [
            "Smooth"
        ]
    },
    "Emotional Undertones": "Positive, Happy",
    "Textual Information": "Exclusive! Jessica Alba Love, Family & a
        Billion-Dollar Empire. September 20, 2021. I'VE HAD TO PAVE MY
        OWN WAY.",
    "Summary": "The image is a magazine cover featuring a smiling person
        with text about Jessica Alba and other headlines.",
}
```

**Example 2: Query Image**

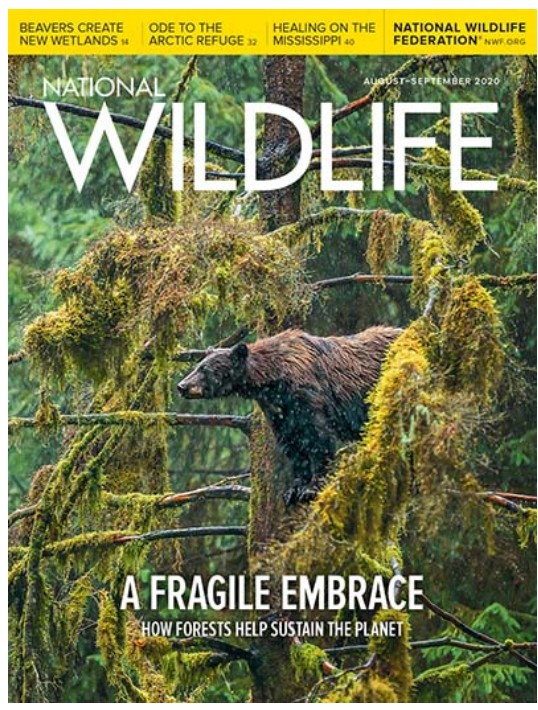

Figure A.3: **Query Image Example.**

```
{
    "Scene": {
        "Description": "Natural forest environment with dense foliage."
    },
    "Objects": {
        "Living Beings": [
            {
                "Species": "Bear",
                "Characteristics": {
                    "Color": "Brown",
                    "Posture": "Climbing or resting on a tree branch"
                }
            }
        ],
        "Inanimate Objects": [
            {
                "Category": "Tree",
```

```
                    "Properties": {
                        "Color": "Green and brown",
                        "Size": "Large",
                        "Material": "Wood",
                        "Shape": "Tall with branches covered in moss"
                    }
                }
            ]
        },
        "Activities": [
            {
                "Description": "Bear climbing or resting on a tree branch",
                "Participants": [
                    "Bear"
                ]
            }
        ],
        "Contextual Elements": {
            "Time of Day": "Not specified",
            "Weather": "Likely damp or rainy, suggested by mossy tree
                branches"
        },
        "Visual Patterns": {
            "Dominant Colors": [
                "Green",
                "Brown"
            ],
            "Textures": [
                "Mossy",
                "Furry"
            ]
        },
        "Emotional Undertones": "Calm, natural setting",
        "Textual Information": {
            "Main Title": "NATIONAL WILDLIFE",
            "Subtitles": [
                "A FRAGILE EMBRACE",
                "HOW FORESTS HELP SUSTAIN THE PLANET"
            ],
            "Additional Text": [
                "BEAVERS CREATE NEW WETLANDS 14",
                "ODE TO THE ARCTIC REFUGE 32",
                "HEALING ON THE MISSISSIPPI 40",
                "NATIONAL WILDLIFE FEDERATION",
                "AUGUST-SEPTEMBER 2020"
            ]
        },
        "Summary": "The image is a cover of a magazine titled 'NATIONAL
            WILDLIFE', featuring a bear in a lush, moss-covered forest. The
            scene conveys a sense of tranquility and highlights the theme of
            forest conservation.",
}
```

These examples demonstrate:

1. **Task-agnosticism:** Descriptions contain no reference to "rule discovery" or the Bongard task structure

2. **Completeness:** Hierarchical structure captures scene, objects, activities, relationships, and patterns

3. **Reusability:** Same schema applied to all three benchmarks (Bongard-OW, Bongard-HOI, Winoground)

## A.7 PROMPTS FOR WINOGROUND COMPONENTIAL ANALYSIS (STAGE 2 REASONING)

For the Winoground benchmark (Thrush et al., 2022), Stage 2 of our Componential Analysis (CA) paradigm requires a reasoning model to evaluate matches between image descriptions (JSON strings generated in CA Stage 1 from the Winoground images) and the provided captions. The following prompts were used to guide the reasoning LLM in selecting the best match, forming the basis for calculating the Text Score and Image Score components as detailed in Appendix A.5.4. Both prompts instruct the model to perform a systematic, step-by-step comparison and to return its analysis and final categorization in a structured JSON format.

### A.7.1 PROMPT FOR TEXT SCORE COMPONENT DECISION

The following Python function defines the prompt presented to the reasoning LLM. Given one image's detailed JSON description and two candidate captions (Caption 0 and Caption 1), the model is tasked to determine which caption has a higher possibility of matching the image description. This process is repeated for the second image description in the Winoground pair to gather the necessary data points for the Text Score.

```python
def text_score_prompt(image_description, caption_0, caption_1):
    '''
    Generates a prompt for an LLM to determine if an image description
        has a higher possibility
    of matching caption_0 or caption_1 by evaluating each match
        individually and comparing them,
    using a detailed JSON description and commonsense reasoning.

    Args:
        image_description (str): A JSON string description of an image.
        caption_0 (str): The first candidate caption.
        caption_1 (str): The second candidate caption.

    Returns:
        str: The formatted prompt string.
    '''
    prompt = f"""You are provided with a detailed JSON description of a
        single image and two different captions (Caption 0 and Caption
        1). Your task is to evaluate how well the image description
        matches *each* caption individually, determine which caption
        provides a stronger match (higher possibility), and explain why.
        Apply commonsense reasoning where needed.

    **Image Description (JSON):**
    ```json
    {image_description}```

    **Caption 0:** "{caption_0}"
    **Caption 1:** "{caption_1}"

    **Instructions:**
    1.  **Deconstruct Image Description:** Identify the main entities
        (using `id`s), actions (`Activities`), attributes
        (`characteristics`, `properties`), and relationships (`Spatial
        Relationships`) detailed in the JSON description. Use
        commonsense to understand the full context implied by the
        description.
    2.  **Evaluate Match with Caption 0:** Systematically check how well
        the key elements identified in Caption 0 (entities, actions,
        attributes, relationships) are supported by the details in the
        `Image Description` JSON.
        * Look for specific `id`s, `characteristics`, `actor_ids`,
            `target_ids`, `action` descriptions, `relationship` types,
            etc., in the JSON that align with Caption 0's elements.
```

```
                * Use commonsense reasoning to map JSON details to caption terms
                    (e.g., `characteristics` like "elderly" might correspond to
                    "old person").
                * Assess the overall strength of the match (e.g., "strong
                    support", "partial support", "weak support",
                    "contradiction"). Note any discrepancies.
        3.  **Evaluate Match with Caption 1:** Perform the same systematic
            check and assessment against Caption 1.
            * Look for specific JSON details supporting or contradicting
                Caption 1's elements.
            * Use commonsense reasoning.
            * Assess the overall strength of the match for Caption 1. Note
                any discrepancies.
        4.  **Compare Matches and Conclude:** Compare the strength of the
            match assessed for Caption 0 versus Caption 1. Explain *why* the
            image description represents one caption with a higher
            possibility or accuracy than the other. Highlight the specific
            JSON details (or lack thereof) that lead to this conclusion.
            Explicitly mention where commonsense was applied during the
            evaluation or comparison.
        5.  **Categorize:** Assign 'cat_0' if the image description has a
            higher possibility of matching Caption 0, or 'cat_1' if it has a
            higher possibility of matching Caption 1.

    Return your response strictly in the following JSON format:
    {{
        "analysis": (Your detailed analysis comparing the match strength
            for each caption against the image description, explaining
            why one is a better fit, and noting the use of commonsense),
        "category": ('cat_0' or 'cat_1')
    }}

    Do not include any text outside of the JSON structure. Your decision
        must be based on evaluating the match between the image
        description and each caption, then comparing those evaluations.
    """
    return prompt
```

### A.7.2 Prompt for Image Score Component Decision

Similarly, the following Python function defines the prompt used for the Image Score component. Given one caption and two candidate image descriptions (Image 0 Description and Image 1 Description, both JSON strings), the model is tasked to determine which image description has a higher possibility of matching the caption. This is repeated for the second caption in the Winoground pair.

```
def image_score_prompt(caption, image_0_description,
    image_1_description):
    '''
    Generates a prompt for an LLM to determine if a caption has a higher
        possibility
    of matching image_0_description or image_1_description by evaluating
        each match
    individually and comparing them, using detailed JSON descriptions
        and commonsense reasoning.

    Args:
        caption (str): The caption to evaluate.
        image_0_description (str): The JSON string description of the
            first image.
        image_1_description (str): The JSON string description of the
            second image.

    Returns:
        str: The formatted prompt string.
```

```
1512        '''
1513        prompt = f"""You are provided with a single caption and detailed
1514            JSON descriptions of two different images (Image 0 and Image 1).
1515            Your task is to evaluate how well the caption matches *each*
1516            image description individually, determine which description
1517            provides a stronger match (higher possibility), and explain why.
1518            Apply commonsense reasoning where needed.
1519
1520        **Caption**: "{caption}"
1521
1522        **Image 0 Description (JSON):**
1523        ```json
1524        {image_0_description}```
1525
1526        **Image 1 Description (JSON):**
1527        ```json
1528        {image_1_description}```
1529
1530        **Instructions:**
1531        1.  **Deconstruct Caption:** Identify the main entities, actions,
1532            attributes, and relationships mentioned in the caption (e.g.,
1533            "old person", "kisses", "young person"). Use commonsense to
1534            understand the full context implied by the caption.
1535        2.  **Evaluate Match with Image 0:** Systematically check how well
1536            the key elements identified in the caption are supported by the
1537            details in 'Image 0 Description'.
1538            * Look for specific 'id's, 'characteristics', 'actor_ids',
1539                'target_ids', 'action' descriptions, 'relationship' types,
1540                etc., in the JSON that align with the caption's elements.
1541            * Use commonsense reasoning to map caption terms to JSON details
1542                (e.g., "old person" might correspond to 'characteristics'
1543                like "elderly").
1544            * Assess the overall strength of the match (e.g., "strong
1545                support", "partial support", "weak support",
1546                "contradiction"). Note any discrepancies.
1547        3.  **Evaluate Match with Image 1:** Perform the same systematic
1548            check and assessment against 'Image 1 Description'.
1549            * Look for specific JSON details supporting or contradicting the
1550                caption's elements.
1551            * Use commonsense reasoning.
1552            * Assess the overall strength of the match for Image 1. Note any
1553                discrepancies.
1554        4.  **Compare Matches and Conclude:** Compare the strength of the
1555            match assessed for Image 0 versus Image 1. Explain *why* one
1556            description represents the caption with a higher possibility or
1557            accuracy than the other. Highlight the specific JSON details (or
1558            lack thereof) from *both* descriptions that lead to this
1559            conclusion. Explicitly mention where commonsense was applied
1560            during the evaluation or comparison.
1561        5.  **Categorize:** Assign 'cat_0' if the caption has a higher
1562            possibility of matching Image 0 Description, or 'cat_1' if it
1563            has a higher possibility of matching Image 1 Description.
1564
1565        Return your response strictly in the following JSON format:
            {{
                "analysis": (Your detailed analysis comparing the match strength
                    for each description against the caption, explaining why one
                    is a better fit, and noting the use of commonsense),
                "category": ('cat_0' or 'cat_1')
            }}

            Do not include any text outside of the JSON structure. Your decision
                must be based on evaluating the match between the caption and
                each description, then comparing those evaluations.
            """
```

```
return prompt
```

## A.8 FRAMEWORK ANALYSIS: PERCEPTION-REASONING SEPARATION AS DESIGN ASSUMPTION

### A.8.1 EVIDENCE SUPPORTING PERCEPTION-REASONING SEPARATION

We demonstrate that assuming perception-reasoning separation enables effective framework design, validated by SOTA results and model generalization. Three key findings support meaningful separation:

1. **Monotonic Paradigm Progression:** Across benchmarks, performance consistently improves with structured decomposition: Bongard-OW DVRL 80.0% → DRL 88.0% → CA 92.8% (+12.8 points); Bongard-HOI DVRL 71.2% → CA 77.3% (+6.1); Winoground DVRL 68.2% → CA 75.5% (+7.3).

2. **Text-Only LLM Success:** Symbolic reasoners (Phi-4, Qwen2.5) achieve 91-93% using *only* descriptions of images never seen during training, proving perception and reasoning can be separated.

3. **Cross-Benchmark Generalization:** Same schema achieves SOTA on three fundamentally different tasks (Bongard-OW, Bongard-HOI, Winoground).

### A.8.2 CLARIFYING TASK-AGNOSTICISM VS. DOMAIN-GENERAL SCHEMAS

We distinguish two concepts:

- **Task-Agnosticism (our claim):** CA Stage 1 receives images + generic schema *without task structure* (rule domains, caption labels, query designation).
- **Domain-General Schema (acknowledged constraint):** Hierarchical JSON (Scene, Objects, Activities) biases toward natural images but does *not* encode which attributes matter for specific rules.

For Winoground example "old person kisses young person":

- Task-conditioned CoT sees caption → targets "old," "young," "kisses";
- Our CA Stage 1 sees image → generates generic description;
- Stage 2 reasoning discovers which attributes matter. Stage 1 provides a "canvas" for reasoning, not task-encoded information. Consistent text-only LLM success confirms descriptions are not implicitly task-conditioned.

## A.9 RULE SYNTHESIS QUALITY: PRAGMATIC ASSESSMENT

### A.9.1 APPROACH: SEMANTIC SIMILARITY + ERROR ANALYSIS

Rather than extensive human annotation (infeasible scope), we assess rule quality via two complementary approaches:

**Semantic Similarity Analysis**  Table A.2 shows cosine similarity between extracted rules (DRL) and query descriptions.

**Error Analysis (Table A.9)**  Manual categorization of 15 misclassified samples reveals:

- 8 cases (53%): Rule extraction error (over-generalization, missing attributes)
- 3 cases (20%): Reasoning application error
- 4 cases (27%): Perceptual/other errors

Rule extraction is dominant failure mode, validating that high-quality descriptions (CA Stage 1) determine success.

**Text-Only LLM Success** Text-only reasoners (Phi-4, Qwen2.5) achieve 91-93% using only descriptions, proving extracted rules contain sufficient information for effective application—even by symbolic reasoners unfamiliar with visual grounding.

### A.9.2 WHY HUMAN ANNOTATION WAS NOT PURSUED

1. **No Ground-Truth Rules:** Bongard-OW provides binary labels, not explicit rules. Annotating 500+ ground-truth rules requires expert analysis, inter-rater assessment—substantial scope beyond paper.

2. **Circularity Risk:** Both extracted rules and hypothetical human-annotated rules derive from LLM interpretation. Without independent symbolic ground-truth, LLM vs. LLM comparison lacks objectivity.

3. **Semantic Similarity is Sufficient Proxy:** High-accuracy samples show strong rule-description alignment (0.90+); misclassified samples show low alignment (0.64). This correlation validates semantic similarity as quality proxy.

Our pragmatic assessment (semantic similarity + error analysis + text-only LLM success) provides sufficient evidence that rule synthesis quality is high for correctly classified samples and that rule extraction remains the primary differentiator.

## A.10 RESULTS AND EXTENDED ANALYSIS

### A.10.1 PERFORMANCE ON BONGARD OPENWORLD

| | Gemini 2.0 | | GPT-4o | |
|---|---|---|---|---|
| Category | Mean | Std Dev | Mean | Std Dev |
| Positive | 0.915 | 0.02 | 0.902 | 0.02 |
| Negative | 0.868 | 0.02 | 0.866 | 0.02 |

Table A.2: Semantic Similarity (Cosine) between query descriptions and rules derived during Deductive Rule Learning.

### A.10.2 PERFORMANCE ON BONGARD-HOI

(Refer to Table A.3 in main text)

| Model | Paradigm | sosa | soua | uosa | uoua | Avg |
|---|---|---|---|---|---|---|
| | DVRL | 50 | 54 | 49 | 50 | 50.8 |
| Gemini 2.0 | DRL | 63 | 62 | 55 | 65 | 61.3 |
| | CA | 77 | 74 | 70 | 77 | 74.5 |
| | DVRL | 68 | 75 | 61 | 70 | 68.5 |
| GPT-4o | DRL | 73 | 77 | 64 | 73 | 71.8 |
| | CA | 83 | 83 | 66 | 77 | 77.3 |
| Human Avg. | – | 87.2 | 90.0 | 93.6 | 94.9 | 91.4 |

Table A.3: Performance (%) on Bongard-HOI splits across paradigms. Human average taken from (Jiang et al., 2022) **Splits:** sosa: seen_obj_seen_act, soua: seen_obj_unseen_act, uosa: unseen_obj_seen_act, uoua: unseen_obj_unseen_act. Human average from cited source.

| Model / Strategy | Text | Image | Group |
|---|---|---|---|
| Gemini (Baseline) | 30.75 | 26.00 | 25.00 |
| Gemini + DDCoT | 45.00 | 25.00 | 23.75 |
| Gemini + CCoT | 22.50 | 33.00 | 20.75 |
| Gemini + CoCoT | 40.00 | 32.50 | 27.75 |
| **Gemini 2.0 + CA (Ours)** | **71.91** | **48.71** | **42.01** |

Table A.4: Performance comparison on Winoground (400 samples). CA refers to our Componential Analysis paradigm. Other results use Gemini Pro Vision with different prompting strategies.

### A.10.3 WINOGROUND PERFORMANCE CONTEXT

To contextualize the performance of our Componential Analysis (CA) paradigm applied to Gemini 2.0 on Winoground (reported in Section 6.2), we also ran evaluations using Gemini Pro Vision with several prompting strategies. Table A.4 shows these comparative results on the 400-sample Winoground set used. While advanced CoT methods like DDCoT and CoCoT improve over the baseline for Gemini Pro Vision, the CA paradigm applied to Gemini 2.0 achieves competitive scores, particularly on the text metric, demonstrating its effectiveness.

### A.10.4 COMPARISON OF DESCRIPTION SOURCES (PIXTRAL-12B VS. GPT-4O)

The results, detailed in Table A.5, consistently show that using image components described by GPT-4o yielded higher downstream reasoning accuracy compared to using components described by Pixtral-12B across all tested reasoning models. While both description sources enabled strong performance, the advantage conferred by GPT-4o's descriptions (ranging from approximately 2% to over 11% improvement depending on the reasoning model) further underscores the critical dependence of reasoning outcomes on the fidelity, richness, and potentially the alignment of the initial perceptual descriptions with the concepts required by the reasoning task. This reinforces the significance of the VLM's front-end visual processing and description capabilities as a key factor influencing overall visual reasoning performance.

| | Components (%) | |
|---|---|---|
| **Model** | **Pixtral-12B** | **GPT-4o** |
| Deepseek-R1-14B | 83.21 | 87.98 |
| Llama3.2-vision-90B | 89.05 | 90.98 |
| Phi-4-14B | 86.86 | 91.98 |
| Qwen2.5-14B | 90.51 | 92.99 |
| LLaVA-7B | 68.61 | 80.56 |
| Llama3.2-vision-11B | 80.29 | 84.17 |
| LLaVA-34B | 79.56 | 81.56 |
| Phi-3-14B | 84.67 | 86.97 |

Table A.5: Performance comparison using Componential Analysis (Stage 2) with image descriptions generated by either Pixtral-12B or GPT-4o. Evaluated across various reasoning models.

### A.10.5 COMPONENTIAL ANALYSIS RESULTS BY COMMONSENSE CATEGORY

Analysis of GPT-4o and Gemini 2.0 performance in CA across commonsense categories (Appendix Table A.6) showed generally strong performance, indicating robustness to varied conceptual rules. Minor variations suggested potential differences in handling specific types of context or attributes, possibly reflecting training data nuances.

| ID | Concept Category | GPT-4o (%) | Gemini 2.0 (%) |
|----|------------------|------------|----------------|
| 0 | Anything else | 92.88 | 94.23 |
| 1 | Human-Object Interaction (HOI) | 86.67 | 92.86 |
| 2 | Taste / Nutrition / Food | 100.00 | 85.71 |
| 3 | Color / Material / Shape | 88.89 | 91.67 |
| 4 | Functionality / Status / Affordance | 100.00 | 100.00 |
| 5 | And / Or / Not | 90.00 | 80.00 |
| 6 | Factual Knowledge | 90.00 | 90.00 |
| 7 | Meta Class | 100.00 | 100.00 |
| 8 | Relationship | 100.00 | 100.00 |
| 9 | Unusual Observations | 92.31 | 92.31 |

Table A.6: Overall accuracy (%) of GPT-4o and Gemini 2.0 on the Bongard-OW test set using Componential Analysis, broken down by Commonsense ID category (visual concept). Performance variations highlight differing model strengths on specific concept types.

### A.10.6 IMPACT OF CoT-LIKE STRUCTURE

(Refer to Table A.7 below)

| Prompt Type | Accuracy (%) | | |
|-------------|--------------|------|------|
| | Overall | neg | pos |
| Minimal (No CoT) | 61.6 | 39.2 | 84.0 |
| Structured (CoT-like) | 80.0 | 66.4 | 93.6 |

Table A.7: Impact of Structured Prompting on DVRL accuracy (GPT-4o).

### A.10.7 DETAILED ERROR ANALYSIS EXAMPLES

(Refer to Table A.8 below)

| No. | Test ID | Caption (Rule) | Reason for Error (Based on GPT-4o o/p) |
|-----|---------|----------------|----------------------------------------|
| 1 | 0021_neg_0 | Cars on the city streets at night | Weak reasoning (similarity): Rule requires vehicles, test image (painting) lacks them explicitly, though context implies city. |
| 2 | 0014_neg_0 | A person playing a guitar. | Rule extraction error: Rule too general (e.g., "person with instrument"), misses specific object (guitar) mentioned in analysis. |
| 3 | 0033_neg_0 | A bicycle is placed in the corner | Rule extraction error: Misses key property (in a corner / specific placement context). Test image (collage) lacks this context. |
| 4 | 0037_neg_0 | The girl has long and thin braids on her head. | Rule extraction error: Rule too general (e.g., "girl with braids"), misses specific property (long and thin). |
| 5 | 0076_pos_0 | Various kinds of rings | Rule extraction error: Rule misses specific object (ring), focuses on property (intricate design) absent in query. |
| 6 | 0076_neg_0 | Various kinds of rings | Rule extraction error: Rule misses specific object (ring), too general. |
| | | | Continued on next page |

| No. | Test ID | Caption (Rule) | Reason for Error (Based on GPT-4o Output) |
|---|---|---|---|
| 7 | 0082_neg_0 | Live coral on the sea floor. | Weak reasoning (similarity): Rule identifies 'coral', but test image description fails to mention it. Perceptual description error. |
| 8 | 0084_neg_0 | A wooden fence surrounding a grassy field. | Rule extraction error: Rule misses specific object (grass), uses broader term (greenery). Test image has greenery but not clearly grass. |
| 9 | 0112_neg_0 | A wooden floor in the living room. | Rule extraction error: Misses key objects (living room, floor), focuses only on 'wooden' and general 'indoor'. |
| 10 | 0117_neg_0 | Colorful ribbons. | Rule extraction error: Rule too general, misses specific object (ribbons). |
| 11 | 0122_neg_0 | A satellite view of Earth. | Rule extraction error: Misses specific viewpoint (top-down satellite), uses more general 'aerial'. |
| 12 | 0136_pos_0 | Spectator seats view in the stadium. | Weak reasoning/Rule Application error: Rule mentions "sports or spectators", query image description lacks both, leading to incorrect negative classification despite being stadium seats. |
| 13 | 0213_neg_0 | Checkerboard pattern fabrics | Rule extraction error: Misses specific object context (fabric), although pattern is identified. |
| 14 | 0234_neg_0 | A beautiful stone sculpture | Rule extraction error: Focuses on wrong property ('prominent' obelisk) instead of the intended rule property ('tall' obelisk). |
| 15 | 0247_pos_0 | Small river filled with reeds | Rule extraction error: Misses key object (reeds), while focusing on negative constraints (no industrial presence) which are weakly present. |

Table A.8: Error Analysis: Examples of Bongard-OW cases misclassified by both GPT-4o and Gemini 2.0 in Componential Analysis. Captions indicate the ground truth rule (Wu et al., 2024). Reasoning based on analyzing GPT-4o's generated analysis, rule, and query description.

### A.10.8  IMPACT OF CONTEXT IMAGE PAIRS ON HOLISTIC PROCESSING

To investigate whether perceptual quality deteriorates with the number of input images in holistic processing, we conducted an additional ablation study on the Direct Visual Rule Learning (DVRL) paradigm.

#### A.10.8.1  MOTIVATION AND SETUP

The DVRL paradigm requires models to process all context images simultaneously (6 positive + 6 negative = 12 images) along with a query image, demanding high cognitive load. This raises the question: how sensitive is DVRL performance to the number of context image pairs provided? Understanding this relationship has implications for:

- The cognitive demands of multi-image processing (holistic vs. sequential reasoning)
- The feasibility of DVRL for resource-constrained scenarios
- The theoretical basis for moving toward compositional paradigms (DRL, CA) as we break down the task

#### A.10.8.2  EXPERIMENTAL DESIGN

We conducted DVRL evaluation on GPT-4o using 200 balanced samples from Bongard-OW, testing three context image pair configurations:

- **2/2:** 2 positive + 2 negative examples + 1 query image (5 images total)
- **4/4:** 4 positive + 4 negative examples + 1 query image (9 images total)

- **6/6:** 6 positive + 6 negative examples + 1 query image (13 images total, standard)

All evaluations used identical prompts (Appendix A.6.1) and temperature settings (T=0).

### A.10.8.3 RESULTS

Results are shown in Table A.9.

| Number of Context Image Pairs | DVRL Accuracy (%) |
|---|---|
| 2/2 (5 total images) | 81.0 |
| 4/4 (9 total images) | 87.5 |
| 6/6 (13 total images, standard) | 93.0 |

Table A.9: Impact of context image pair count on DVRL accuracy. Results show consistent performance degradation as fewer examples are provided to GPT-4o during holistic, simultaneous multi-image processing.

### A.10.8.4 ANALYSIS AND INTERPRETATION

Our findings reveal a clear pattern: **accuracy degrades consistently as the number of context images decreases**, dropping from 93.0% (full 6/6 set) to 87.5% (4/4) to 81.0% (2/2)—a total 12-point spread. This demonstrates that:

1. **Cognitive Load is Non-Trivial:** The holistic processing demanded by DVRL shows sensitivity to context abundance. Models achieve substantially higher accuracy when provided richer exemplar sets, suggesting the simultaneous integration of 12 context images provides meaningful information density compared to sparser inputs.

2. **Justification for Paradigm Progression:** The performance degradation with fewer images provides empirical support for the progression from DVRL (holistic, information-dense) through DRL (separating extraction/application) to CA (sequential, modular processing). As we reduce simultaneous cognitive load through structured decomposition, we can maintain or improve accuracy despite potentially lower total information availability.

3. **Implication for Few-Shot Learning:** While our standard experiments use 6/6 examples, this ablation shows that models can still achieve reasonable (though reduced) performance with fewer examples. This has practical implications for few-shot scenarios or resource-constrained settings where providing fewer images might be necessary.

4. **Alignment with Human Cognition:** The performance decrease with fewer examples mirrors human visual reasoning, where more diverse exemplars typically enable more robust rule generalization. This alignment with cognitive principles further validates the cognitive grounding of our DVRL paradigm.

### A.10.8.5 CONNECTION TO MAIN RESULTS

This ablation contextualizes our primary DVRL results (Table 2). The 80-82% accuracy for GPT-4o and Gemini 2.0 in standard DVRL (6/6 pairs) represents performance at the *upper end* of what holistic multi-image processing can achieve. The consistent 2-4 point improvement when moving to DRL and then 5-8 point improvement to CA suggests that structured decomposition not only accommodates sparse information but leverages it more effectively than simultaneous holistic processing.

## A.11 COMPUTATIONAL COST ANALYSIS

While our framework prioritizes diagnostic interpretability over production efficiency, we provide cost estimates to contextualize the performance-efficiency trade-off.

### A.11.1 API CALL COUNTS PER PARADIGM

In Table A.10 we discuss computational cost for proprietary models on Bongard-OW samples.

| Paradigm | API Calls/Sample | Est. Time (s) | Est. Cost (USD) |
|----------|------------------|---------------|-----------------|
| DVRL | 1 (13 images) | 8–12 | $0.025 |
| DRL | 2 (12 + 1 images) | 12–18 | $0.040 |
| CA | 14 (13 desc. + 1 reason) | 50–70 | $0.080 |
| ICA | 15–16 (CA + feedback) | 80–100 | $0.120 |

Table A.10: Computational cost estimates per Bongard-OW sample for proprietary models (GPT-4o). Time includes API latency; cost based on OpenAI pricing ($0.01/1K input tokens, $0.03/1K output tokens, May 2024). Open-source models have similar call patterns but no API cost.

### A.11.2 ACCURACY VS. LATENCY TRADE-OFF

CA achieves 92.8% accuracy with $3\times$ the latency of DVRL (80.0%). ICA further improves to 95.25% (Winoground Group Score: 55.25%) at $4$–$5\times$ DVRL latency. For research diagnostics, this trade-off is acceptable; for production deployment, practitioners may prioritize static CA over interactive ICA based on application requirements.

### A.11.3 SCALABILITY CONSIDERATIONS

Full Bongard-OW evaluation (500 samples):

- **CA**: $500 \times \$0.08 = \$40$ (2.8–4.2 GPU hours)
- **ICA**: $500 \times \$0.12 = \$60$ (4.9–6.9 GPU hours)

These costs are modest for research but would scale linearly for large-scale deployment (e.g., 100K images: $8K–$12K for CA/ICA).

## A.12 LIMITATIONS OF THE CA PARADIGM

### A.12.1 DOMAIN-GENERAL SCHEMA CONSTRAINTS

CA relies on hierarchical schema (Scene, Objects, Activities) effective for **natural images** (objects, agents, actions) but requiring adaptation for other domains:

- **Formal Symbolic Bongard:** Require schemas like "ShapeType," "Topology," not "Objects"
- **Function Graphs:** Require "AxisLabels," "Trends," not "Activities"
- **Mathematical Diagrams:** Require "ProofSteps," not natural-image categories

**Refined Generalizability:** Our framework is task-agnostic but domain-constrained to natural-image benchmarks. We demonstrate cross-task generalization *within this domain* but do not claim universal applicability (see main text Limitations, Section 8).

### A.12.2 VERBALIZABLE REASONING ASSUMPTION

CA effectiveness assumes critical visual properties are expressible in language. Tasks requiring non-verbalizable reasoning (e.g., BLINK depth correspondence) may show reduced effectiveness. Discussed in main text Section 8.

### A.12.3 DECOMPOSABILITY ASSUMPTION

Our framework assumes that perception and reasoning can be meaningfully separated for visual reasoning tasks. This assumption holds for natural-image rule-based reasoning (Bongard-OW, Bongard-HOI, Winoground) but may not hold universally. Tasks requiring non-verbalizable visual reasoning (geometric correspondence, pure spatial transformation) or tasks where visual and semantic information are inherently intertwined would violate this assumption. We do not claim decomposability is fundamental but rather that assuming it is productive for analyzing certain classes of visual reasoning tasks.

### A.12.4 ROBUSTNESS EXPERIMENT

We evaluated both DVRL (holistic, one-pass) and CA (componential, two-stage) paradigms on noisy image variants:

- **Noise Type:** Gaussian noise (additive, zero-mean)
- **Noise Intensity:** 0.08 (8% pixel-level perturbation)
- **Test Set:** Class-balanced 100-sample subset from Bongard-OW
- **Clean Baseline:** DVRL 80.0%, CA 92.8%

| Paradigm | Clean Images | Noisy Images | Degradation | Robustness |
|---|---|---|---|---|
| DVRL (holistic) | 80.0% | 49.0% | -31.0 pp | Low |
| CA (componential) | 92.8% | 90.91% | -1.89 pp | High |

Table A.11: Robustness comparison: DVRL and CA paradigms evaluated on Gaussian-noise-corrupted Bongard-OW images (intensity 0.08). DVRL's holistic processing severely degrades (-31 points), while CA's two-stage architecture maintains near-baseline performance (-1.89 points). This demonstrates that decomposition provides inherent robustness via semantic abstraction.

## A.13 INTERACTIVE CA

### A.13.1 BIDIRECTIONAL PERCEPTION-REASONING DIALOGUE

ICA demonstrates that **interactive feedback** (reasoning $\leftrightarrow$ perception) improves performance beyond static decoupling. Largest gains on perception-sensitive metrics (Winoground Image Score +4.25 for GPT-4o, +6.5 for Gemini 2.0) validate that reasoning-directed re-perception resolves specific visual ambiguities.

**Key Insight:** Perception bottleneck is not immutable but often reflects insufficient focus. When reasoning identifies ambiguities and directs perception to "look again," accuracy improves. This models human verification processes and establishes foundation for systems where reasoning actively shapes perception.

### A.13.2 CURRENT SCOPE

**Current Evaluation:** Winoground only (GPT-4o, Gemini 2.0, single-round feedback).

**Future Work:**

1. Extend to Bongard-OW/HOI (test if multi-turn feedback improves abstract rule discovery)
2. Multi-round analysis (2-3 feedback rounds, diminishing returns analysis)
3. Model expansion (Claude 3.5 Sonnet, Gemini 3 Pro)
4. Question generation strategies (systematic analysis of question types)

