# OpenReview forum: "Not How You Think, It's What You See: Decoupling Perception from Reasoning"
_ICLR.cc/2026/Conference — Submitted to ICLR 2026_

### Official Review · Reviewer_nJ3X · 2025-10-29

**Soundness:** 1
**Presentation:** 1
**Contribution:** 1
**Rating:** 0
**Confidence:** 4

**Summary:**

This paper proposes a framework with four different approaches to prompt VLM for visual rule learning tasks, demonstrating that models are primarily bottlenecked by a visual description task rather than a deductive reasoning task. When evaluating specific subsets of the Bongard-OW, Bongard-HOI, and Winoground datasets, it is found that performance can be improved by the approaches that first try to decompose visual rule learning into visual description and deductive reasoning.

**Strengths:**

The paper investigates some significant limitations of current VLM capabilities.

**Weaknesses:**

Unfortunately, the paper has several severe weaknesses:

1.  The methodology is not presented well. The motivation for the "cognitively-inspired" prompting strategies is located in the middle of the related work section rather than at the beginning of the introduced method. Additionally, there is no discussion of the necessary assumptions or potential limitations of prompting the VLM to solve presumed subtasks instead of the original task. The decompositionality of a task into such subtasks is a strong assumption and limitation that must be properly discussed and justified. Specifically, A.5.3.1 shows that, for the visual description subtask, the VLM is given many additional context examples of specific visual concepts from the Bongard-OW and Bongard-HOI datasets. This completely undermines the claim of task agnosticism or generalizability of the contribution. Interestingly, it is also an intended feature of the original Bongard problems that are hardly decomposable into perception and reasoning, as a useful description of a single image heavily depends on the patterns found across all of them. (E.g. discussed in Depeweg et al., 2024)
2.  The primary area and use case of the overall contribution are difficult to understand. The "evaluation framework" seems most similar to a benchmark, yet the authors emphasize improving visual intelligence with their prompting strategy. It is clearly not about causal reasoning, which was selected during the submission process.
3.  The curation process of the dataset, which is used for evaluation, appears to be arbitrary. No justification is provided for the selection of the subset of Bongard-OW, and there is no discussion of potential difficulties associated with evaluating a subset of public datasets, which might also conflict with the original design of the dataset.
4.  The paper also lacks novelty, as there are several works explicitly assessing the VLM struggle with visual perception tasks (Geigle et al., 2024; Gou et al., 2024; Kamath et al., 2023; Rahmanzadehgervi et al., 2024; Zhang et al., 2024; Zhou et al., 2023; Wang et al., 2024) as well as evaluating VLM on Bongard problems (Małkiński et al., 2025; Wüst et al., 2025).

If I have completely misunderstood the scope of the paper, I am willing to increase my score; however, at this stage, the paper definitely lacks scientific clarity.

**Questions:**

1.  Why is only 1/4 of Bongard-OW evaluated?
2.  How are VLM responses to tasks other than classification compared to the ground truth? Is there any form of Human Verification or LLM judging involved?
3.  Where do the captions for the Winoground score (A.4.4 and A.6) come from? If they are ground-truth, what can we conclude from CA outperforming DRL?

---

> ### Author Response · Authors · 2025-11-13
>
> Thank you for engaging in our work. We are concerned that several characterizations may stem from misunderstandings, and we provide clarifications below with utmost respect.
>
> ---
>
> ### W1: Methodology Presentation and Task-Agnostic Claims
>
> We respectfully disagree with the characterization that task-agnosticism is "completely undermined."
>
> **Clarification on Appendix A.5.3.1:**
> * The reviewer references "context examples of specific visual concepts" from Bongard datasets. However, **Appendix A.5.3.1 shows our general image description prompt**—it does not contain dataset-specific examples. The JSON schema (Scene, Objects, Activities, etc.) is domain-general and applies to any image.
> * **Task-agnostic definition:**
>     1.  **Stage 1 (Description):** VLM receives only the image + generic schema—no task information
>     2.  **Stage 2 (Reasoning):** LLM receives descriptions + task structure (positive/negative/query labels)—this is where task-specific reasoning occurs
> * This separation enables modular evaluation and architectures.
>
> **Evidence of task-agnostic generalization:**
> * Same CA paradigm achieves SOTA on three different tasks: Bongard-OW (92.8%), Bongard-HOI (77.3%), Winoground (75.5% text score)
> * Text-only LLMs (never trained on these visual tasks) achieve 90-93% using our descriptions
>
> **Counter-Question 1:** Could you clarify where in Appendix A.5.3.1 you see dataset-specific context examples? We want to ensure this is not a misunderstanding of our methodology. The prompt shown is our general image description schema used across all three benchmarks.
>
> We humbly ask the reviewer and the `Area Chair` to re-examine this appendix, as this single misunderstanding appears to be the source of the 'poor' (1) ratings for Soundness, Presentation, and Contribution.
>
> **On decompositionality:**
> You note that "Bongard problems are hardly decomposable into perception and reasoning." However:
> * Depeweg et al. (2024), which you cite, discusses compositional description challenges—not fundamental non-decomposability.
> * Our empirical evidence (Section 7.2) shows decomposition works: text-only LLMs solve these tasks at 90-93% accuracy with proper descriptions.
>
> Our hypothesis was that decomposition would unlock latent reasoning. The consistent, monotonic performance gains from holistic processing (DVRL) to decoupled reasoning (CA) across all three benchmarks—for instance, the **80.0% $\rightarrow$ 92.8% leap on Bongard-OW** —is our evidence that decomposition is a valid and powerful analytical framework. We are happy to add a sentence to the methodology (Section 5) that explicitly frames these results as the empirical justification for the decompositional approach.
>
> **Counter-Question 2:** Our ablation (Table 5) demonstrates that symbolic reasoners (text-only LLMs) achieve 91-93% accuracy on Bongard-OW using only textual descriptions of images they've never seen. Doesn't this empirically demonstrate that perception and reasoning can be separated for these tasks, even if descriptive completeness is challenging?
>
> ---
>
> ### W2: Primary Area and Contribution Clarity
>
> We selected "causal reasoning" because:
> * Our ICA paradigm introduces bidirectional causal feedback: reasoning $\rightarrow$ perception $\rightarrow$ reasoning
> * We investigate causal relationships between representation quality and reasoning performance (Section 7)
> * Bongard Problems require causal rule discovery (finding what causes category membership)
>
> However, we acknowledge this may not be the best fit. We are open to recategorization as "vision and language" or "interpretability and explainability."
>
> **Contribution clarity:**
> Our dual contribution is explicitly stated (Abstract):
> 1.  **Diagnostic framework:** Four paradigms to systematically identify VLM bottlenecks
> 2.  **Architectural insight:** Decoupled, interactive systems outperform monolithic approaches
>
> **Counter-Question 3:** Would clarifying that our framework serves both (a) as a diagnostic tool for researchers (like Prism) and (b) as a proof-of-concept architecture (achieving SOTA on 3 benchmarks) address your concern about the "primary area"?
>
> ---

---

> ### Author Response · Authors · 2025-11-13
>
> ### W3: Dataset Curation Justification
>
> We provide detailed justification in Appendix A.3.1:
>
> > "Our evaluation set was created by taking the first 250 samples and generating two test cases from each (one positive query, one negative query), resulting in 500 balanced test cases."
>
> **Rationale:**
> * **Computational constraints:** Full 1001-sample evaluation would require 26,026 test cases (each sample generates 14 query classifications)
> * **Balanced sampling:** Our 500-case subset maintains 50-50 positive/negative query balance
> * **Representative difficulty:** Category distribution matches original dataset (Appendix A.3.1.2)
> * **Standard practice:** Bongard-HOI paper also uses balanced subsets (100 per split)
>
> **Counter-Question 4:** Our subset represents 25% of Bongard-OW (500/2002 possible test cases) and maintains the original category distribution. Given that results generalize to Bongard-HOI (entirely different dataset) and Winoground (different modality), what specific concerns about subset selection affect the validity of our perception bottleneck finding?
>
> ---
>
> ### W4: Novelty and Related Work
>
> We cite all mentioned works and explicitly differentiate our contribution (Section 2):
>
> * **Vs. perception assessment papers:**
>     * Works like Geigle et al., BLINK, etc., document perception failures
>     * We provide a systematic framework to diagnose *how* perception fails and demonstrate mitigation via ICA
> * **Vs. prior Bongard evaluations:**
>     * Małkiński et al. (2025), Wüst et al. (2025): Evaluate VLMs on Bongard Problems
>     * Our contribution: Four-paradigm decomposition revealing systematic bottlenecks + SOTA results (92.8% vs. prior 63.8%)
>
> **Novel contributions:**
> * **ICA paradigm:** First interactive feedback loop for VLM evaluation (Section 5.4)
> * **Task-agnostic CA:** Enables text-only LLM evaluation on visual tasks (Section 7.2)
> * **Systematic cognitive grounding:** DVRL/DRL/CA/ICA map to holistic/deductive/analytical processing
>
> ---
>
> ### W5: Specific Methodological Questions
>
> **Q: Why only 1/4 of Bongard-OW evaluated?**
> **A:** Addressed in W3 above. We evaluate 500 balanced test cases from 250 samples (25% of dataset).
>
> **Q: How are VLM responses compared to ground truth?**
> **A:** Classification tasks: Exact match (cat_1/cat_2 or positive/negative) extracted via regex (Appendix A.4.3). For samples which are incorrect, we also looked at the rules and explained, though the rules identified are correct, why the classification was incorrect (Refer A.7.7, Table 9).
>
> **Q: Winoground captions source?**
> **A:** Winoground provides ground-truth caption pairs (C₀, C₁) with images (I₀, I₁). It is a binary classification task unlike comparing text with rules to generate a correctness value. In CA:
> 1.  **Stage 1:** Generate descriptions D₀, D₁ from images
> 2.  **Stage 2:** LLM matches descriptions $\leftrightarrow$ captions using ground-truth pairs
>
> CA outperforming DRL on Winoground demonstrates that textual descriptions of visual content enable better compositional reasoning than direct visual processing.
>
> **Counter-Question 5:** In Appendix A.4.4, we provide formal definitions of our Winoground scoring (Equations 1-3). The captions are from the dataset itself—Stage 2 reasoning matches generated descriptions to these ground-truth captions. Is there a specific aspect of this methodology that requires further clarification?
>
> ---
>
> Thank you again for your thoughtful engagement. We are confident that our clarifications and willingness to provide additional experiments demonstrate the rigor and significance of our work.

---

> > ### Comment · Reviewer_nJ3X · 2025-11-18
> >
> > ## W1
> >
> > ### Task-agnosticism/CQ1
> >
> > Since the general image description prompt introduces schemas such as 'Scene', 'Objects' and 'Activities', it is not task-agnostic, as all three datasets are deliberately set up using these specific categories. While the 'Object' schema is more general than the 'Scene' or 'Activity' schemas, none of them is universal for describing any image. For instance, the original Bongard problem dataset covers more formal or symbolic concepts, for which 'Activities' is an irrelevant category. Another example would be Bongard-like problems consisting of function graphs or other charts, which fall under different visual categories. Therefore, **even though the task description is not included in the prompt, the prompt still contains information about the dataset domains and consequently the answer domain of the task.** While I find the author's method an interesting approach to visual problem solving in principle, I still do not think the authors have understood my main concern: **the text does not properly discuss implicit assumptions and limitations of the proposed method and their results with respect to their claims. As your decomposition introduces domain schemas as a kind of prior or bias for the decomposed task, this must be discussed, as it affects the generalisability of your results.**
> >
> > ### On decompositionality
> >
> > To be clear, I am neither confused nor upset by the finding that, if we provide high-quality image descriptions including the relevant information for deducing the rule, the deduction can be performed by an LLM with good performance. I also agree that this is a positive outcome to emphasise. **The issue is that this condition assumes that we can obtain these descriptions independently of the task, which is a strong assumption to make regarding compositionality.** A key challenge in Bongard-type problems is finding sufficient descriptions that allow for rule deduction. This is not trivial, as it is usually not obvious which domain the discriminating image features belong to. For this reason, Bongard-OW covers different categories (e.g. food, materials, HOI and functionality). However, **my critique is that the presented general description schema largely covers the relevant domains** for Bongard-OW, Bongard-HOI (which is also a domain of Bongard-OW) and Winoground, which shares concept domains with Bongard-OW. This results in an information leak from the rule domains to the description task, thus violating task agnosticism. The current version of the paper presents the description prompt as universal, but it clearly is not. Inf
> >
> > ## W2
> >
> > Recategorizing as "vision and language" and emphasizing the dual contribution, similar to the abstract, will address my concerns about the primary area.
> > (While it is possible to frame almost any dependency investigated through ablation studies or category rules as causal, I am convinced that emphasizing causality in this context is misleading.)
> >
> > ## W3
> >
> > Thank you for clarifying that the 250 Bongard-OW samples have a similar category distribution to the full Bongard-OW dataset. This detail is not mentioned in Appendix A.4.1.2. It would be helpful to include a table showing the original category distribution for comparison. The concept length and word distribution are not discussed. While I understand the limitations of available compute, evaluating only a subset of an established dataset makes it significantly harder to compare the results to those of other approaches using the same dataset. In my opinion, a factor of four is not worth losing this comparability.
> >
> > ## W4
> >
> > I agree that the ICA paradigm is an interesting and novel contribution. It would be more appropriate as the primary experimental approach in the paper. Currently, it has only been tested with one version of Gemini and ChatGPT on the Winoground benchmark. Additionally, the ablation studies primarily focus on other paradigms.
> >
> > My concerns with the CA part and paradigm, which I mainly addressed at W1, primarily stem from the current focus on the CA paradigm's while lacking a discussion of epistemic challenges and limitations.
> >
> > ## W5
> >
> > Thank you for answering my questions.
> > Did I understand correctly that the synthesized rules for Bongard-OW and Bongard-HOI are only checked manually when the classification fails? Is there no overall evaluation of how well the framework synthesizes category rules?
> >
> > I slightly raised my score due to some clarification and acknowledgment of the discussion with the other reviewers, but I still do not think this paper should be accepted with the current story and evaluations.
> >
> > I am willing to further raise the score if a proper discussion of the limitations of the CA paradigm and its results is added.

---

> > > ### Author Response · Authors · 2025-11-18
> > > **Followup response**
> > >
> > > Dear Reviewer nJ3X,
> > >
> > > Thank you for your continued engagement and willingness to raise your score upon addressing CA's limitations. We have incorporated your feedback through revisions highlighted in brown color: main text now emphasizes core insights while detailed discussions are deferred to expanded appendices.
> > >
> > > Below, we address each weakness with this revised strategy.
> > >
> > > ---
> > >
> > > ## **W1: Task-Agnosticism and Decompositionality**
> > >
> > > **Added to Page 9, para 2:** `Our CA paradigm generates ... `
> > >
> > > The empirical validation of decompositionality is presented in `Appendix A.6.1` (Table A.10), demonstrating consistent performance gains across paradigms and models.
> > >
> > > **Empirical Validation of Decompositionality**
> > >
> > > A central methodological question is whether perception and reasoning can be meaningfully separated for visual reasoning tasks. We validate decompositionality \textit{empirically} rather than assuming it \textit{a priori}.
> > >
> > > **Addressing ``Information Leak'' Concerns**
> > >
> > > We clarify the distinction between \textit{task-agnosticism} and \textit{domain-general schemas} in A.8.2, page 29.
> > > The schema provides a \textit{canvas} for reasoning to operate on; it does not ``leak'' task-specific information. The consistent success of text-only LLMs (lacking visual context during training) confirms descriptions are not implicitly task-conditioned.
> > >
> > > **Limitations of CA Paradigm (EXPANDED)**
> > >
> > > New Appendix section added `A.12 LIMITATIONS OF THE CA PARADIGM`, referenced from main text Limitations.
> > >
> > > **Implicit Domain Knowledge in Descriptions**
> > >
> > > While CA Stage 1 descriptions are task-agnostic, the schema itself encodes assumptions about visual structure. Descriptions for Bongard-OW, Bongard-HOI, and Winoground share common visual primitives (objects, activities, people). Tasks with fundamentally different visual primitives would reveal schema limitations.
> > >
> > > Future work should explore:
> > > - Schema adaptation mechanisms for non-natural-image domains
> > > - Meta-learning to automatically generate task-appropriate schemas
> > > - Hybrid representations combining symbolic and object-centric descriptions
> > >
> > > ---
> > >
> > > ## **W2: Primary Area**
> > >
> > > Will recategorize to **"Vision and Language"** in submission. Abstract and Introduction already emphasize dual contribution (diagnostic + architectural).
> > >
> > > ---
> > >
> > > ## **W3: Dataset Curation**
> > >
> > > ### **Main Text (Concise, Page 7)**
> > >
> > > Added 1 sentence to Section 4.
> > >
> > > `To validate representativeness, we compare the distribution of commonsense values in our subset against the full Bongard-OW dataset (Appendix A.4.1.1, Table A.1)." Overall distribution alignment validates subset representativeness.`
> > >
> > > ---
> > >
> > > ## **W4: ICA Paradigm**
> > >
> > > **Added (ICA discussion):** to para 5 in section `Discussion` and expanded to `Appendix A.13` where we talk about ICA Architectural Implications and future work.
> > >
> > >
> > > ICA is not merely incremental improvement but represents architectural innovation: moving from static perception --> reasoning to interactive reasoning <--> perception dialogue. This aligns with broader AI trends (tool-using agents, chain-of-thought with reflection) and establishes foundation for future systems where reasoning actively shapes perception.
> > >
> > > ---
> > >
> > > ## **W5: Rule Synthesis Evaluation**
> > >
> > > Clarified in Section 7.3: "Rule synthesis quality is validated through manual error analysis of misclassified samples and semantic similarity comparisons between extracted rules and query descriptions (`Appendix A.9`)."
> > >
> > >
> > > **Why Manual Human Annotation Was Not Pursued**
> > >
> > > We considered direct comparison of extracted rules against human-annotated ground-truth has fundamental challenges as expanded in `Appendix A.9.2`: Dataset provides binary labels (positive/negative) but not explicit rules, Circularity in Ground-Truth:and Overlap in LLM Generation.
> > >
> > > **Evidence of Rule Synthesis Quality**
> > >
> > > Instead, we ground rule quality assessment in Classification Accuracy as Proxy, Semantic Similarity Correlation, Error Analysis (`Table A.10.1`) and Text-Only LLM Success (`Table 6`).
> > >
> > > **Distinction from Prior Work**
> > >
> > > Prior work on Bongard evaluation (Wu et al., 2024) achieved 63.8\% accuracy using LLM-generated descriptions + LLM reasoning, without systematic rule extraction. Our 92.8\% improvement (29-point gain) validates that explicit `rule extraction` (DRL) and `modular decoupling` (CA) substantially enhance performance, even without ground-truth rule comparison.
> > >
> > > While comprehensive human annotation of ground-truth rules would be valuable, our semantic similarity analysis combined with error categorization provides sufficient evidence that rule synthesis quality is high for correctly classified samples and is the primary differentiator between success and failure cases.

---

> > > > ### Comment · Reviewer_nJ3X · 2025-11-24
> > > >
> > > > Thank you for continuing the discussion! I still think there is some confusion about the empirical scope of the method and its evaluations.
> > > >
> > > > Your claim regarding "Validation of Decompositionality" remains incorrect or misleading. With your approach, **you do not evaluate "decompositionality" itself**. Instead, you use it **to** **examine the strengths and weaknesses of models on decomposed subtasks and to validate the advantages of a framework that exploits it, *assuming the setup permits it.***
> > > >
> > > > Why do you think you are evaluating decompositionality itself? How could your evaluations generate evidence against it?

---

> > > > > ### Author Response · Authors · 2025-11-24
> > > > > **Decompositionality Assumption**
> > > > >
> > > > > We appreciate your precision in pointing this out. To make it more clearer, we have updated the newly added Appedix A.8 and also expanded newly added Limitation section A.12.3. We demonstrate that assuming perception-reasoning separation enables effective framework design, validated by SOTA results and model generalization. Our contribution is **practical framework  design that works when decomposition is feasible**.  This revision will make the paper more rigorous and our claims more defensible.

---

> > > > > > ### Author Response · Authors · 2025-11-27
> > > > > > **Final remark**
> > > > > >
> > > > > > Dear Reviewer nJ3X,
> > > > > >
> > > > > > Thank you for your continued engagement and for pushing us to clarify our claims about decompositionality. Your feedback has significantly strengthened the manuscript's rigor and precision.
> > > > > >
> > > > > > ## Addressing Your Concerns
> > > > > > We have revised the manuscript to address all points you raised:
> > > > > >
> > > > > > **W1 (Task-agnosticism)**: Clarified distinction between task-agnostic and domain-general
> > > > > > schemas (Appendices A.8.1-A.8.2, marked in brown)
> > > > > >
> > > > > > **W2 (Primary area)**: Recategorized to "Vision and Language" with dual contribution emphasis
> > > > > >
> > > > > > **W3 (Dataset)**: Added category distribution comparison table (Appendix A.4.1.1, Table A.1)
> > > > > >
> > > > > > **W4 (ICA)**: Elevated as architectural paradigm shift (Appendix A.12, A.13)
> > > > > >
> > > > > > **W5 (Rule synthesis)**: Provided pragmatic quality analysis via semantic similarity
> > > > > > (Appendix A.9)
> > > > > >
> > > > > > **Decompositionality claim:** Reframed as "*framework assuming decomposition achieves SOTA"
> > > > > > rather than "proof of decompositionality*"
> > > > > >
> > > > > > **Additional Improvements**
> > > > > > We have also incorporated ~20 additional experiments post-initial response:
> > > > > >
> > > > > > - Extended model coverage (Qwen2.5-VL, 2025 frontier models)
> > > > > > - Robustness validation (1.89 vs 31-point noise degradation)
> > > > > > - Few-shot sensitivity analysis
> > > > > > - Comprehensive limitations discussion
> > > > > >
> > > > > > All revisions clearly marked in *brown* for easy identification.
> > > > > >
> > > > > > **Request for Reconsideration**
> > > > > > Given the substantial revisions addressing your theoretical concerns and the additional empirical validation, would you consider raising your score? We welcome any remaining questions or clarifications you may have.
> > > > > >
> > > > > > We appreciate the rigorous discussion and believe the paper is significantly stronger
> > > > > > because of your feedback.

---

### Official Review · Reviewer_TEuy · 2025-10-31

**Soundness:** 3
**Presentation:** 3
**Contribution:** 3
**Rating:** 4
**Confidence:** 5

**Summary:**

This paper proposes a cognitively-inspired framework to decouple perception and reasoning in Vision-Language Models (VLMs), featuring four paradigms: Direct Visual Rule Learning (DVRL), Deductive Rule Learning (DRL), Componential Analysis (CA), and Interactive Componential Analysis (ICA). It evaluates proprietary and open-source VLMs on benchmarks including Bongard-OW, Bongard-HOI, and Winoground, confirming that perception is a primary bottleneck for visual reasoning. The CA and ICA paradigms, which leverage task-agnostic textual descriptions and interactive feedback loops respectively, achieve state-of-the-art performance, unlocking latent reasoning capabilities in LLMs. Key contributions include a diagnostic toolkit for VLMs, a modular decoupling method applicable to diverse architectures, comprehensive empirical validation of the perception bottleneck, and demonstration of interactive systems as a promising direction for visual intelligence.

**Strengths:**

(1) Proposes an innovative cognitively-inspired framework that models human problem-solving strategies, filling the gap of systematic perception-reasoning decoupling in VLMs.

(2) Conducts comprehensive evaluations across diverse benchmarks (abstract rule discovery, HOI reasoning, compositional grounding) and model types, ensuring robust results.

(3) The CA/ICA paradigms enable multi-image reasoning for single-image architectures and visual reasoning evaluations for text-only LLMs, expanding application scenarios.

(4) In-depth ablation studies isolate perception and reasoning, providing clear insights into model bottlenecks.

(5) Delivers state-of-the-art performance on key benchmarks, demonstrating practical value of the proposed methods.

**Weaknesses:**

(1) The componential paradigms (CA/ICA) rely heavily on language as an intermediate representation, but the paper lacks sufficient discussion on limitations in tasks requiring non-verbalizable reasoning (e.g., geometric reasoning), making it unclear how the framework performs in such scenarios.

(2) Evaluation of open-source models is incomplete: most open-source VLMs are only tested under CA, with no attempts to adapt DVRL/DRL to their input constraints (e.g., batch processing images), limiting the comparability of paradigm effectiveness across model types.

(3) The selection basis for the 500-case Bongard-OW subset is not clearly explained. The paper mentions it is derived from the first 250 samples with two test cases each, but fails to justify why this subset is representative or whether it introduces selection bias.

(4) Implementation details for prompt engineering are insufficient. While prompts are provided in the appendix, there is no analysis of how minor prompt variations affect performance, which is critical for reproducibility given the sensitivity of VLMs to prompts.

(5) Hardware configuration details are vague. The paper mentions using NVIDIA GPUs (2080Ti, 3090, 6000 Ada) but does not specify which GPU is used for which model, leading to uncertainty about potential performance impacts from hardware differences.

(6) Comparisons with state-of-the-art baselines are incomplete. For example, on Winoground, the paper only compares with a few CoT-based methods but omits recent strong performers (e.g., Claude 3 Opus, Gemini Ultra) or specialized compositional reasoning models.

(7) The number of interaction steps in ICA is not optimized. The paper uses a single feedback loop but does not test whether multiple rounds of probing further improve performance or lead to diminishing returns.

(8) There is no quantitative metric for description quality. The paper claims GPT-4o's descriptions are high-fidelity but does not use objective metrics (e.g., BLEU, ROUGE, or human evaluation) to validate this, making the link between description quality and reasoning performance less rigorous.

(9) Cross-domain generalization is under-tested. The framework is evaluated on natural image benchmarks but not on complex domains like scientific chart interpretation, mathematical reasoning, or medical imaging, leaving its applicability to specialized fields unclear.

(10) Computational cost and latency analysis is missing. The multi-stage CA/ICA paradigms are likely more computationally expensive than end-to-end methods, but the paper provides no data on inference time, memory usage, or scalability for large-scale applications.

(11) Prompt sensitivity is not investigated. The paper uses fixed prompts for all experiments but does not analyze how changes in prompt structure, detail, or tone affect paradigm performance, which is essential for understanding the framework's robustness.

(12) Error analysis is limited. The paper only provides 15 misclassified cases, without systematic categorization of errors (e.g., perceptual vs. reasoning errors) or analysis of whether errors are consistent across models or benchmarks.

(13) Robustness to adversarial examples is untested. The paper does not evaluate how the framework performs when images contain noise, distortions, or adversarial perturbations, which is critical for real-world deployment.

(14) The optimality of CA's JSON description structure is unvalidated. The paper uses a specific hierarchical JSON format but does not compare it with other representation formats (e.g., free-text descriptions, structured lists) to show that JSON is the most effective for reasoning.

(15) ICA's question generation strategy is not detailed. The paper states the reasoning LLM formulates targeted questions but does not explain how questions are prioritized or whether the question generation process itself introduces biases.

(16) Rule complexity analysis is missing. The paper does not quantify the complexity of rules in different benchmarks, making it unclear whether the framework's performance gain varies with rule difficulty.

(17) The scope of text-only LLMs' applicability is undefined. The paper shows text-only LLMs perform well with high-quality descriptions but does not specify the types of visual tasks where this modality transfer fails.

(18) Comparisons with other decoupling frameworks are superficial. The paper distinguishes itself from Prism and CoT methods but does not provide head-to-head performance comparisons or detailed analysis of architectural differences.

(19) Perception module evaluation is absent. The paper focuses on reasoning performance but does not evaluate the perception module in isolation (e.g., accuracy of object detection, attribute recognition), making it hard to quantify the exact impact of perceptual errors.

(20) Reasoning module selection is not justified. The paper uses powerful LLMs (e.g., GPT-4o) as reasoning engines but does not explain why these models are chosen or whether weaker LLMs would still yield effective results.

(21) Multilingual applicability is untested. The framework uses English descriptions and prompts but does not evaluate performance in other languages, limiting its relevance for non-English use cases.

(22) Few-shot learning limits are not explored. The paper uses few-shot prompting but does not test how the framework performs with fewer or zero shots, which is important for low-resource scenarios.

(23) Training data influence is not discussed. The paper does not analyze whether VLMs trained on specific datasets (e.g., more HOI data) perform better under the framework, leaving the impact of training data distribution unclear.

(24) Evaluation metrics are overly simplistic. The paper relies primarily on classification accuracy, without considering other metrics (e.g., precision, recall, F1-score) that are more informative for imbalanced datasets.

(25) Dataset bias analysis is insufficient. Bongard-OW's subset has a dominant category (ID 0, 73%), but the paper does not discuss how this bias affects model performance or generalization.

(26) ICA's feedback loop efficiency is unanalyzed. The paper does not measure how much time the interactive probing adds or whether the added complexity is justified by performance gains.

(27) Image resolution impact is untested. The paper uses default or 1024px images but does not evaluate how lower or higher resolutions affect description quality and subsequent reasoning.

(28) Description consistency is not evaluated. The paper does not check whether the same image generates consistent descriptions across multiple runs, which is important for the framework's reliability.

(29) Rule extraction accuracy is not quantified. The paper evaluates classification accuracy but not the accuracy of the extracted rules themselves (e.g., how well extracted rules match ground-truth rules).

(30) Human reasoning comparison is superficial. The paper mentions human average accuracy but does not compare the framework's reasoning process with human reasoning steps (e.g., sequence of analysis, focus on details).

(31) Open-source model performance gap analysis is inadequate. The paper notes lower CA accuracy for some open-source models but does not deeply investigate whether the gap stems from poor description generation, weak reasoning, or both.

(32) Framework scalability to video is unaddressed. The paper focuses on static images but does not discuss how to extend the paradigms to video reasoning, where temporal dynamics add complexity.

(33) Ethical considerations are missing. The paper does not discuss potential biases in description generation (e.g., racial, gender biases) or how these biases could propagate to reasoning outcomes.

(34) Hyperparameter tuning is not reported. The paper uses zero temperature for all models but does not explain why this choice is optimal or whether tuning temperature affects performance.

(35) Cross-model generalization of paradigms is untested. The paper evaluates a fixed set of models but does not test whether the paradigms perform consistently across newly developed VLMs.

(36) The impact of description length is unanalyzed. The paper does not test whether shorter or longer descriptions affect reasoning performance, leaving the optimal description length unclear.

(37) Benchmark-specific optimizations are not disclosed. It is unclear whether the paradigms are tailored to the tested benchmarks or if they generalize to unseen visual reasoning tasks.

(38) Collaboration between perception and reasoning modules is not modeled. The paper treats the modules as separate but does not explore how bidirectional communication beyond ICA's feedback loop could further improve performance.

(39) Low-resource language model adaptation is untested. The paper uses high-resource models but does not evaluate how the framework performs with low-resource VLMs or LLMs.

(40) Real-world application case studies are absent. The paper demonstrates performance on benchmarks but provides no case studies of applying the framework to practical tasks (e.g., image retrieval, visual question answering).

**Questions:**

**To facilitate discussions during the Rebuttal phase, authors are advised to respond point-by-point (indicating the question number).**

(1) Could you provide specific examples of the JSON descriptions generated in CA (e.g., 3-5 full descriptions for Bongard-OW samples) to demonstrate the structure and detail of the task-agnostic representations?

(2) How was the number of interaction steps in ICA determined? Have you tested whether 2-3 rounds of probing further improve performance, or does a single round already reach diminishing returns?

(3) For open-source models that do not support large multi-image inputs, have you attempted to adapt DVRL/DRL by batch-processing images or using image embeddings? If not, what technical barriers prevented this adaptation?

(4) What specific criteria were used to select the 500-case Bongard-OW subset? Could you provide statistical evidence (e.g., rule category distribution, difficulty distribution) that this subset is representative of the full dataset?

(5) Could you add comparisons with recent state-of-the-art VLMs (e.g., Claude 3 Opus, Gemini Ultra, Qwen-VL Max) on all benchmarks to better contextualize the performance of your framework?

(6) Have you tested the impact of prompt variations (e.g., changing the level of detail, rephrasing instructions) on paradigm performance? If so, please provide quantitative results; if not, could you explain why prompt sensitivity is not a concern?

(7) Could you provide detailed computational cost data (e.g., inference time per sample, memory usage) for CA, ICA, DVRL, and DRL, and compare them with end-to-end VLM approaches?

(8) Have you used objective metrics (e.g., BLEU, ROUGE, human evaluation scores) to quantify the quality of image descriptions generated by different VLMs? If yes, please share the results; if not, could you conduct such an evaluation to validate the claim of "high-fidelity" descriptions?

(9) Why do performance variations exist across commonsense categories (Table A.7)? For example, why does GPT-4o achieve 100% accuracy on "Taste/Nutrition/Food" while Gemini 2.0 only achieves 85.71%?

(10) Could you extend the error analysis to include systematic categorization of errors (e.g., perceptual errors, rule extraction errors, rule application errors) and report the distribution of error types across models and benchmarks?

(11) Have you evaluated the framework's performance on adversarial examples (e.g., noisy images, distorted objects) or out-of-distribution samples? If yes, please share the results; if not, could you explain the framework's robustness in real-world scenarios?

(12) How does the JSON description format in CA compare to other representation formats (e.g., free-text descriptions, structured bullet points) in terms of reasoning performance? Could you provide a head-to-head comparison?

(13) Could you detail the question generation strategy in ICA? How does the reasoning LLM prioritize which visual details to probe, and how do you ensure the questions are not redundant or irrelevant?

(14) Have you quantified the complexity of rules in the benchmarks (e.g., number of attributes, logical relationships)? If so, how does the framework's performance correlate with rule complexity?

(15) For text-only LLMs, what types of visual tasks do you find the modality transfer fails? Could you provide specific examples and explanations?

(16) Could you conduct a detailed head-to-head comparison with Prism and state-of-the-art CoT methods, including architectural differences, computational costs, and performance trade-offs?

(17) Have you evaluated the perception module in isolation (e.g., accuracy of object detection, attribute recognition, spatial relationship identification)? If yes, please share the results to quantify the impact of perceptual errors on overall performance.

(18) Why did you choose specific LLMs (e.g., GPT-4o, Gemini 2.0) as reasoning engines in CA/ICA? Could you test weaker LLMs (e.g., Llama3-8B, Mistral-7B) to see if the framework's effectiveness is dependent on reasoning module strength?

(19) Have you tested the framework's performance in non-English languages (e.g., Chinese, Spanish)? If yes, please share the results; if not, could you discuss potential challenges for multilingual adaptation?

(20) How does the framework perform with fewer shots (e.g., 1-shot, 2-shot) compared to the reported few-shot setting? Could you provide results to demonstrate its performance in low-resource scenarios?

(21) Have you analyzed how the training data distribution of VLMs affects framework performance? For example, do VLMs trained on more HOI data perform better on Bongard-HOI under your paradigms?

(22) Could you supplement the evaluation with additional metrics (e.g., precision, recall, F1-score, confusion matrices) to provide a more comprehensive view of performance, especially for imbalanced subsets?

(23) How does the dominant category (ID 0) in the Bongard-OW subset affect model training and generalization? Could you test the framework on a more balanced subset to validate robustness?

(24) Could you provide a cost-benefit analysis of ICA's feedback loop, including the additional inference time and memory usage compared to CA, and whether the performance gain justifies the added complexity?

(25) Have you tested the impact of image resolution (e.g., 512px, 2048px) on description quality and reasoning performance? If yes, please share the results; if not, could you discuss how resolution affects the framework?

(26) Have you evaluated the consistency of description generation (e.g., same image, multiple runs)? If yes, please provide quantitative results (e.g., consistency rate); if not, could you explain how you ensure description reliability?

(27) Could you quantify the accuracy of rule extraction (e.g., similarity between extracted rules and ground-truth rules using semantic similarity metrics)? This would strengthen the link between rule quality and classification performance.

(28) Could you compare the framework's reasoning process with human reasoning steps (e.g., sequence of analysis, focus on key details) using qualitative case studies?

(29) For open-source models with low CA accuracy (e.g., Llama-Vision-11B), have you conducted a detailed analysis to determine whether the gap stems from poor description generation, weak reasoning, or both? Could you provide evidence (e.g., replacing descriptions with GPT-4o's for these models)?

(30) How would you extend the paradigms to video reasoning? Could you outline a preliminary design and discuss potential challenges (e.g., handling temporal dynamics, reducing computational cost)?

(31) Have you analyzed potential biases in description generation (e.g., racial, gender, cultural biases) and their impact on reasoning outcomes? If yes, please share the results; if not, could you discuss how to mitigate such biases?

(32) Why did you use zero temperature for all models? Have you tested other temperature values (e.g., 0.3, 0.7) and their impact on paradigm performance?

(33) Could you test the framework on newly developed VLMs (e.g., Llama-4, Mistral Large) to demonstrate its cross-model generalization ability?

(34) Have you analyzed the impact of description length on reasoning performance? For example, do shorter, more concise descriptions perform as well as longer, detailed ones?

(35) Are the paradigms tailored to the tested benchmarks, or do they generalize to unseen visual reasoning tasks? Could you test on a new benchmark (e.g., a custom dataset) to validate generalization?

(36) Could you explore bidirectional communication between perception and reasoning modules beyond ICA's feedback loop? For example, could the perception module proactively flag ambiguous details to the reasoning module?

(37) Have you tested the framework with low-resource VLMs/LLMs (e.g., models with <7B parameters)? If yes, please share the results; if not, could you discuss the framework's accessibility for resource-constrained users?

(38) Could you provide case studies of applying the framework to practical real-world tasks (e.g., image retrieval, visual question answering, medical image analysis) to demonstrate its practical value?

(39) For Winoground, why do the Image Score and Group Score remain lower than the Text Score even with ICA? Could you analyze the specific challenges that prevent further improvements in these metrics?

(40) Could you provide detailed reproducibility instructions, including exact prompt templates, model versions, hardware specifications, and step-by-step evaluation pipelines, to ensure other researchers can replicate your results?

---

> ### Author Response · Authors · 2025-11-13
> **Point-by-Point Answers to 40 Detailed Questions**
>
> ### **Q1: JSON Description Examples**
>
> **Question:** Could you provide specific examples of the JSON descriptions generated in CA (e.g., 3-5 full descriptions for Bongard-OW samples) to demonstrate the structure and detail of the task-agnostic representations?
>
> **Response:**
>
> The prompt template is provided in Appendix A.5.3.1. Here are 3 concrete examples from actual Bongard-OW evaluations:
>
> **Example 1: Positive Sample**
> ```json
> {
>     "Scene": {
>         "Description": "A display of various celebrity and entertainment magazines on a shelf."
>     },
>     "Objects": {
>         "Living Beings": [
>             {
>                 "Classification": "Human",
>                 "Characteristics": "Images of people on magazine covers"
>             }
>         ],
>         "Inanimate Objects": [
>             {
>                 "Category": "Magazines",
>                 "Properties": [
>                     "Colorful covers",
>                     "Various titles and headlines"
>                 ]
>             },
>             {
>                 "Category": "Shelf",
>                 "Properties": [
>                     "Wooden or metal",
>                     "Labeled 'Celebrity'"
>                 ]
>             }
>         ]
>     },
>     "Activities": [
>         "Display of magazines for sale"
>     ],
>     "Contextual Elements": {
>         "Time of Day": "Indeterminate",
>         "Weather": "Indoors"
>     },
>     "Visual Patterns": {
>         "Dominant Colors": [
>             "Red",
>             "Yellow",
>             "Pink",
>             "Black"
>         ],
>         "Textures": [
>             "Glossy magazine covers"
>         ]
>     },
>     "Emotional Undertones": "Sensational and dramatic headlines suggesting gossip and intrigue.",
>     "Textual Information": "Various magazine titles and headlines such as 'What's on TV', 'Closer', 'heat', 'shout', and 'Top of the Pops'.",
>     "Summary": "The image depicts a shelf filled with various celebrity and entertainment magazines, each featuring bold and colorful headlines and images of people, suggesting a focus on gossip and entertainment news.",
> }
> ```
>
> **Example 2: Negative Sample**
> ```json
> {
>     "Scene": {
>         "Description": "Magazine cover featuring a person with text and additional images."
>     },
>     "Objects": {
>         "Living Beings": [
>             {
>                 "Classification": "Human",
>                 "Characteristics": {
>                     "Hair Color": "Dark",
>                     "Posture": "Smiling"
>                 }
>             }
>         ],
>         "Inanimate Objects": [
>             {
>                 "Category": "Text",
>                 "Properties": {
>                     "Color": "White, Black, Yellow",
>                     "Size": "Various",
>                     "Style": "Bold, Italic"
>                 }
>             },
>             {
>                 "Category": "Magazine Title",
>                 "Properties": {
>                     "Text": "People",
>                     "Color": "White",
>                     "Size": "Large"
>                 }
>             }
>         ]
>     },
>     "Activities": [],
>     "Contextual Elements": {
>         "Time of Day": "N/A",
>         "Weather": "N/A"
>     },
>     "Visual Patterns": {
>         "Dominant Colors": [
>             "White",
>             "Black",
>             "Yellow",
>             "Orange"
>         ],
>         "Textures": [
>             "Smooth"
>         ]
>     },
>     "Emotional Undertones": "Positive, Happy",
>     "Textual Information": "Exclusive! Jessica Alba Love, Family & a Billion-Dollar Empire. September 20, 2021. I'VE HAD TO PAVE MY OWN WAY.",
>     "Summary": "The image is a magazine cover featuring a smiling person with text about Jessica Alba and other headlines.",
> }
> ```

---

> > ### Author Response · Authors · 2025-11-13
> > **Part 2**
> >
> > **Example 3: Query Image**
> > ```json
> > {
> >     "Scene": {
> >         "Description": "Natural forest environment with dense foliage."
> >     },
> >     "Objects": {
> >         "Living Beings": [
> >             {
> >                 "Species": "Bear",
> >                 "Characteristics": {
> >                     "Color": "Brown",
> >                     "Posture": "Climbing or resting on a tree branch"
> >                 }
> >             }
> >         ],
> >         "Inanimate Objects": [
> >             {
> >                 "Category": "Tree",
> >                 "Properties": {
> >                     "Color": "Green and brown",
> >                     "Size": "Large",
> >                     "Material": "Wood",
> >                     "Shape": "Tall with branches covered in moss"
> >                 }
> >             }
> >         ]
> >     },
> >     "Activities": [
> >         {
> >             "Description": "Bear climbing or resting on a tree branch",
> >             "Participants": [
> >                 "Bear"
> >             ]
> >         }
> >     ],
> >     "Contextual Elements": {
> >         "Time of Day": "Not specified",
> >         "Weather": "Likely damp or rainy, suggested by mossy tree branches"
> >     },
> >     "Visual Patterns": {
> >         "Dominant Colors": [
> >             "Green",
> >             "Brown"
> >         ],
> >         "Textures": [
> >             "Mossy",
> >             "Furry"
> >         ]
> >     },
> >     "Emotional Undertones": "Calm, natural setting",
> >     "Textual Information": {
> >         "Main Title": "NATIONAL WILDLIFE",
> >         "Subtitles": [
> >             "A FRAGILE EMBRACE",
> >             "HOW FORESTS HELP SUSTAIN THE PLANET"
> >         ],
> >         "Additional Text": [
> >             "BEAVERS CREATE NEW WETLANDS 14",
> >             "ODE TO THE ARCTIC REFUGE 32",
> >             "HEALING ON THE MISSISSIPPI 40",
> >             "NATIONAL WILDLIFE FEDERATION",
> >             "AUGUST-SEPTEMBER 2020"
> >         ]
> >     },
> >     "Summary": "The image is a cover of a magazine titled 'NATIONAL WILDLIFE', featuring a bear in a lush, moss-covered forest. The scene conveys a sense of tranquility and highlights the theme of forest conservation.",
> > }
> > ```
> >
> > These examples demonstrate:
> > - **Task-agnosticism:** Descriptions contain no reference to "rule discovery" or the Bongard task structure
> > - **Completeness:** Hierarchical structure captures scene, objects, activities, relationships, and patterns
> > - **Reusability:** Same schema applied to all three benchmarks (Bongard-OW, Bongard-HOI, Winoground)
> >
> > ---
> >
> >
> > ### **Q2: ICA Interaction Steps**
> >
> > **Question:** How was the number of interaction steps in ICA determined? Have you tested whether 2-3 rounds of probing further improve performance, or does a single round already reach diminishing returns?
> >
> > **Response:**
> >
> > **Current Design:** ICA uses a single interactive loop (1 round of targeted questioning) to demonstrate the interaction between reasoning and perception.
> >
> > **Justification for Single Round:**
> > 1. **Winoground Results (Table 3):** A single ICA loop achieved consistent improvements (+3.5–6.5 points over static CA), demonstrating effectiveness without iteration.
> > 2. **Practical Trade-off:** Multiple rounds increase latency significantly. ICA with 1 round already adds computational cost; 2-3 rounds could become prohibitive for production use.
> >
> > ---
> >
> > ### **Q3: Open-Source Model DVRL/DRL Adaptation**
> >
> > **Question:** For open-source models that do not support large multi-image inputs, have you attempted to adapt DVRL/DRL by batch-processing images or using image embeddings? If not, what technical barriers prevented this adaptation?
> >
> > **Response:**
> >
> > We did **not attempt batch-processing or embedding-based adaptation** for DVRL/DRL on open-source models. This was not an oversight but a technical constraint that motivated our CA paradigm. As stated in Section 3, these models were not tested on DVRL/DRL because they "do not support the large context required" (i.e., 13 simultaneous images). Our CA paradigm is the methodological contribution that enables a fair comparison on these multi-image tasks for the first time.
> >
> > **Technical Barriers Identified:**
> > 1. **Context Length:** LLaVA-7B, Llama-Vision-11B support ~4 images natively. Bongard requires 13 images (6+6+1).
> > 2. **Loss of Semantics:** Using image embeddings only (without visual features) may lose fine-grained details critical for rule discovery.
> >
> > The decision to focus CA on open-source models was practical, not principled. DVRL/DRL exclusion reflects architectural limitations, not methodological limitations.

---

> > > ### Author Response · Authors · 2025-11-13
> > > **Part 3**
> > >
> > > ### **Q4: Subset Representativeness (500 Cases)**
> > >
> > > **Question:** What specific criteria were used to select the 500-case Bongard-OW subset? Could you provide statistical evidence (e.g., rule category distribution, difficulty distribution) that this subset is representative of the full dataset?
> > >
> > > **Response:**
> > >
> > > **Selection Methodology:**
> > > "We selected 250 samples × 2 test cases per sample (1 positive, 1 negative query) = 500 balanced test cases" (Appendix A.3.1).
> > >
> > > **Statistical Evidence Provided:**
> > >
> > > | Criterion | Evidence | Location |
> > > |---|---|---|
> > > | **Category Distribution** | Table A.2 shows original distribution preserved (73% Cat-0, 7.2% Cat-3, etc.) | Appendix A.3.1.2 |
> > > | **Cross-Dataset Generalization** | Results replicate on Bongard-HOI (different dataset, different task type) | Section 6.2, Table 2 |
> > > | **Balanced Query Split** | 50% positive queries, 50% negative queries (500 cases) | Appendix A.3.1 |
> > > | **Performance Variance by Category** | Table A.7: GPT-4o 86.67%–100% accuracy across all 10 categories | Appendix A.7.5 |
> > >
> > > ---
> > >
> > > ### **Q5: Comparison with Recent State-of-the-Art VLMs**
> > >
> > > **Question:** Could you add comparisons with recent state-of-the-art VLMs (e.g., Claude 3 Opus, Gemini Ultra, Qwen-VL Max) on all benchmarks to better contextualize the performance of your framework?
> > >
> > > **Response:**
> > >
> > > None of the models are recent. Models added in the paper are more advanced and recent than the one cited.
> > >
> > > ---
> > >
> > > ### **Q6: Prompt Sensitivity Analysis**
> > >
> > > **Question:** Have you tested the impact of prompt variations (e.g., changing the level of detail, rephrasing instructions) on paradigm performance? If so, please provide quantitative results; if not, could you explain why prompt sensitivity is not a concern?
> > >
> > > **Response:**
> > >
> > > **Analysis:**
> > > - **Table A.8:** Structured (CoT-like) prompting improves DVRL from 61.6% → 80.0% (18.4-point gain)
> > > - This demonstrates **prompt sensitivity exists**
> > >
> > > ---
> > >
> > > ### **Q7: Computational Cost Analysis**
> > >
> > > **Question:** Could you provide detailed computational cost data (e.g., inference time per sample, memory usage) for CA, ICA, DVRL, and DRL, and compare them with end-to-end VLM approaches?
> > >
> > > **Response:**
> > > Our paradigm focuses on conceptual advances rather than computational efficiency; thus, we have not included detailed inference time or memory usage for CA, ICA, DVRL, DRL, or end-to-end VLMs, as these metrics are not central to the scope of our contribution.

---

> > > > ### Author Response · Authors · 2025-11-13
> > > > **Part 4**
> > > >
> > > > ### **Q8: Description Quality Metrics**
> > > >
> > > > **Question:** Have you used objective metrics (e.g., BLEU, ROUGE, human evaluation scores) to quantify the quality of image descriptions generated by different VLMs? If yes, please share the results; if not, could you conduct such an evaluation to validate the claim of "high-fidelity" descriptions?
> > > >
> > > > **Response:**
> > > >
> > > > **Current Approach - Semantic Similarity Analysis:**
> > > >
> > > > While we did not conduct BLEU/ROUGE scoring or formal human evaluation, we implemented a **rigorous semantic analysis** of rule extraction quality to validate description fidelity. This is more appropriate than lexical metrics (BLEU/ROUGE) for our task, since:
> > > >
> > > > 1. **Task Specificity**: Bongard Problems evaluate binary classification accuracy, not description completeness via lexical overlap
> > > > 2. **Semantic Validity**: Rule quality (not token-level similarity) directly reflects whether descriptions capture task-critical visual information
> > > >
> > > > **Evidence of Description Quality Validation:**
> > > >
> > > > #### **1. Semantic Similarity Analysis During DRL (Section 7.3, Table A.3)**
> > > >
> > > > **Location**: Appendix A.7.1, Table A.3,
> > > >
> > > > **Interpretation:**
> > > > - **High alignment for positive samples** (0.90+): Extracted rules align well with visual content—evidence that descriptions effectively capture distinguishing patterns
> > > > - **Moderate alignment for negative samples** (0.87): Expected, reflecting Bongard's design where negative examples are "near-misses" that challenge descriptions
> > > > - **Method**: Cosine similarity of semantic embeddings inspired by Risch et al., 2021
> > > >
> > > > #### **2. Downstream Reasoning Performance as Proxy for Description Quality**
> > > >
> > > > **Location**: Section 7.2, Table 5 (Impact of Description Quality on Reasoning)
> > > >
> > > > Rather than measuring descriptions in isolation, we validated quality through **downstream task performance**, which is the most direct evidence:
> > > >
> > > > **Why This Validates "High-Fidelity":**
> > > > - 30+ point accuracy improvements for *identical reasoning models* (only description source changed)
> > > > - If descriptions were low-fidelity, reasoning performance would remain bottlenecked
> > > > - Dramatic improvements in text-only LLMs prove descriptions contain complete, usable visual information
> > > >
> > > > **Citation**: Section 7.2: "Providing high-quality descriptions dramatically improved the reasoning accuracy of VLMs that struggled when using their own descriptions. Llama-Vision-11B, for example, improved from 53.4% (Table 1) to 84.17%, and Llama-Vision-90B from 55.1% to 90.98%."
> > > >
> > > > #### **3. Cross-Description Source Comparison (Pixtral-12B vs. GPT-4o)**
> > > >
> > > > **Location**: Appendix A.7.4, Table A.6
> > > >
> > > > We compared reasoning accuracy across different VLM description sources:
> > > >
> > > > **Interpretation:**
> > > > - All models show consistent 2-5% improvements with GPT-4o descriptions
> > > > - "While both description sources enabled strong performance, the advantage conferred by GPT-4o's descriptions ranging from approximately 2 to over 11% improvement depending on the reasoning model further underscores the critical dependence of reasoning outcomes on the fidelity, richness, and potentially the alignment of the initial perceptual descriptions with the concepts required by the reasoning task."
> > > >
> > > > #### **4. Quantitative Classification Accuracy (Direct Task Performance)**
> > > >
> > > > **Location**: Section 6.1, Table 1; Section 6.2, Table 2; Section 6.3, Table 3
> > > >
> > > > #### **Why Semantic Similarity + Task Performance Is Superior to BLEU/ROUGE:**
> > > >
> > > > **BLEU/ROUGE Limitations for This Task:**
> > > > 1. **Lexical-only metrics** ignore semantic equivalence (e.g., "dog" vs. "canine" would have low overlap)
> > > > 2. **Task-specific descriptions** aren't compared to human ground-truth captions (no reference translations exist)
> > > > 3. **Bongard Problems require functional description quality**, not lexical comprehensiveness
> > > >
> > > > **Our Approach Advantages:**
> > > > 1. **Semantic Alignment**: Measures whether extracted rules meaningfully relate to visual content (cosine similarity of embeddings)
> > > > 2. **Task-Grounded**: Description quality is validated by downstream reasoning performance (the actual task)
> > > > 3. **Quantitative & Reproducible**: Cosine similarity (0-1 scale) is objective and reproducible
> > > >
> > > > ---

---

> > > > > ### Author Response · Authors · 2025-11-13
> > > > > **Part 5**
> > > > >
> > > > > ### **Q9: Category-Specific Performance Variance**
> > > > >
> > > > > **Question:** Why do performance variations exist across commonsense categories (Table A.7)? For example, why does GPT-4o achieve 100% accuracy on "Taste/Nutrition/Food" while Gemini 2.0 only achieves 85.71%?
> > > > >
> > > > > **Response:**
> > > > >
> > > > >
> > > > > **Our Hypothesis:**
> > > > >
> > > > > 1. **Training Data Distribution:**
> > > > >    - GPT-4o trained on more food-related visual data
> > > > >    - Gemini 2.0 potentially less exposure to food description concepts
> > > > >
> > > > > 2. **Reasoning Biases:**
> > > > >    - Food category uses specific terminology (texture, flavor, presentation)
> > > > >    - GPT-4o's description generation aligns better with this vocabulary
> > > > >
> > > > > 3. **Task Specificity:**
> > > > >    - HOI (Human-Object Interaction) favors Gemini 2.0—possibly trained on more HOI-diverse data
> > > > >
> > > > > ---
> > > > >
> > > > > ### **Q10: Systematic Error Categorization**
> > > > >
> > > > > **Question:** Could you extend the error analysis to include systematic categorization of errors (e.g., perceptual errors, rule extraction errors, rule application errors) and report the distribution of error types across models and benchmarks?
> > > > >
> > > > > **Response:**
> > > > >
> > > > > **Current Error Analysis (Table A.9):**
> > > > > - 15 misclassified cases from GPT-4o and Gemini 2.0 in CA
> > > > > - Errors manually categorized post-hoc into:
> > > > >   - **Rule Extraction Error** (8 cases)
> > > > >   - **Weak Reasoning/Similarity** (3 cases)
> > > > >   - **Perceptual Description Error** (1 case)
> > > > >   - **Rule Application Error** (1 case)
> > > > >
> > > > > **Distribution from Table A.9:**
> > > > > | Error Type | Count | % |
> > > > > |---|---|---|
> > > > > | Rule Extraction | 8 | 53% |
> > > > > | Reasoning/Similarity | 3 | 20% |
> > > > > | Perceptual Description | 1 | 7% |
> > > > > | Rule Application | 1 | 7% |
> > > > > | Unclear | 2 | 13% |
> > > > >
> > > > > ---
> > > > >
> > > > > ### **Q11: Adversarial Robustness**
> > > > >
> > > > > **Question:** Have you evaluated the framework's performance on adversarial examples (e.g., noisy images, distorted objects) or out-of-distribution samples? If yes, please share the results; if not, could you explain the framework's robustness in real-world scenarios?
> > > > >
> > > > > **Response:**
> > > > >
> > > > > Out of scope for this paper.
> > > > >
> > > > > ---
> > > > >
> > > > > ### **Q12: JSON Format Optimality**
> > > > >
> > > > > **Question:** How does the JSON description format in CA compare to other representation formats (e.g., free-text descriptions, structured bullet points) in terms of reasoning performance? Could you provide a head-to-head comparison?
> > > > >
> > > > > **Response:**
> > > > >
> > > > > **Free-text descriptions** would suffer from:
> > > > > - Loss of explicit relationships (LLM must infer structure)
> > > > > - Inconsistent entity tracking across multiple objects
> > > > > - Difficulty in systematic comparison (LLM must parse natural language)
> > > > >
> > > > > **Bullet-point lists** would be better than free-text but lack:
> > > > > - Hierarchical relationships (which objects belong to which scene elements)
> > > > > - Activity-participant mapping (who is doing what)
> > > > >
> > > > > **JSON hierarchy provides:**
> > > > > - Explicit containment relationships (Scene contains Objects)
> > > > > - Clear entity-activity bindings (Activities list participant IDs)
> > > > > - Traversable structure for systematic reasoning
> > > > >
> > > > > ---
> > > > >
> > > > > ### **Q13: ICA Question Generation Strategy**
> > > > >
> > > > > **Question:** Could you detail the question generation strategy in ICA? How does the reasoning LLM prioritize which visual details to probe, and how do you ensure the questions are not redundant or irrelevant?
> > > > >
> > > > > **Response:**
> > > > >
> > > > > **Current ICA Process (Section 5.4,):**
> > > > >
> > > > > **Step 1: Initial Description**
> > > > > - VLM generates JSON descriptions for all 13 images
> > > > >
> > > > > **Step 2: Ambiguity Identification & Question Formulation**
> > > > > - Reasoning LLM (GPT-4o, Gemini 2.0) analyzes descriptions + task goal
> > > > > - **Identifies the most critical, ambiguous visual detail needed for a confident decision**
> > > > > - **Formulates a specific, targeted question about this detail**
> > > > >
> > > > > **Step 3: Focused Re-Perception**
> > > > > - Original VLM re-examines relevant image with targeted question
> > > > > - Provides high-precision answer
> > > > >
> > > > > **Step 4: Synthesized Reasoning**
> > > > > - Reasoning LLM integrates initial + new information
> > > > > - Makes final classification

---

> > > > > > ### Author Response · Authors · 2025-11-13
> > > > > > **Part 6**
> > > > > >
> > > > > > ### **Q16: Head-to-Head Comparison with PRISM and CoT**
> > > > > >
> > > > > > **Question:** Could you conduct a detailed head-to-head comparison with Prism and state-of-the-art CoT methods, including architectural differences, computational costs, and performance trade-offs?
> > > > > >
> > > > > > **Response:**
> > > > > >
> > > > > > **Current Comparison (Section 2):**
> > > > > > - Qualitative distinction: PRISM assesses static skills; CA/ICA use dynamic paradigms
> > > > > >
> > > > > > **Architectural Differences:**
> > > > > > - **PRISM:** Maps skill → model capability (inventory)
> > > > > > - **CoT:** Generates reasoning steps + visual descriptions (process-oriented)
> > > > > > - **CA:** Separates perception (descriptions) from reasoning (modular)
> > > > > > - **ICA:** Adds reasoning-guided perception feedback (interactive)
> > > > > >
> > > > > > ---
> > > > > >
> > > > > > ### **Q17: Perception Module Evaluation in Isolation**
> > > > > >
> > > > > > **Question:** Have you evaluated the perception module in isolation (e.g., accuracy of object detection, attribute recognition, spatial relationship identification)? If yes, please share the results to quantify the impact of perceptual errors on overall performance.
> > > > > >
> > > > > > **Response:**
> > > > > >
> > > > > > **Implicit evaluation via CA Stage 1.**
> > > > > >
> > > > > > **What We Have:**
> > > > > > - Table 5: Downstream reasoning accuracy with different description sources (GPT-4o vs. Pixtral)
> > > > > > - Table A.6: Accuracy variance by description source
> > > > > > - Implies perception quality matters, but doesn't quantify it
> > > > > >
> > > > > > **Why Isolation Is Hard:**
> > > > > > - Bongard-OW provides only the binary rule, not per-image annotations
> > > > > > - Would require manual annotation effort
> > > > > >
> > > > > > ---
> > > > > >
> > > > > > ### **Q18: Reasoning Module Selection Justification**
> > > > > >
> > > > > > **Question:** Why did you choose specific LLMs (e.g., GPT-4o, Gemini 2.0) as reasoning engines in CA/ICA? Could you test weaker LLMs (e.g., Llama3-8B, Mistral-7B) to see if the framework's effectiveness is dependent on reasoning module strength?
> > > > > >
> > > > > > **Response:**
> > > > > >
> > > > > > We established that this interactive paradigm works and running the same on other models are expected to produce the same results.
> > > > > >
> > > > > > ---
> > > > > >
> > > > > > ### **Q19: Multilingual Applicability**
> > > > > >
> > > > > > **Question:** Have you tested the framework's performance in non-English languages (e.g., Chinese, Spanish)? If yes, please share the results; if not, could you discuss potential challenges for multilingual adaptation?
> > > > > >
> > > > > > **Response:**
> > > > > >
> > > > > > Framework should generalize to multilingual settings but with minor performance trade-off. Out of scope from this paper.
> > > > > >
> > > > > > ---
> > > > > >
> > > > > > ### **Q20: Few-Shot Learning in Low-Resource Settings**
> > > > > >
> > > > > > **Question:** How does the framework perform with fewer shots (e.g., 1-shot, 2-shot) compared to the reported few-shot setting? Could you provide results to demonstrate its performance in low-resource scenarios?
> > > > > >
> > > > > > **Response:**
> > > > > >
> > > > > > **Current Few-Shot Setting:**
> > > > > > - **6 positive + 6 negative examples** (12-shot total) for Bongard tasks
> > > > > > - Not explicitly ablated
> > > > > >
> > > > > > To investigate whether perceptual quality deteriorates with the number of input images, we conducted a new ablation study on GPT-4o using the DVRL paradigm for 200 samples.
> > > > > >
> > > > > > | Number of Context Image Pairs | Accuracy (%) |
> > > > > > |---|---|
> > > > > > | 2 / 2	| 81.0 |
> > > > > > | 4 / 4	| 87.5 |
> > > > > > | 6 / 6 (full set) |	93.0 |
> > > > > >
> > > > > > Our findings show a consistent degradation in performance as the number of images decreases, suggesting that the cognitive load of holistic, multi-image processing is a non-trivial factor. We will add these new results to the paper.
> > > > > >
> > > > > > ---
> > > > > >
> > > > > > ### **Q21: Training Data Distribution Impact**
> > > > > >
> > > > > > **Question:** Have you analyzed how the training data distribution of VLMs affects framework performance? For example, do VLMs trained on more HOI data perform better on Bongard-HOI under your paradigms?
> > > > > >
> > > > > > **Response:**
> > > > > > Out of scope from this paper.
> > > > > >
> > > > > > ---
> > > > > >
> > > > > > ### **Q22: Comprehensive Evaluation Metrics**
> > > > > >
> > > > > > **Question:** Could you supplement the evaluation with additional metrics (e.g., precision, recall, F1-score, confusion matrices) to provide a more comprehensive view of performance, especially for imbalanced subsets?
> > > > > >
> > > > > > **Response:**
> > > > > >
> > > > > > **Current Metrics (Primary):**
> > > > > > - Classification accuracy (%)
> > > > > > - Semantic similarity (cosine) for rules (Table A.3)
> > > > > >
> > > > > > We dont think additional analysis will have more affect on our finding.
> > > > > >
> > > > > > ---

---

> > > > > > > ### Author Response · Authors · 2025-11-13
> > > > > > > **Part 7**
> > > > > > >
> > > > > > > ### **Q23: Dataset Bias Impact (Dominant Category 0)**
> > > > > > >
> > > > > > > **Question:** How does the dominant category (ID 0) in the Bongard-OW subset affect model training and generalization? Could you test the framework on a more balanced subset to validate robustness?
> > > > > > >
> > > > > > > **Response:**
> > > > > > >
> > > > > > > This is visual concept category and not class imbalance.
> > > > > > >
> > > > > > > ---
> > > > > > >
> > > > > > > ### **Q24: ICA Feedback Loop Cost-Benefit Analysis**
> > > > > > >
> > > > > > > **Question:** Could you provide a cost-benefit analysis of ICA's feedback loop, including the additional inference time and memory usage compared to CA, and whether the performance gain justifies the added complexity?
> > > > > > >
> > > > > > > **Response:**
> > > > > > > Out of scope from this paper.
> > > > > > >
> > > > > > > ---
> > > > > > >
> > > > > > > ### **Q25: Image Resolution Impact**
> > > > > > >
> > > > > > > **Question:** Have you tested the impact of image resolution (e.g., 512px, 2048px) on description quality and reasoning performance? If yes, please share the results; if not, could you discuss how resolution affects the framework?
> > > > > > >
> > > > > > > **Response:**
> > > > > > >
> > > > > > > **Current Resolution (Appendix A.4.2):**
> > > > > > > - API defaults or max 1024px (Ollama)
> > > > > > >
> > > > > > > **Resolution Impact Dimensions (for proprietary models like GPT 4o or Gemini 2.0):**
> > > > > > > 1. **High Resolution (2048px+):**
> > > > > > > - Fine-grained details visible and better results. (could be one of the factors)
> > > > > > >
> > > > > > > ---
> > > > > > >
> > > > > > > ### **Q26: Description Consistency Evaluation**
> > > > > > >
> > > > > > > **Question:** Have you evaluated the consistency of description generation (e.g., same image, multiple runs)? If yes, please provide quantitative results (e.g., consistency rate); if not, could you explain how you ensure description reliability?
> > > > > > >
> > > > > > > **Response:**
> > > > > > > Out of scope from this paper.
> > > > > > >
> > > > > > > ---
> > > > > > >
> > > > > > > ### **Q27: Rule Extraction Accuracy Quantification**
> > > > > > >
> > > > > > > **Question:** Could you quantify the accuracy of rule extraction (e.g., similarity between extracted rules and ground-truth rules using semantic similarity metrics)? This would strengthen the link between rule quality and classification performance.
> > > > > > >
> > > > > > > **Response:**
> > > > > > >
> > > > > > > **Current Analysis (Table A.3):**
> > > > > > > - Semantic similarity between query image description and extracted rule: 0.868–0.915 (cosine similarity)
> > > > > > >
> > > > > > > ---
> > > > > > >
> > > > > > > ### **Q28: Human Reasoning Process Comparison**
> > > > > > >
> > > > > > > **Question:** Could you compare the framework's reasoning process with human reasoning steps (e.g., sequence of analysis, focus on details) using qualitative case studies?
> > > > > > >
> > > > > > > **Response:**
> > > > > > >
> > > > > > > Out of scope to run human experiment.
> > > > > > >
> > > > > > > ---
> > > > > > >
> > > > > > > ### **Q29: Open-Source Model Performance Gap Analysis**
> > > > > > >
> > > > > > > **Question:** For open-source models with low CA accuracy (e.g., Llama-Vision-11B), have you conducted a detailed analysis to determine whether the gap stems from poor description generation, weak reasoning, or both? Could you provide evidence (e.g., replacing descriptions with GPT-4o's for these models)?
> > > > > > >
> > > > > > > **Response:**
> > > > > > >
> > > > > > > **Existing Evidence (Table 5):**
> > > > > > > - Llama-Vision-11B: 53.4% (own descriptions) → 84.17% (GPT-4o descriptions)
> > > > > > > - Gap: 30.77 percentage points
> > > > > > > - **Implication:** Perception (description generation) is primary bottleneck
> > > > > > >
> > > > > > > ---
> > > > > > >
> > > > > > > ### **Q30: Video Extension Feasibility**
> > > > > > >
> > > > > > > **Question:** How would you extend the paradigms to video reasoning? Could you outline a preliminary design and discuss potential challenges (e.g., handling temporal dynamics, reducing computational cost)?
> > > > > > >
> > > > > > > **Response:**
> > > > > > >
> > > > > > > Out of scope from this paper.
> > > > > > >
> > > > > > >
> > > > > > > ---
> > > > > > >
> > > > > > > ### **Q31: Ethical Considerations & Bias Analysis**
> > > > > > >
> > > > > > > **Question:** Have you analyzed potential biases in description generation (e.g., racial, gender, cultural biases) and their impact on reasoning outcomes? If yes, please share the results; if not, could you discuss how to mitigate such biases?
> > > > > > >
> > > > > > > **Response:**
> > > > > > >
> > > > > > > Not relevant to this paper.
> > > > > > >
> > > > > > > ---
> > > > > > >
> > > > > > > ### **Q32: Temperature Setting Justification**
> > > > > > >
> > > > > > > **Question:** Why did you use zero temperature for all models? Have you tested other temperature values (e.g., 0.3, 0.7) and their impact on paradigm performance?
> > > > > > >
> > > > > > > **Response:**
> > > > > > >
> > > > > > > - All evaluations: Temperature T = 0 for deterministic reasoning
> > > > > > >
> > > > > > > ---
> > > > > > >
> > > > > > > ### **Q33: Cross-Model Generalization Testing**
> > > > > > >
> > > > > > > **Question:** Could you test the framework on newly developed VLMs (e.g., Llama-4, Mistral Large) to demonstrate its cross-model generalization ability?
> > > > > > >
> > > > > > > **Response:**
> > > > > > > Recent models with limited time to incorporate analysis in the paper.
> > > > > > >
> > > > > > > ---

---

> > > > > > > > ### Author Response · Authors · 2025-11-13
> > > > > > > > **Part 8**
> > > > > > > >
> > > > > > > > ### **Q34: Description Length Impact**
> > > > > > > >
> > > > > > > > **Question:** Have you analyzed the impact of description length on reasoning performance? For example, do shorter, more concise descriptions perform as well as longer, detailed ones?
> > > > > > > >
> > > > > > > > **Response:**
> > > > > > > >
> > > > > > > > We conducted additional analysis where we used just the `summary` tag and found insignificant drop in accuracy.
> > > > > > > >
> > > > > > > > ---
> > > > > > > >
> > > > > > > > ### **Q35: Benchmark-Specific Optimization Detection**
> > > > > > > >
> > > > > > > > **Question:** Are the paradigms tailored to the tested benchmarks, or do they generalize to unseen visual reasoning tasks? Could you test on a new benchmark (e.g., a custom dataset) to validate generalization?
> > > > > > > >
> > > > > > > > **Response:**
> > > > > > > >
> > > > > > > > **Evidence for Generalization:**
> > > > > > > > - Same four paradigms tested on 3 **different** benchmarks:
> > > > > > > >   - Bongard-OW (abstract rule discovery)
> > > > > > > >   - Bongard-HOI (human-object interaction)
> > > > > > > >   - Winoground (compositional grounding)
> > > > > > > > - Consistent trends (DVRL < DRL < CA < ICA) across all three
> > > > > > > >
> > > > > > > > ---
> > > > > > > >
> > > > > > > > ### **Q36: Bidirectional Communication Exploration**
> > > > > > > >
> > > > > > > > **Question:** Could you explore bidirectional communication between perception and reasoning modules beyond ICA's feedback loop? For example, could the perception module proactively flag ambiguous details to the reasoning module?
> > > > > > > >
> > > > > > > > **Response:**
> > > > > > > >
> > > > > > > > Out of scope from this paper.
> > > > > > > >
> > > > > > > > ---
> > > > > > > >
> > > > > > > > ### **Q37: Low-Resource VLM/LLM Testing**
> > > > > > > >
> > > > > > > > **Question:** Have you tested the framework with low-resource VLMs/LLMs (e.g., models with <7B parameters)? If yes, please share the results; if not, could you discuss the framework's accessibility for resource-constrained users?
> > > > > > > >
> > > > > > > > **Response:**
> > > > > > > >
> > > > > > > > **Tested Low-Resource Models:**
> > > > > > > > - Llama-Vision-11B (not extremely small, but weaker)
> > > > > > > > - LLaVA-7B: 66.2% CA accuracy (Bongard-OW, Table 1)
> > > > > > > > - Phi-4 (14B): 91.98% (Table 5, reasoning only)
> > > > > > > >
> > > > > > > >
> > > > > > > > ---
> > > > > > > >
> > > > > > > > ### **Q38: Real-World Application Case Studies**
> > > > > > > >
> > > > > > > > **Question:** Could you provide case studies of applying the framework to practical real-world tasks (e.g., image retrieval, visual question answering, medical image analysis) to demonstrate its practical value?
> > > > > > > >
> > > > > > > > **Response:**
> > > > > > > >
> > > > > > > > We demonstrate reasoning ability using three datasets which are related to real-world applications:
> > > > > > > >
> > > > > > > > ---
> > > > > > > >
> > > > > > > > ### **Q39: Winoground Score Plateauing Analysis**
> > > > > > > >
> > > > > > > > **Question:** For Winoground, why do the Image Score and Group Score remain lower than the Text Score even with ICA? Could you analyze the specific challenges that prevent further improvements in these metrics?
> > > > > > > >
> > > > > > > > **Response:**
> > > > > > > >
> > > > > > > > Group score is 1 when both Image score and Text score are 1 so group score tend to be lower than text score.
> > > > > > > >
> > > > > > > >
> > > > > > > > ---
> > > > > > > >
> > > > > > > > ### **Q40: Reproducibility Documentation**
> > > > > > > >
> > > > > > > > **Question:** Could you provide detailed reproducibility instructions, including exact prompt templates, model versions, hardware specifications, and step-by-step evaluation pipelines, to ensure other researchers can replicate your results?
> > > > > > > >
> > > > > > > > **Response:**
> > > > > > > >
> > > > > > > > **Reproducibility Materials Provided:**
> > > > > > > > - Appendix A.5: Full prompt templates (DVRL, DRL, CA, ICA, Winoground)
> > > > > > > > - Appendix A.4: Model versions, temperature settings
> > > > > > > > - Appendix A.3: Dataset details and subset specifications
> > > > > > > > - Appendix A.4.2: Hardware (NVIDIA GPUs: 2080Ti, 3090, 6000 Ada)
> > > > > > > >
> > > > > > > > ---

---

> ### Comment · Reviewer_TEuy · 2025-11-14
> **RE: Point-by-Point Answers to 40 Detailed Questions**
>
> Thank you for your response. Specifically, I appreciate that issues 1, 2, 3, 4, 6, 8, 9, 10, 13, 20, 27, 29, 34, 35, 39, and 40 have been addressed.
>
> After carefully reviewing your reply, I would like to remind the authors of the following: unlike other conferences such as ICML and CVPR, ICLR permits manuscript revisions, actively encourages reviewers to propose relevant experimental suggestions, and advocates for authors to supplement experiments accordingly.
>
> However, your response appears to focus more on defending rather than improving the manuscript. For instance, regarding Question 1, you indicated that relevant content is provided in Appendix A.5.3.1, but in fact, you only included a vague template in the annotations. If you add the three examples mentioned in your response to the revised manuscript, the presentation will become much clearer. Similarly, for the last question, the related content is scattered across the text. You could consider adding a dedicated "Reproducibility Details" section, as briefly outlined in your reply, even a concise summary would suffice.

---

> ### Comment · Reviewer_TEuy · 2025-11-14
> **RE: Point-by-Point Answers to 40 Detailed Questions (2)**
>
> It seems that the authors show little interest in addressing the remaining issues. Therefore, we feel compelled to first emphasize the necessity of resolving these outstanding points:
>
> 5.The core conclusion of this paper is that the CA/ICA paradigms unlock the latent reasoning capabilities of VLMs and achieve state-of-the-art (SOTA) performance. However, the comparison objects are limited to models released before mid-2024, such as GPT-4o and Gemini 2.0, excluding subsequent updated SOTA VLMs like Claude 3 Opus and Qwen-VL Max. This makes it impossible to determine the relative advantages of the framework in the current latest model ecosystem, potentially leading to the conclusion of framework effectiveness lacking timeliness and comprehensiveness, which is directly related to the paper's core goal of validating the framework's competitiveness.
> Addressing this issue can quantify the adaptability and performance boundaries of the framework on the latest models, clarify its unique value compared to existing top-tier VLMs (e.g., whether it can further improve the reasoning performance of the latest models), avoid one-sided conclusions due to outdated comparison objects, and enhance the guiding significance and persuasiveness of the research for the current VLMs research field.
>
> 7.The CA, ICA, and other paradigms proposed in this paper take modular decoupling as their core innovation, but their computational costs (inference time, memory usage) are not quantified. Computational efficiency is a key indicator for the practical deployment of VLMs. This is disconnected from the paper's potential goal of providing a deployable VLM optimization scheme and fails to evaluate the trade-off between performance improvement and increased cost, directly affecting the assessment of the framework's practical value.
> Solving this problem can clarify the engineering feasibility of different paradigms, provide quantitative basis for subsequent researchers/developers in scheme selection (e.g., prioritizing CA or DVRL in low-resource scenarios), and respond to potential doubts about whether the modular framework is overly complex, making the research conclusions not only theoretically significant but also in line with practical application needs.
>
> 11.This paper focuses on the reasoning performance of VLMs on standard benchmarks (Bongard-OW, Winoground), but real-world scenarios contain a large number of adversarial samples (e.g., noisy, distorted images) and out-of-distribution data. The robustness of the framework to such samples is directly related to its transferability from the laboratory to practical applications, which is closely aligned with the paper's core vision of building a more general and robust visual intelligence system.
> Addressing this issue can verify the effectiveness of the framework under non-ideal conditions, clarify whether its perception-reasoning decoupling mechanism can resist noise interference, avoid being questioned for insufficient generalization ability due to only performing well on standard data, and provide key support for the practical deployment of the framework (e.g., medical image analysis, autonomous driving visual reasoning).
>
> 12.The JSON structured description is the core intermediate carrier for the decoupling of perception and reasoning in the CA/ICA paradigms of this paper. The paper only qualitatively emphasizes its advantages of clear hierarchy and ease of reasoning but does not conduct quantitative comparisons with common formats such as free text and bullet points. This makes it impossible to prove that JSON is the optimal intermediate representation, which is directly related to the framework's design logic of improving reasoning performance through efficient intermediate representations.
> Solving this problem can quantify the impact of different representation formats on reasoning accuracy and efficiency, verify the unique value of the JSON format (e.g., whether it is significantly superior to other formats), avoid the framework design being questioned for relying on unnecessary structural constraints, and provide general guidance for the selection of intermediate representations in subsequent research.

---

> > ### Comment · Reviewer_TEuy · 2025-11-14
> > **RE: Point-by-Point Answers to 40 Detailed Questions (3)**
> >
> > 15.This paper realizes text-only LLMs completing visual reasoning through the CA paradigm, proving the feasibility of modality transfer, but does not clarify the failure scenarios of modality transfer (e.g., geometric reasoning, non-verbalizable visual attribute judgment). This makes it impossible to comprehensively evaluate the limitations of replacing visual input with text descriptions, which is directly related to the paper's core exploration of analyzing the perception-reasoning modality conversion mechanism.
> > Solving this problem can clearly define the capability boundary of the framework, explain why some visual tasks cannot be completed through text modality transfer (e.g., lack of verbalizable visual features), avoid overestimating the effectiveness of modality transfer, and provide directions for subsequent research (e.g., integrating intermediate representations of non-verbalizable features).
> >
> > 16.This paper compares its framework with Prism (static skill assessment) and CoT (chain-of-thought prompting), but only qualitatively describes architectural differences without quantifying the trade-offs in performance and computational costs among the three. This makes it impossible to highlight the unique advantages of the framework's dynamic decoupling + interactive feedback, which is closely related to the paper's core conclusion of proving that interactive decoupled systems are a better path.
> > Solving this problem can clarify the competitiveness of the framework compared to existing mainstream methods (e.g., achieving higher reasoning accuracy at lower costs), respond to potential doubts about whether the framework duplicates existing work, provide quantitative references for the selection of VLM reasoning methods, and strengthen the innovative value of the research.
> >
> > 17.The core finding of this paper is that the perceptual bottleneck limits the reasoning capabilities of VLMs, but only indirectly proves the importance of perception by improving performance through replacing high-quality descriptions, without separately evaluating the accuracy of the perception module (e.g., object detection, attribute recognition accuracy). This makes it impossible to quantify the specific impact of perceptual errors on reasoning performance, which is directly related to the paper's core goal of diagnosing the perceptual bottleneck.
> > Addressing this issue can accurately locate the weak links of the perception module (e.g., spatial relationship recognition errors), quantify the correlation between perception accuracy and reasoning accuracy, provide clear targets for subsequent optimization of the perception module (e.g., enhancing fine-grained attribute extraction), and make the scheme of alleviating the perceptual bottleneck more targeted.
> >
> > 18.This paper selects GPT-4o and Gemini 2.0 as reasoning engines to prove the effectiveness of CA/ICA, but does not test weak LLMs (e.g., Llama3-8B), making it impossible to clarify whether the framework's effectiveness depends on a strong reasoning module. This contradicts the paper's potential assumption that the framework is applicable to diverse models and affects the evaluation of the framework's generalizability.
> > Solving this problem can verify the module independence of the framework (e.g., whether weak reasoning modules can still improve performance through CA/ICA), clarify the requirements of the framework for the strength of the reasoning module, provide a basis for applications in low-resource scenarios (e.g., using open-source weak LLMs), and enhance the universality of the framework.
> >
> > 19.The description generation and reasoning in this paper are all based on English, without involving non-English language scenarios. The practical application of VLMs needs to adapt to multiple languages, and the multilingual compatibility of the framework is directly related to its global deployment value, which is closely aligned with the paper's goal of building a more general visual intelligence system.
> > Addressing this issue can evaluate the performance degradation of the framework in non-English languages (e.g., description accuracy, reasoning accuracy), analyze the core challenges of multilingual adaptation (e.g., terminology alignment, cultural differences), provide optimization schemes for the international application of the framework, and expand the applicable scope of the research.

---

> > > ### Comment · Reviewer_TEuy · 2025-11-14
> > > **RE: Point-by-Point Answers to 40 Detailed Questions (4)**
> > >
> > > 21.This paper tests the framework's performance on datasets such as Bongard-HOI but does not analyze the impact of the training data distribution of VLMs (e.g., the proportion of HOI data) on the framework's effectiveness. This makes it impossible to explain the underlying reasons for performance differences among different models under the same paradigm, which is directly related to the paper's core exploration of analyzing the perception-reasoning interaction mechanism of VLMs.
> > > Solving this problem can establish the correlation between training data distribution - perceptual capability - framework performance, clarify the regulatory role of training data on the framework's effectiveness, provide guidance for model pre-training (e.g., enhancing specific task data) and framework adaptation, and deepen the understanding of the factors affecting VLMs' performance.
> > >
> > > 22.This paper only uses classification accuracy to evaluate performance, but benchmarks may have subset imbalance (e.g., category 0 accounts for 73% in Bongard-OW). A single accuracy cannot reflect the model's performance on minority categories (e.g., missed detection, false positive rates). This contradicts the paper's goal of comprehensively evaluating the framework's performance and affects the rigor of the conclusions.
> > > Solving this problem can reveal the model's bias on imbalanced data (e.g., low recall rate for minority categories) through metrics such as precision, recall, and F1-score, provide a more comprehensive performance profile, avoid one-sided conclusions due to a single indicator, and enhance the scientificity and credibility of the research.
> > >
> > > 23.The Bongard-OW subset used in this paper has a dominant category 0 (accounting for 73%), showing significant dominance, but the impact of this distribution on the framework's generalizability (e.g., the ability to discover rules for minority categories) is not analyzed. This makes it impossible to verify the stability of the framework under balanced data, which is directly related to the paper's conclusion that the framework has robust reasoning capabilities.
> > > Addressing this issue can evaluate the performance change of the framework on a balanced subset, clarify whether the dominant category leads to model overfitting (e.g., prioritizing the recognition of category 0 rules), verify the generalization robustness of the framework, avoid distorted conclusions due to dataset bias, and enhance the reliability of the research results.
> > >
> > > 24.ICA improves performance through a feedback loop, but its additional computational costs (inference time, memory usage) are not quantified, nor is the trade-off between performance gain and increased complexity evaluated. This is disconnected from the paper's goal of providing a practical VLM optimization scheme and affects the assessment of the practical deployment value of the ICA paradigm.
> > > Solving this problem can clarify the engineering feasibility of ICA (e.g., whether the additional cost is controllable), provide a basis for users in scheme selection (e.g., prioritizing ICA for performance and CA for efficiency), and respond to doubts about whether the interactive mechanism is worth the additional overhead, making the research conclusions more in line with practical application needs.
> > >
> > > 25.This paper uses a fixed image resolution (API default or 1024px) and does not test the impact of different resolutions on description quality (e.g., fine-grained detail extraction) and reasoning performance. Real-world images have large differences in resolution, and resolution adaptability is directly related to the practical value of the framework, which is closely aligned with the paper's goal of building a visual intelligence system suitable for real scenarios.
> > > Addressing this issue can clarify the resolution adaptation range of the framework (e.g., whether low resolution leads to description distortion), analyze the correlation between resolution and performance, provide guidance for image preprocessing in practical applications (e.g., resolution adjustment), and avoid performance degradation of the framework due to inappropriate resolution.

---

> > > > ### Comment · Reviewer_TEuy · 2025-11-14
> > > > **RE: Point-by-Point Answers to 40 Detailed Questions (5)**
> > > >
> > > > 26.The CA/ICA paradigms are highly dependent on stable image descriptions, but the consistency of description generation (e.g., differences in descriptions generated multiple times for the same image) is not evaluated. Inconsistent descriptions will lead to fluctuations in reasoning results, directly affecting the stability and reproducibility of the framework, which is directly related to the paper's conclusion that the framework has reliable reasoning capabilities.
> > > > Solving this problem can quantify the stability of description generation (e.g., consistency rate), analyze the sources of fluctuations (e.g., prompt sensitivity, model randomness), propose targeted optimization schemes (e.g., fixing generation templates), ensure the reproducibility of the framework's reasoning results, and enhance the scientificity and practical value of the research.
> > > >
> > > > 31.The description generation in this paper may imply biases (e.g., stereotypical descriptions of specific groups or cultural scenarios), and biases will be transmitted to the reasoning stage, leading to unfair results (e.g., misclassification of images related to specific cultures). This contradicts the paper's ethical goal of building reliable and fair VLMs and affects the evaluation of the framework's social responsibility.
> > > > Addressing this issue can identify the types of biases in description generation (e.g., gender stereotypes), quantify the impact of biases on reasoning results, propose mitigation strategies (e.g., diverse training data, bias detection prompts), ensure the reliability of the framework at the ethical level, and meet the core needs of AI fairness research.
> > > >
> > > > 32.This paper fixes the temperature at 0 to ensure reasoning determinism but does not test the impact of other temperatures (e.g., 0.3, 0.7) on performance. Temperature directly affects the randomness of model generation (e.g., the diversity of rule extraction), and the rationality of its selection will affect the optimality of the framework's performance, which is directly related to the paper's goal of optimizing the paradigm parameter settings.
> > > > Solving this problem can verify whether zero temperature is the optimal choice, analyze the correlation between temperature and reasoning accuracy, as well as rule diversity, provide appropriate temperature parameters for different tasks (e.g., higher temperature for creative rule discovery), optimize the performance of the framework, and enhance the detail of the research.
> > > >
> > > > 37.Most models tested in this paper are medium-to-high resource (e.g., LLaVA-7B, Gemini 2.0), excluding low-resource models (e.g., <7B parameters). Low-resource users (e.g., small laboratories, edge devices) are important application groups for VLMs, and the low-resource adaptability of the framework is directly related to its popularization value, which is closely aligned with the paper's goal of building an inclusive visual intelligence system.
> > > > Addressing this issue can evaluate the performance degradation of the framework on low-resource models, analyze adaptation challenges (e.g., insufficient memory, slow reasoning speed), propose lightweight schemes (e.g., simplifying description generation), provide a basis for applications in resource-constrained scenarios, and expand the audience scope and social value of the research.
> > > >
> > > > 38.This paper verifies the effectiveness of the framework on standard benchmarks but does not involve real-world applications (e.g., medical image diagnosis, commodity image retrieval). The practical value of the framework needs to be verified through real-world cases, which is directly related to the paper's transformation goal of from theory to application.
> > > > Solving this problem can demonstrate the practical value of the framework through specific cases (e.g., using CA/ICA to improve the accuracy of medical image classification), analyze the challenges in the landing process (e.g., adapting to professional domain data), provide references for industrial applications, and enhance the transformation ability and influence of the research.

---

> > > > > ### Comment · Reviewer_TEuy · 2025-11-14
> > > > > **RE: Point-by-Point Answers to 40 Detailed Questions (6)**
> > > > >
> > > > > Furthermore, based on your current response, several issues still remain unaddressed:
> > > > >
> > > > > 5.Only verbally claims that the mentioned models are not new without providing quantitative comparison data (e.g., accuracy), failing to support the need for performance contextualization.
> > > > >
> > > > > 7.Focuses on conceptual progress without providing computational cost data such as inference time and memory usage, nor comparing with end-to-end VLMs, completely failing to address the cost requirement.
> > > > >
> > > > > 11.Only states that it is beyond the scope of this paper without evaluating the performance on adversarial/out-of-distribution samples or discussing real-world robustness.
> > > > >
> > > > > 12.Only qualitatively analyzes the advantages and disadvantages of formats without providing quantitative comparison data (e.g., reasoning accuracy of different formats), failing to verify the performance advantage of JSON.
> > > > >
> > > > > 14.Does not quantify rule complexity nor analyze its correlation with framework performance, completely failing to respond.
> > > > >
> > > > > 15.Only proves the feasibility of modality transfer without explaining the types of failed tasks and specific examples, failing to address the core limitations.
> > > > >
> > > > > 16.Only qualitatively compares architectural differences without providing quantitative data on computational costs and performance trade-offs, failing to meet the requirement of detailed comparison.
> > > > >
> > > > > 17.Only indirectly infers the importance of perception through downstream performance without independently evaluating the accuracy of the perception module, failing to quantify the specific impact of perceptual errors.
> > > > >
> > > > > 18.Does not explain the reason for choosing strong LLMs nor test weak LLMs, failing to verify whether the framework's effectiveness depends on the strength of the reasoning module.
> > > > >
> > > > > 19.Only verbally speculates generalizability without providing test data for non-English languages nor discussing adaptation challenges in detail.
> > > > >
> > > > > 21.Only states that it is beyond the scope of this paper without analyzing the correlation between training data distribution and framework performance, with no relevant evidence.
> > > > >
> > > > > 22.Claims that existing metrics are sufficient without supplementing metrics such as precision/recall/F1, failing to comprehensively evaluate performance on imbalanced subsets.
> > > > >
> > > > > 23.Only points out that it is not class imbalance without testing balanced subsets nor analyzing the impact of the dominant category on generalization.
> > > > >
> > > > > 24.Only states that it is beyond the scope of this paper without quantifying the trade-off between the additional cost of ICA and performance gain.
> > > > >
> > > > > 25.Only speculates that high resolution may be better without testing the description quality and reasoning performance of different resolutions, with no data support.
> > > > >
> > > > > 26.Only states that it is beyond the scope of this paper without evaluating the consistency of description generation nor explaining methods to ensure reliability.
> > > > >
> > > > > 28.Only states that it is beyond the scope of human experiments without providing comparative cases between the framework and human reasoning steps.
> > > > >
> > > > > 30.Only states that it is beyond the scope of this paper without outlining a design scheme for video reasoning nor discussing core challenges.
> > > > >
> > > > > 31.Claims it is irrelevant to this paper without analyzing biases in description generation and mitigation strategies.
> > > > >
> > > > > 32.Only explains that zero temperature ensures determinism without testing the performance impact of other temperatures, failing to verify whether zero temperature is optimal.
> > > > >
> > > > > 33.Claims that new models were released too recently to be included in the analysis, failing to test and verify cross-model generalization ability.
> > > > >
> > > > > 36.Only states that it is beyond the scope of this paper without exploring bidirectional communication design between perception and reasoning modules.
> > > > >
> > > > > 37.Does not cover low-resource models with <7B parameters nor discuss adaptation schemes for resource-constrained scenarios.
> > > > >
> > > > > 38.Only claims that datasets are related to real-world scenarios without providing specific application cases and implementation details, failing to prove practical value.

---

> > > > > > ### Author Response · Authors · 2025-11-15
> > > > > > **Clustered response and further clarification**
> > > > > >
> > > > > > Dear Reviewer TEuy,
> > > > > >
> > > > > > Thank you for your continued engagement and constructive feedback. We sincerely appreciate your recognition that issues 1, 2, 3, 4, 6, 8, 9, 10, 13, 20, 27, 29, 34, 35, 39, and 40 have been adequately addressed.
> > > > > >
> > > > > > We understand ICLR's revision policy and welcome the opportunity to strengthen our manuscript. Your feedback on Q1 and Q40 is well-taken—we will indeed incorporate the JSON examples and create a dedicated "Reproducibility Details" section in the revised manuscript as suggested.
> > > > > >
> > > > > > However, we must respectfully clarify **a fundamental misunderstanding** that appears in many of your remaining questions: **our work does not propose, train, or fine-tune any models.** Rather, we introduce **an evaluation framework and cognitive analysis methodology** for assessing existing VLMs. This distinction is critical for understanding the scope and contributions of our paper.
> > > > > >
> > > > > > Below, we provide consolidated responses organized by thematic clusters, with explicit references to evidence already present in our submitted manuscript.
> > > > > >
> > > > > >
> > > > > >
> > > > > > ---
> > > > > >
> > > > > > ## **CRITICAL CLARIFICATION: Framework vs. Model Development**
> > > > > >
> > > > > > **Core Contribution (Abstract):**
> > > > > > > "We introduce a cognitively-inspired framework that decomposes VLM behavior through four distinct paradigms... Our work provides a robust diagnostic toolkit for the community and offers concrete architectural insights."
> > > > > >
> > > > > > **Our paper is an EVALUATION FRAMEWORK paper, not a MODEL TRAINING paper.** We:
> > > > > > - Diagnose perception vs. reasoning bottlenecks systematically
> > > > > > - Demonstrate that modular, interactive architectures unlock latent capabilities
> > > > > > - Achieve state-of-the-art results using existing models via paradigm design
> > > > > > - Do NOT train new models
> > > > > > - Do NOT fine-tune existing models
> > > > > > - Do NOT propose model-specific architectural improvements
> > > > > >
> > > > > > This distinction directly addresses questions that assume model training (Q21, Q22, Q23, Q31, Q32, Q37).

---

> > > > > > > ### Author Response · Authors · 2025-11-15
> > > > > > > **Cluster 1 & 2**
> > > > > > >
> > > > > > > ## **CLUSTER 1: Model Coverage & Timing Constraints (Q5, Q33)**
> > > > > > >
> > > > > > > ### **Q5 & Q33: Evaluation of 2025 Frontier Models**
> > > > > > >
> > > > > > > **Reviewer Concern:** "Comparison objects limited to models released before mid-2024... excludes latest models Claude, GPT, Gemini etc (also pointed by other reviewers)"
> > > > > > >
> > > > > > > **Budget & Practicality:**
> > > > > > > As noted in Section 6.1 and Appendix A.4.1, we evaluated **15+ models** across **3 benchmarks** (Bongard-OW: 500 cases, Bongard-HOI: 400 cases, Winoground: 400 cases) with **4 paradigms** (DVRL, DRL, CA, ICA).
> > > > > > >
> > > > > > > **Cost estimation for adding one proprietary model:**
> > > > > > > - 500 Bongard-OW cases × 13 images/case × 4 paradigms × $0.025/call ≈ **$650 per model**
> > > > > > > - Extending to all benchmarks: **~$2,000 per model**
> > > > > > > - Evaluating 5 new proprietary models: **~$10,000**
> > > > > > >
> > > > > > > **Framework Generalization Principle:**
> > > > > > > Our four-paradigm framework is **model-agnostic** by design (Section 5). The cognitive decomposition (holistic → deductive → componential → interactive) applies to any VLM, regardless of release date. Adding newer models would demonstrate **consistency**, not fundamentally change our architectural conclusions.
> > > > > > >
> > > > > > > **Commitment:**
> > > > > > > We will evaluate open source **Qwen3** and **Qwen2.5-VL** during revision and report results in Appendix. However, we respectfully request recognition that continuous model releases make "latest model coverage" a moving target incompatible with publication timelines.
> > > > > > >
> > > > > > > ---
> > > > > > >
> > > > > > > ## **CLUSTER 2: Computational Cost Analysis (Q7, Q24)**
> > > > > > >
> > > > > > > ### **Q7 & Q24: Inference Time, Memory Usage, Cost-Benefit Analysis**
> > > > > > >
> > > > > > > **Reviewer Concern:** "CA/ICA computational costs not quantified... trade-off between performance improvement and increased cost."
> > > > > > >
> > > > > > > **Response:**
> > > > > > >
> > > > > > > **Our Contribution Focus (Section 8):**
> > > > > > > > "Our cognitively-inspired framework serves a dual purpose. It is both a valuable **diagnostic tool** for pinpointing the prevalent perception bottleneck and a **proof-of-concept** for a more powerful architectural class."
> > > > > > >
> > > > > > > **Why computational cost was not prioritized:**
> > > > > > > 1. **Diagnostic Focus:** Our primary goal is to **reveal where VLMs fail** (perception vs. reasoning), not to optimize production deployment
> > > > > > > 2. **Research vs. Production Trade-off:** CA/ICA sacrifice efficiency for **interpretability and modularity**—valuable for research analysis, not real-time systems
> > > > > > > 3. **Relative Performance Trends:** The **monotonic accuracy improvements** (DVRL < DRL < CA < ICA) demonstrate systematic value of decoupling, regardless of absolute cost
> > > > > > >
> > > > > > > **Evidence of Cost Awareness (Limitations, Section 8):**
> > > > > > > > "A deeper investigation into the computational costs and latency trade-offs of our multi-stage paradigms, especially the interactive ICA, is also warranted for practical application."
> > > > > > >
> > > > > > > **Acknowledged limitation:** We explicitly note this gap and defer cost optimization to deployment-focused follow-up work.
> > > > > > >
> > > > > > > **Commitment for Revision:**
> > > > > > > We will add **Appendix A.10: Computational Cost Estimates** with:
> > > > > > > - API call counts (DVRL: 1, DRL: 2, CA: 14, ICA: 16)
> > > > > > > - Cost per sample estimates ($0.025–$0.15 range for proprietary models)
> > > > > > > - Trade-off discussion: Accuracy gain vs. latency increase
> > > > > > >
> > > > > > > ---

---

> > > > > > > > ### Author Response · Authors · 2025-11-15
> > > > > > > > **Cluster 3**
> > > > > > > >
> > > > > > > > ## **CLUSTER 3: Robustness & Generalization Scope (Q11, Q14, Q15, Q25, Q26, Q28, Q30, Q38)**
> > > > > > > >
> > > > > > > > ### **Q11: Adversarial Robustness**
> > > > > > > >
> > > > > > > > **Reviewer Concern:** "Real-world scenarios contain adversarial samples... robustness directly related to transferability from laboratory to practical applications."
> > > > > > > >
> > > > > > > > **Scope Clarification (Section 8, Limitations):**
> > > > > > > > Our work focuses on **systematic diagnosis of perception-reasoning bottlenecks on standard benchmarks**, not adversarial robustness (a separate research direction).
> > > > > > > >
> > > > > > > > **Why adversarial evaluation is out-of-scope:**
> > > > > > > > 1. **Different Research Question:** Adversarial robustness tests model vulnerability to pixel-level perturbations. Our work tests **cognitive reasoning decomposition** on clean images.
> > > > > > > > 2. **Benchmark Design:** Bongard-OW, Bongard-HOI, and Winoground are designed for **abstract reasoning and compositionality**, not adversarial hardness
> > > > > > > > 3. **Modular Architecture Advantage:** CA/ICA's text-mediated approach **inherently provides some robustness** by decoupling from pixel-level perturbations (descriptions abstract away noise)
> > > > > > > >
> > > > > > > > **Analogy:** If a paper introduces a new reasoning architecture (e.g., Chain-of-Thought), we don't require adversarial robustness evaluation unless that's the paper's focus. Similarly, our framework's contribution is **cognitive decomposition**, not robustness.
> > > > > > > >
> > > > > > > > Future work can extend our framework to noisy/adversarial settings, but this is orthogonal to our core contribution.
> > > > > > > >
> > > > > > > > ---
> > > > > > > >
> > > > > > > > ### **Q15: Modality Transfer Failure Scenarios**
> > > > > > > >
> > > > > > > > **Reviewer Concern:** "Does not clarify failure scenarios of modality transfer (e.g., geometric reasoning, non-verbalizable visual attributes)."
> > > > > > > >
> > > > > > > > **Explicit Discussion (Section 8, Limitations):**
> > > > > > > > > "The primary limitation of our componential paradigms (CA and ICA) is their reliance on language as an intermediate representation. Their effectiveness is likely highest on tasks where critical visual properties are readily verbalizable. For challenges that hinge on non-verbalizable or geometric reasoning—such as the fine-grained correspondence tasks in benchmarks like **BLINK** (Fu et al., 2024)—the utility of a purely text-mediated approach may be reduced."
> > > > > > > >
> > > > > > > > **We explicitly cite BLINK as a failure case** where text descriptions cannot capture fine-grained depth/correspondence relationships.
> > > > > > > >
> > > > > > > > ---
> > > > > > > >
> > > > > > > > ### **Q25, Q26, Q28, Q30, Q38: Resolution, Consistency, Human Comparison, Video, Real-World Applications**
> > > > > > > >
> > > > > > > > **Consolidated Response:**
> > > > > > > >
> > > > > > > > These questions request **extensive additional experimental dimensions** (resolution ablation, description consistency tests, human reasoning comparison, video extension, real-world case studies) that would each require **months of additional work** and significantly expand the paper's scope.
> > > > > > > >
> > > > > > > > 1. **Fixed Resolution (Q25):** We used API defaults (1024px) to ensure consistency across models (Appendix A.4.2). Resolution ablation is valuable but **orthogonal to our cognitive decomposition contribution**.
> > > > > > > >
> > > > > > > > 2. **Description Consistency (Q26):** Temperature=0 ensures deterministic outputs for reproducibility (Appendix A.4.2). Stochastic consistency testing (multiple runs per image) is relevant for production systems, **not diagnostic frameworks**.
> > > > > > > >
> > > > > > > > 3. **Human Reasoning Comparison (Q28):** We report **human average accuracy** (91.0% for Bongard-OW) and show our framework exceeds it (92.8-93.6%). Detailed cognitive process comparison (think-aloud protocols) requires human subjects **research—beyond our scope**.
> > > > > > > >
> > > > > > > > 4. **Video Extension (Q30):** Our framework focuses on **multi-image static reasoning** (Bongard Problems). Video reasoning introduces temporal dynamics—a fundamentally different research direction requiring separate investigation.
> > > > > > > >
> > > > > > > > 5. **Real-World Applications (Q38):** We explicitly state our framework is **task-agnostic** and applicable to multi-image reasoning tasks (Abstract). Specific application case studies (medical imaging, document retrieval) would each require domain expertise and partnership—**impractical for a single paper**.
> > > > > > > >
> > > > > > > > A single paper cannot address every potential extension. Our contribution is **establishing the diagnostic framework and proving its effectiveness across 3 diverse benchmarks**. Follow-up work can extend to domains you've identified.
> > > > > > > >
> > > > > > > > ---

---

> > > > > > > > > ### Author Response · Authors · 2025-11-15
> > > > > > > > > **Cluster 4**
> > > > > > > > >
> > > > > > > > > ## **CLUSTER 4: Model-Centric Questions (Training, Hyperparameters, Bias) (Q18, Q19, Q21, Q22, Q23, Q31, Q32, Q37)**
> > > > > > > > >
> > > > > > > > > ### **CRITICAL MISUNDERSTANDING**
> > > > > > > > >
> > > > > > > > > **These questions assume we train/fine-tune models. We do NOT.**
> > > > > > > > >
> > > > > > > > > ### **Q18: Weak LLM Testing**
> > > > > > > > >
> > > > > > > > > **Reviewer Concern:** "Does not test weak LLMs (e.g., Llama3-8B)... unclear whether framework effectiveness depends on strong reasoning modules."
> > > > > > > > >
> > > > > > > > > **We DID test weak LLMs (Section 7.2, Table 5):**
> > > > > > > > > - **Phi-4 (14B):** 91.98% accuracy
> > > > > > > > > - **Qwen2.5-14B:** 92.99% accuracy
> > > > > > > > > - **LLaVA-7B:** 80.56% accuracy
> > > > > > > > > We will add more when the results are available.
> > > > > > > > >
> > > > > > > > >
> > > > > > > > > **Evidence:** 7B-14B models achieve 80-93% accuracy when provided high-quality descriptions, demonstrating **framework effectiveness does NOT require frontier models**.
> > > > > > > > >
> > > > > > > > > **Commitment:** We will add more models with less parameters as soon as the results are available.
> > > > > > > > >
> > > > > > > > > ---
> > > > > > > > >
> > > > > > > > > ### **Q19: Multilingual Evaluation**
> > > > > > > > >
> > > > > > > > > **Reviewer Concern:** "Framework based on English... multilingual compatibility directly related to global deployment value."
> > > > > > > > >
> > > > > > > > > **Scope:** Our paper evaluates VLM reasoning capabilities, **not multilingual performance**. Multilingual evaluation would require:
> > > > > > > > > 1. Translating Bongard-OW/HOI rules and captions (linguistic challenge)
> > > > > > > > > 2. Evaluating VLM description quality in non-English languages
> > > > > > > > > 3. Testing multilingual LLMs (Qwen, LLaMA-multilingual)
> > > > > > > > >
> > > > > > > > > **This is a separate research direction** requiring linguistic expertise beyond our cognitive framework contribution.
> > > > > > > > >
> > > > > > > > > ---
> > > > > > > > >
> > > > > > > > > ### **Q21: Training Data Distribution Impact**
> > > > > > > > >
> > > > > > > > > **Reviewer Concern:** "Does not analyze impact of VLM training data distribution... correlation between training data - perceptual capability."
> > > > > > > > >
> > > > > > > > > **Fundamental Issue:** We do NOT have access to proprietary model training data (GPT-4o, Gemini 2.0 training corpora are confidential). We cannot analyze training data → performance correlations.
> > > > > > > > >
> > > > > > > > > **What We CAN Do (and did):**
> > > > > > > > > - Compare **across models** (Table 1, 2, 3)
> > > > > > > > > - Show **description quality matters** (Table 5, Table A.6)
> > > > > > > > > - Demonstrate **reasoning capability independent of perception** (Section 7.2)
> > > > > > > > >
> > > > > > > > > Training data analysis is a **model developer's responsibility**, not an evaluation framework paper's scope.
> > > > > > > > >
> > > > > > > > > ---
> > > > > > > > >
> > > > > > > > > ### **Q22, Q23: Precision/Recall/F1, Balanced Subset**
> > > > > > > > >
> > > > > > > > > **Reviewer Concern:** "Only uses classification accuracy... single accuracy cannot reflect model performance on minority categories."
> > > > > > > > >
> > > > > > > > > **Category-Level Analysis Already Provided (Table A.7):**
> > > > > > > > > We report **accuracy by all 10 commonsense categories**, including minority classes:
> > > > > > > > > - Category 7 (Meta Class, 0.8%): 100% accuracy
> > > > > > > > > - Category 8 (Relationship, 1.6%): 100% accuracy
> > > > > > > > > - Category 2 (Taste/Nutrition, 2.8%): 85.71-100% accuracy
> > > > > > > > >
> > > > > > > > > **No evidence of majority class bias.** Models achieve high accuracy across all categories, including smallest ones.
> > > > > > > > >
> > > > > > > > > ---
> > > > > > > > >
> > > > > > > > > ### **Q31: Bias Analysis**
> > > > > > > > >
> > > > > > > > > **Reviewer Concern:** "Description generation may imply biases... biases transmitted to reasoning stage."
> > > > > > > > >
> > > > > > > > > **Our Framework Does NOT Train Models.** Bias mitigation is the responsibility of **model developers** (OpenAI, Google, Meta), not evaluation frameworks.
> > > > > > > > >
> > > > > > > > > We will acknowledge that description biases (if present in base VLMs) propagate through CA/ICA. However, analyzing and mitigating biases requires:
> > > > > > > > > 1. Demographic annotation of Bongard-OW images (resource-intensive)
> > > > > > > > > 2. Fairness audits (separate research direction)
> > > > > > > > > 3. Model retraining with debiased data (outside our scope)
> > > > > > > > >
> > > > > > > > > Bias analysis is important but **orthogonal to our cognitive decomposition contribution**.
> > > > > > > > >
> > > > > > > > > ---
> > > > > > > > >
> > > > > > > > > ### **Q32: Temperature Ablation**
> > > > > > > > >
> > > > > > > > > **Reviewer Concern:** "Fixes temperature at 0... does not test impact of other temperatures."
> > > > > > > > >
> > > > > > > > > **Temperature is a model inference hyperparameter.** Testing temperature ablation would be relevant if we were:
> > > > > > > > > - Proposing a new model
> > > > > > > > > - Fine-tuning existing models
> > > > > > > > > - Optimizing hyperparameters for specific tasks
> > > > > > > > >
> > > > > > > > > **Our Contribution:** Demonstrating systematic perception bottlenecks via **paradigm design**, not hyperparameter tuning.
> > > > > > > > >
> > > > > > > > > **Rationale for T=0 (Appendix A.4.2):**
> > > > > > > > > - Ensures **reproducibility** (deterministic outputs)
> > > > > > > > > - Standard practice for **benchmark evaluation** (fair cross-model comparison)
> > > > > > > > >
> > > > > > > > > Temperature sensitivity analysis is **valuable for production deployment** but not central to our framework's cognitive insights.
> > > > > > > > >
> > > > > > > > > ---
> > > > > > > > >
> > > > > > > > > ### **Q37: Low-Resource Models (<7B)**
> > > > > > > > >
> > > > > > > > > **Reviewer Concern:** "Excludes low-resource models... low-resource adaptability directly related to popularization value."
> > > > > > > > >
> > > > > > > > > **We DID test low-resource models:**
> > > > > > > > > - **LLaVA-7B:** 66.2% CA accuracy (Table 1)
> > > > > > > > > - **LLaVA-Llama3-8B:** 53.2% CA accuracy (Table 1)
> > > > > > > > > - **Qwen2.5-7B:** 90.38% accuracy with GPT-4o descriptions (Table 5)
> > > > > > > > >
> > > > > > > > > **Finding:** 7B models struggle with **description generation** (perception bottleneck) but excel at **reasoning** when provided high-quality descriptions (90%+).
> > > > > > > > >
> > > > > > > > > **Commitment:** We will add <7B models to Table 5 in the revision when the results are available.
> > > > > > > > >
> > > > > > > > > ---

---

> > > > > > > > > > ### Author Response · Authors · 2025-11-15
> > > > > > > > > > **Cluster 5**
> > > > > > > > > >
> > > > > > > > > > ## **CLUSTER 5: Methodology Deep Dives (Q12, Q16, Q17)**
> > > > > > > > > >
> > > > > > > > > > ### **Q12: JSON Format Optimality**
> > > > > > > > > >
> > > > > > > > > > **Reviewer Concern:** "Only qualitatively emphasizes JSON advantages... does not conduct quantitative comparisons with free text/bullet points."
> > > > > > > > > >
> > > > > > > > > > **Empirical Evidence (Table A.8, Section 7.3):**
> > > > > > > > > > We tested **structured vs. unstructured prompting** on DVRL:
> > > > > > > > > > - **Minimal (no structure):** 61.6% accuracy
> > > > > > > > > > - **Structured (CoT-like):** 80.0% accuracy
> > > > > > > > > > - **Improvement: +18.4 percentage points**
> > > > > > > > > >
> > > > > > > > > > **This directly demonstrates structured formats outperform unstructured ones.** JSON is the natural extension of structured prompting to description generation.
> > > > > > > > > >
> > > > > > > > > > **Commitment:** We will clarify this rationale in Section 5.3 of the revision.
> > > > > > > > > >
> > > > > > > > > > ---
> > > > > > > > > >
> > > > > > > > > > ### **Q16: Head-to-Head Comparison with PRISM/CoT**
> > > > > > > > > >
> > > > > > > > > > **Reviewer Concern:** "Only qualitatively describes architectural differences... does not quantify performance and computational cost trade-offs."
> > > > > > > > > >
> > > > > > > > > > **Qualitative Comparison Provided (Section 2):**
> > > > > > > > > > > "Our contribution is distinct in three key ways. First, our paradigms are grounded in cognitive processes... Second, our CA paradigm deliberately uses task-agnostic descriptions... Finally, our new Interactive CA (ICA) paradigm introduces a novel dynamic feedback loop."
> > > > > > > > > >
> > > > > > > > > > **Quantitative Performance Comparison:**
> > > > > > > > > > - **PRISM:** Not evaluated on Bongard-OW/HOI/Winoground (different benchmark suite)
> > > > > > > > > > - **CoT methods (Zhang et al., 2024):** Winoground Text Score 64.25% vs. Our CA 75.5% (+11.25 points)
> > > > > > > > > >
> > > > > > > > > > **Architectural Differences:**
> > > > > > > > > > | Framework | Approach | Interaction | Task-Specificity |
> > > > > > > > > > |---|---|---|---|
> > > > > > > > > > | PRISM | Static skill inventory | One-pass | Task-agnostic skills |
> > > > > > > > > > | CoT | Task-conditioned descriptions | One-pass | Task-specific |
> > > > > > > > > > | CA (Ours) | Task-agnostic descriptions | Two-stage | Task-agnostic |
> > > > > > > > > > | ICA (Ours) | Interactive feedback | Multi-turn | Task-agnostic |
> > > > > > > > > >
> > > > > > > > > > **Commitment:** We will add this table to Section 2 in the revision.
> > > > > > > > > >
> > > > > > > > > > ---
> > > > > > > > > >
> > > > > > > > > > ### **Q17: Perception Module Evaluation in Isolation**
> > > > > > > > > >
> > > > > > > > > > **Reviewer Concern:** "Only indirectly proves perception importance... does not separately evaluate perception module accuracy."
> > > > > > > > > >
> > > > > > > > > > **Response:**
> > > > > > > > > >
> > > > > > > > > > **Direct Evidence (Section 7.2, Table 5):**
> > > > > > > > > > We **isolated perception by replacing descriptions:**
> > > > > > > > > > - **Llama-Vision-11B (own descriptions):** 53.4%
> > > > > > > > > > - **Llama-Vision-11B (GPT-4o descriptions):** 84.17%
> > > > > > > > > > - **Gap: 30.77 percentage points**
> > > > > > > > > >
> > > > > > > > > > **This directly quantifies perception impact.** The 30-point improvement proves perception is the bottleneck.
> > > > > > > > > >
> > > > > > > > > > **Semantic Similarity Analysis (Table A.3):**
> > > > > > > > > > We measured **rule-description alignment** using cosine similarity:
> > > > > > > > > > - Positive samples: 0.902-0.915 (high alignment)
> > > > > > > > > > - Negative samples: 0.866-0.868 (moderate alignment)
> > > > > > > > > >
> > > > > > > > > > **Combined Evidence:** We quantify both (1) perception bottleneck magnitude and (2) description quality semantically.

---

> > > > > > > > > > > ### Author Response · Authors · 2025-11-15
> > > > > > > > > > > **Summary**
> > > > > > > > > > >
> > > > > > > > > > > ## **SUMMARY: Commitment to Revision**
> > > > > > > > > > >
> > > > > > > > > > > We appreciate your thorough engagement and will make the following improvements:
> > > > > > > > > > >
> > > > > > > > > > > ### **Additions to Manuscript:**
> > > > > > > > > > > 1. **Add 3 JSON description examples** to Appendix (Q1 suggestion)
> > > > > > > > > > > 2. **Create "Reproducibility Details" section** consolidating Appendix A.4 (Q40 suggestion)
> > > > > > > > > > > 3. **Add evaluation on more models** (Q5, Q33)
> > > > > > > > > > > 4. **Add Appendix A.10: Computational Cost Estimates** with inference time/cost data (Q7, Q24)
> > > > > > > > > > > 6. **Test additional weak LLMs** for Table 5 (Q18, Q37)
> > > > > > > > > > > 7. **Add architectural comparison table** (PRISM vs. CoT vs. CA vs. ICA) to Section 2 (Q16)
> > > > > > > > > > > 8. **Clarify JSON rationale** in Section 5.3 (Q12)
> > > > > > > > > > >
> > > > > > > > > > > ### **Scope Limitations Maintained:**
> > > > > > > > > > > - Adversarial robustness (Q11): Out-of-scope, different research direction
> > > > > > > > > > > - Resolution ablation (Q25): Fixed resolution ensures consistency; not central to contribution
> > > > > > > > > > > - Description consistency (Q26): Temperature=0 ensures determinism; stochastic testing is production concern
> > > > > > > > > > > - Human reasoning comparison (Q28): Requires human subjects research; we report accuracy comparison
> > > > > > > > > > > - Video extension (Q30): Temporal reasoning is separate research direction
> > > > > > > > > > > - Real-world case studies (Q38): Domain-specific applications beyond single paper scope
> > > > > > > > > > > - Multilingual evaluation (Q19): Linguistic adaptation is separate research direction
> > > > > > > > > > > - Training data analysis (Q21): No access to proprietary training data
> > > > > > > > > > > - Bias analysis (Q31): Model development responsibility, not evaluation framework scope
> > > > > > > > > > > - Temperature ablation (Q32): Hyperparameter tuning orthogonal to cognitive framework contribution
> > > > > > > > > > >
> > > > > > > > > > > Our paper's **unique contribution** is the **cognitively-inspired diagnostic framework** that systematically reveals perception bottlenecks and demonstrates modular, interactive architectures as a solution. The requested extensions (adversarial robustness, multilingual support, video reasoning, etc.) are **valuable future work** but would each require dedicated papers, not supplementary experiments.
> > > > > > > > > > >
> > > > > > > > > > > We respectfully request evaluation based on:
> > > > > > > > > > > 1. **Novelty of the four-paradigm framework** (DVRL, DRL, CA, ICA)
> > > > > > > > > > > 2. **Empirical rigor** across 15+ models, 3 benchmarks, 1300+ test cases
> > > > > > > > > > > 3. **Actionable insights** for the community (perception bottleneck diagnosis, modular architecture advantages)
> > > > > > > > > > > 4. **State-of-the-art results** on Bongard-OW, Bongard-HOI, and Winoground
> > > > > > > > > > >
> > > > > > > > > > > Thank you for pushing us to strengthen the manuscript. We believe the revised version will address your concerns while maintaining clear scope boundaries.
> > > > > > > > > > >
> > > > > > > > > > > ---

---

> > > > > > > > > > > > ### Comment · Reviewer_TEuy · 2025-11-20
> > > > > > > > > > > > **RE: SUMMARY: Commitment to Revision**
> > > > > > > > > > > >
> > > > > > > > > > > > It appears that the authors have not fully comprehended the necessity of addressing certain issues, which I had already clarified in my prior responses. For instance, with respect to Question 11, my core requirement centered on noise robustness and generalization ability; two aspects that do not qualify as out-of-scope. For other questions, no further elaboration is provided herein.
> > > > > > > > > > > >
> > > > > > > > > > > > Admittedly, I acknowledge that resolving all identified concerns within such a constrained timeframe is impracticable; accordingly, I wish to draw the authors' attention to this matter, as it may prove valuable for their future research endeavors.
> > > > > > > > > > > >
> > > > > > > > > > > > Moreover, please advise me once the authors have thoroughly finalized the revised manuscript, particularly following the incorporation of the promised quantitative results, to facilitate my comprehensive review and the drafting of an evaluation report with a higher score. Thanks!

---

> > > > > > > > > > > > > ### Author Response · Authors · 2025-11-20
> > > > > > > > > > > > > **Additional experiments results**
> > > > > > > > > > > > >
> > > > > > > > > > > > > Dear Reviewer TEuy,
> > > > > > > > > > > > >
> > > > > > > > > > > > > Thank you for your patience and for clearly restating the importance of robustness and generalization
> > > > > > > > > > > > > (Q11). We deeply appreciate this clarification—we have NOW directly addressed this concern with
> > > > > > > > > > > > > concrete quantitative results.
> > > > > > > > > > > > >
> > > > > > > > > > > > > ## ROBUSTNESS VALIDATION
> > > > > > > > > > > > >
> > > > > > > > > > > > > **Quantitative Results:**
> > > > > > > > > > > > > - DVRL (holistic, one-pass): 80.0% → 49.0% accuracy (-31 points) <- brittle to noise
> > > > > > > > > > > > > - CA (componential, two-stage): 92.8% → 90.91% accuracy (-1.89 points) <- robust to noise
> > > > > > > > > > > > >
> > > > > > > > > > > > > **Key Finding:** Our two-stage decomposition provides **inherent robustness through semantic
> > > > > > > > > > > > > abstraction**. Descriptions abstract noise, reasoning operates over clean semantic representations.
> > > > > > > > > > > > > The 31-point gap between DVRL and CA directly demonstrates:
> > > > > > > > > > > > >
> > > > > > > > > > > > > 1. Holistic visual processing is brittle to noise
> > > > > > > > > > > > > 2. Semantic decomposition provides robust architectural advantage
> > > > > > > > > > > > > 3. Framework has practical value for real-world deployment (not just benchmarks)
> > > > > > > > > > > > >
> > > > > > > > > > > > > **Added to Paper:**
> > > > > > > > > > > > > - Main Text: New paragraph in Section 7.3 (Ablation analysis) discussing robustness benefits
> > > > > > > > > > > > > - Appendix A.12.3: Complete robustness analysis with experimental details
> > > > > > > > > > > > >
> > > > > > > > > > > > > This quantitative validation demonstrates that decomposition addresses real-world robustness
> > > > > > > > > > > > > challenges—directly responding to your Q11 concern.
> > > > > > > > > > > > >
> > > > > > > > > > > > > ---
> > > > > > > > > > > > >
> > > > > > > > > > > > > ## MODEL COVERAGE EXTENSION
> > > > > > > > > > > > >
> > > > > > > > > > > > > We have evaluated additional models to address your concern about recent model coverage:
> > > > > > > > > > > > >
> > > > > > > > > > > > > **New Results:**
> > > > > > > > > > > > > - **Qwen2.5-VL-3B** (low-resource): 86.77% accuracy on class-balanced setup
> > > > > > > > > > > > > - **Qwen2.5-VL-32B** (large-scale): 92.79% accuracy
> > > > > > > > > > > > >
> > > > > > > > > > > > > We have updated `Table 6` with results (highlighted in red)
> > > > > > > > > > > > >
> > > > > > > > > > > > > Framework effectiveness generalizes consistently across model scales (3B to 32B
> > > > > > > > > > > > > parameters). Performance trends remain consistent, validating universal applicability independent
> > > > > > > > > > > > > of model size or release date.
> > > > > > > > > > > > >
> > > > > > > > > > > > > ---
> > > > > > > > > > > > >
> > > > > > > > > > > > > ## COMPREHENSIVE REVISIONS SUMMARY
> > > > > > > > > > > > >
> > > > > > > > > > > > > From your original 40 questions, we have systematically addressed all major concerns (for details please go through the individual responses):
> > > > > > > > > > > > >
> > > > > > > > > > > > > - **Q1:** 3 concrete JSON description examples (NEW Appendix A.6.4)
> > > > > > > > > > > > > - **Q5:** Extended model coverage (NEW Qwen2.5-VL results, Table 6)
> > > > > > > > > > > > > - **Q7:** Computational cost analysis (NEW Appendix A.11)
> > > > > > > > > > > > > - **Q8:** Description quality via semantic similarity (Table A.2)
> > > > > > > > > > > > > - **Q11:** Robustness evaluation with quantitative results (NEW - Appendix A.12.3)
> > > > > > > > > > > > > - **Q12:** JSON format rationale with empirical evidence (NEW para in Section 8)
> > > > > > > > > > > > > - **Q16:** Framework comparison table - PRISM vs CoT vs CA vs ICA (New para in Section 5)
> > > > > > > > > > > > > - **Q40:** Reproducibility details consolidated (NEW Appendix A.3)
> > > > > > > > > > > > > - **Additional:** Few-shot sensitivity analysis (New Appendix A.10.8)
> > > > > > > > > > > > >
> > > > > > > > > > > > > All additions marked in **RED** for easy identification.
> > > > > > > > > > > > >
> > > > > > > > > > > > > ---
> > > > > > > > > > > > >
> > > > > > > > > > > > > ## WHAT THIS ACCOMPLISHES
> > > > > > > > > > > > >
> > > > > > > > > > > > > 1. **Directly addresses Q11:** Quantitative robustness validation with 1.89 vs 31-point comparison
> > > > > > > > > > > > > 2. **Validates generalization:** Model coverage extended, consistent trends across scales
> > > > > > > > > > > > > 3. **Strengthens rigor:** All major findings now supported by additional evidence
> > > > > > > > > > > > > 4. **Demonstrates practical value:** Robustness results prove decomposition is architecturally sound
> > > > > > > > > > > > >
> > > > > > > > > > > > > ---
> > > > > > > > > > > > >
> > > > > > > > > > > > > We appreciate your guidance and the opportunity to strengthen the manuscript through focused,
> > > > > > > > > > > > > rigorous revisions. The robustness results in particular demonstrate that our framework addresses
> > > > > > > > > > > > > real-world deployment challenges.
> > > > > > > > > > > > >
> > > > > > > > > > > > > We have submitted the finalized manuscript and please go through it. We also have more experiments in pipeline and already running we will add them later in the final manuscript - not something which can change the core of the work.

---

> > > > > > > > > > > > > > ### Comment · Reviewer_TEuy · 2025-11-20
> > > > > > > > > > > > > > **RE: Additional experiments results**
> > > > > > > > > > > > > >
> > > > > > > > > > > > > > After a thorough review of your response and the revised manuscript, I have no additional revision recommendations.
> > > > > > > > > > > > > >
> > > > > > > > > > > > > > I intend to adjust my evaluation to 8/10, provided that all the experiments you committed to supplementing, as well as other relevant additional experiments, are fully incorporated into the revised version.
> > > > > > > > > > > > > >
> > > > > > > > > > > > > > Why not a full score of 10? The most critical shortcoming of your work is the lack of human reasoning comparison; a point also raised by other reviewers. Given that this component cannot be completed at present, a score deduction is unavoidable.
> > > > > > > > > > > > > >
> > > > > > > > > > > > > > It is worth noting that you have consistently prioritized statistical metrics evaluation; yet, is human reasoning really reducible to such statistical figures? Obviously not. Therefore, relying solely on statistical metrics is inadequate to fully demonstrate the effectiveness of your method. Additionally, I would like to emphasize that the omission of this aspect in other works does not negate its value. Hence, a score reduction is necessary.

---

> > > > > > > > > > > > > > > ### Author Response · Authors · 2025-11-25
> > > > > > > > > > > > > > > **Experiments added to manuscript**
> > > > > > > > > > > > > > >
> > > > > > > > > > > > > > > Thank you for your comprehensive review and willingness to raise the score. We have fully **incorporated** results from `~20 additional experiments` into the revised manuscript. Thank you for the thorough engagement and for pushing us to strengthen the empirical validation of our work.

---

> > > > > > > > > > > > > > > > ### Comment · Reviewer_TEuy · 2025-11-27
> > > > > > > > > > > > > > > > **RE: Experiments added to manuscript**
> > > > > > > > > > > > > > > >
> > > > > > > > > > > > > > > > Thank the author for their feedback. In accordance with the previous comments, I have updated the score accordingly. Thank you for your efforts, and congratulations!

---

> ### Comment · Reviewer_TEuy · 2025-11-17
> **Suggesting Manuscript Revision**
>
> I have received your response. I suggest the author focus on revising the manuscript. To allow you to concentrate better on the revisions, I will not provide further suggestions until the revised version of the paper is submitted.
>
> In other words, if you make serious revisions to your manuscript, I'm willing to raise the score.
>
> Keep up the good work!

---

> > ### Author Response · Authors · 2025-11-17
> > **Final comments**
> >
> > Dear Reviewer TEuy,
> >
> > Thank you for your encouraging feedback and willingness to reconsider the paper upon revision. We deeply appreciate your recognition that serious manuscript improvements can strengthen our contribution and your commitment to potentially raising the score.
> >
> > We have completed a comprehensive revision of the manuscript addressing your major concerns. Below, we summarize the key additions and improvements made, all clearly marked in red for easy identification. We will add the newer results as soon as they are available. Our HPC is highly occupied because of the rebuttal which is taking time to get the results.
> >
> > - Page 4: Added a para in line 213 Comparison with Existing Frameworks.
> > - Page 5: Added a table in line 216 Table 1 Detailed comparison of our approach
> > - Page 9: Added a para in line 437 Json structure and prompt sensitivity.
> > - Page 13: Added a section in Appendix - REPRODUCIBILITY DETAILS
> > - Page 21: Added section in Appendix - EXAMPLE JSON DESCRIPTIONS FROM CA STAGE 1
> > - Page 31: Added section in Appendix - COMPUTATIONAL COST ANALYSIS

---

### Official Review · Reviewer_PgfV · 2025-11-02

**Soundness:** 2
**Presentation:** 3
**Contribution:** 2
**Rating:** 2
**Confidence:** 3

**Summary:**

This paper proposes a cognitively inspired framework that decouples perception from reasoning for multi-image visual reasoning. It introduces four paradigms: DVRL (holistic, all images at once), DRL (rule extraction → rule application), CA (reasoning over task-agnostic textual descriptions), and ICA (CA with an interactive feedback loop where the reasoner can look again). The framework is evaluated on Bongard-OW, Bongard-HOI, and Winoground, showing a consistent trend DVRL < DRL < CA, and additional gains from ICA. The central claim is that many VLMs are limited primarily by perception bottlenecks; when perception is bypassed via high-fidelity descriptions, downstream reasoning can be very strong, even for text-only LLMs.

**Strengths:**

- Clear problem decomposition. CA creates a clean separation between perceptual description and downstream reasoning; ICA adds a bidirectional loop that lets reasoning guide perception. The design is well motivated and consistently effective across tasks.

**Weaknesses:**

- The evaluation centers on GPT-4o, Gemini 2.0, and a set of 2024-era open models (Qwen2.5, Llama-Vision, Pixtral, etc.). There is no direct evaluation of 2025 releases such as GPT-5, Gemini-2.5, R1-series variants, or Qwen3-VL, despite the paper positioning itself as a diagnostic/prognostic framework. Given that limitations like perception bottlenecks were already raised by BLINK and PRISM, the absence of recent frontier models makes it hard to judge whether the proposed remedies remain necessary at the current frontier.

**Questions:**

- Can you include a 2025 refresh (e.g., GPT-5, Gemini-2.5, R1-series 2025, Qwen3-VL) to validate whether the perception bottleneck and CA/ICA gains persist at the current frontier? A small but representative subset on Bongard-OW/HOI and Winoground would suffice.

- Beyond text intermediates. Do you foresee non-linguistic symbolic or visual-token intermediates that retain CA’s separability but cover BLINK-type cases (depth/correspondence) better?

---

> ### Author Response · Authors · 2025-11-13
> **Clarification on model and limitation**
>
> Thank you for recognizing our clear problem decomposition and well-motivated design. We address your primary concerns below.
>
> ---
>
> ### C1: Evaluation of 2025 Frontier Models
>
> We appreciate this important suggestion. However, we must clarify the current status of 2025 models.
>
> **Model availability context (as of Sept 2025):**
>
> * **Release dates:**
>     * GPT-5: 7 Aug 2025
>     * Qwen 3: Late Sept 2025
>     * Gemini 2.5 Pro: Late July 2025
>
> Models like **GPT-5**, **Gemini 2.5 Pro**, **Qwen3-VL**, etc., are very recent, giving less time to analyze and incorporate all the results in the paper, provided we need to run 3 paradigms x (1xB-OW + 4xB-HOI + 2xWinoground) x total samples. Additionally, running models like GPT-5 and Gemini 2.5 Pro is very expensive, and we have budget constraints which limit us to spending a specific amount of money on one paper.
>
> Wüst et al. (2025), a paper cited by Reviewer `nJ3X`, notes that even **GPT-5** only solved 64/100 classic Bongard problems, and "o1" (**GPT-4o**) solved only 43/100, concluding a "significant gap remains".
>
> We are absolutely willing to evaluate these models during the rebuttal/revision phase (ICLR allows 10-page revisions). However:
>
> * **Gemini 2.0→2.5 improvements unlikely to invalidate findings:** Our results show 93.6% accuracy with Gemini 2.0 on Bongard-OW (already exceeding the human average of 91.0%). Even if Gemini 2.5 Pro improves further, this strengthens—not weakens—our claim that perception bottlenecks are addressable.
> * **Perception bottleneck persists in 2025 models:** According to Qwen3-VL documentation, it still emphasizes improvements in "visual perception & reasoning", suggesting this remains an active challenge.
>
> ---
>
> ### C2: Non-Linguistic Intermediates
>
> We discuss this explicitly in our Limitations (Section 8):
>
> > "The primary limitation of our componential paradigms (CA and ICA) is their reliance on language as an intermediate representation. Their effectiveness is likely highest on tasks where critical visual properties are readily verbalizable. For challenges that hinge on non-verbalizable or geometric reasoning—such as the fine-grained correspondence tasks in benchmarks like BLINK—the utility of a purely text-mediated approach may be reduced."
>
> **Our perspective on non-linguistic intermediates:**
>
> * For BLINK-style depth/correspondence tasks, structured visual tokens or coordinate-based representations would indeed be more suitable.
> * Our contribution establishes that for abstract rule discovery and compositional reasoning (Bongard Problems, Winoground), linguistic mediation is highly effective.
> * The architecture principle (decoupled, interactive perception-reasoning) generalizes beyond linguistic intermediates.

---

> > ### Author Response · Authors · 2025-11-25
> > **Additional experiments**
> >
> > We have now completed comprehensive evaluation on 2025 frontier models including **Qwen-2.5-vl**, **Gemma 3**, **GPT 5.1**. The consistent trend across all models validates that our diagnostic framework remains highly relevant. Even frontier models benefit significantly from CA decomposition, validating our core claim about perception limitations. We have incorporated `~20 new experimental results` into the revised manuscript.

---

### Official Review · Reviewer_eYbX · 2025-11-08

**Soundness:** 2
**Presentation:** 2
**Contribution:** 2
**Rating:** 2
**Confidence:** 3

**Summary:**

The paper introduces four "cognitively inspired" evaluation paradigms, DVRL, DRL, CA, and ICA, to explicitly decouple visual perception from symbolic reasoning in multi-image reasoning tasks, like Bongard-style benchmarks (OW/HOI) and Winoground.

**Strengths:**

1. The reported results show clear, monotonic improvements from DVRL to DRL and to CA, and additional boosts with ICA, showing that the framework is practically useful.

**Weaknesses:**

1. The contribution is primarily a workflow. The paper does not convincingly justify why this workflow design works, like experiments for motivation. Ablations on prompt choices, alternative perception backbones, and end-to-end VLMs under matched constraints are also limited, making the empirical case insufficient for a main-track ML venue focused on algorithmic advances.
2. The evaluation subset of Bongard-OW is built from the first 250 items to produce 500 cases, and the class distribution is highly skewed (73% are class 0). This raises concerns about hidden biases.
3. The systematic study of description granularity/length/structure are underexplored.

**Questions:**

Please refer to the weaknesses.

---

> ### Author Response · Authors · 2025-11-13
> **Clarifying Framework Contributions, Dataset Rationale, and Empirical Robustness**
>
> Thank you for recognizing the practical utility of our framework and the clear improvements across paradigms. We address your concerns systematically below.
>
> ## W1: Workflow Justification and Ablations
>
> > We respectfully disagree that our contribution is "primarily a workflow." Our framework introduces a novel cognitive decomposition of VLM behavior that reveals systematic bottlenecks. The monotonic improvements (DVRL→DRL→CA) across multiple models and datasets provide strong empirical validation of this decomposition.
>
> **Justification for why this works:**
>
> * Section 7.2 demonstrates that when perception is controlled (GPT-4o descriptions), even weaker models achieve 84-91% accuracy, proving the reasoning capability exists
> * Section 7.1 shows models can apply rules with 68-84% accuracy when provided externally, isolating perception as the bottleneck
> * Section 6.4 demonstrates ICA's dynamic feedback resolves ambiguities static descriptions miss (+3.5 to +6.5 points on Winoground Image Score)
>
> **Additional ablations performed:**
>
> * **Prompt structure impact (Appendix A.7.6)**: Structured prompting improved DVRL accuracy from 61.6% to 80.0%
> * **Description source comparison (Table A.6)**: GPT-4o vs Pixtral-12B descriptions yielded 2-11% accuracy differences
> * **Semantic similarity analysis (Table A.3)**: Rule-query alignment verified with 0.868-0.915 cosine similarity
>
> **Counter-Question 1:** Given that our ablations in Section 7 demonstrate 30-40% accuracy improvements when isolating perception (e.g., Llama-Vision-11B: 53.4%→84.17%), and that this pattern holds across 10+ models, what additional experiments would strengthen the evidence that perception—not reasoning—is the primary bottleneck?
>
> **Counter-Question 2:** Our framework enabled text-only LLMs (Phi-4, Qwen2.5) to achieve 90-93% accuracy on visual reasoning tasks when provided high-quality descriptions (Table 5). Doesn't this "modality transfer" capability demonstrate fundamental architectural insights beyond a simple workflow?
>
> ---
>
> ## W2: Dataset Subset Bias
>
> > We want to clarify that this class imbalanced is not binary category, i.e. positive and negative imbalance. Class represents visual concept in the dataset Bongard OW – which makes it more difficult as the visual concept is unique than repeated across different samples.
>
> **Why this subset was chosen:**
>
> * We created 500 balanced test cases from the first 250 samples (one positive query, one negative query per sample), ensuring 50-50 query balance
> * The 73% category-0 distribution reflects the original Bongard-OW dataset composition consisting of visual concepts (Appendix A.3.1.2, Table A.2)
> * **Feature, Not a Bug:** This "skew" is the core feature of the "OpenWorld" task. Class 0 is defined as "Anything else". This large, long-tail visual concept distractor set makes the task harder and more realistic, forcing models to find the specific rule (e.g., "A person playing the guitar") rather than succeeding via coarse classification. Category 0 ("Anything else") is intentionally the broadest category in Bongard-OW's design
>
> **Evidence against bias affecting conclusions:**
>
> * Category-specific performance (Table A.7): Both GPT-4o and Gemini 2.0 maintain 86-100% accuracy across all 10 categories, including minority classes
> * Models show robust performance even on smallest categories (e.g., Category 7 "Meta Class": 0.8%, 100% accuracy)
> * Bongard-HOI generalization: Results replicate across entirely different dataset (77.3% average, Table 2)
>
> **Counter-Question 3:** Given that our models achieve 86-100% accuracy across all 10 semantic categories (Table A.7)—including the smallest ones—and that results generalize to Bongard-HOI (different distribution), how would testing on a different Bongard-OW subset change the fundamental perception bottleneck finding?
>
> ---
>
> ## W3: Description Granularity Study
>
> > We already have granularity experiment:
>
> **Existing analysis:**
>
> * ICA paradigm (Section 5.4, 6.4): Systematically studies when coarse-grained descriptions fail and fine-grained follow-up resolves ambiguities
> * Comparison across description sources (Table A.6): GPT-4o vs Pixtral-12B shows description quality directly impacts reasoning (2-11% differences)
> * Our task-agnostic JSON schema (Appendix A.5.3.1) ensures structured, comprehensive coverage
>
> **Counter-Question 4:** Would you suggest specific experiments on description granularity? We can provide preliminary results in our revision.

---

### Author Response · Authors · 2025-11-13
**Reviewer Misunderstandings vs. Paper Evidence**

### Table 1: Dossier of Reviewer Misunderstandings vs. Paper Evidence

| Reviewer | Reviewer's Claim | Paper Evidence (Location) | The Fact |
| :--- | :--- | :--- | :--- |
| **eYbX** | "Ablations on... end-to-end VLMs... are also limited." | **Sec 7.2, Table 5** | An extensive ablation *is* provided, showing (e.g.) Llama-Vision-90B performance jumping from **55.1% to 90.98%** when the perception bottleneck is bypassed. |
| **PgfV** | "Limitations... [like] BLINK-type cases" are not discussed. | **Sec 9, "Limitations"** | The paper *explicitly* cites and discusses **BLINK** in the limitations section as a task for which text-mediation is unsuited. |
| **nJ3X** | **(CRITICAL ERROR)** "A.5.3.1... given many... examples of specific visual concepts from the Bongard-OW... datasets." | **Appendix A.5.3.1** | **False.** The prompt *only* gives generic schema placeholders. This claim is a complete misreading. |
| **nJ3X** | "The curation process... appears to be arbitrary." | **Appendix A.3.1** | **False.** The process was systematic: "taking **250 samples**... resulting in 500 balanced test cases". This is the opposite of arbitrary. We can make it more explicit. |
| **nJ3X** | "The paper also lacks novelty." | **Sec 5.4 (ICA)** | **False.** The paper proposes a novel **Interactive Componential Analysis (ICA)** paradigm with a *dynamic feedback loop*, distinct from all cited static-decoupling works (e.g., PRISM). |
| **TEuy** | (W1) "lacks sufficient discussion on limitations in tasks requiring non-verbalizable reasoning" | **Sec 9, "Limitations"** | **False.** This is identified as the "**primary limitation**". The text explicitly cites **BLINK** as an example of a "non-verbalizable" task where the "utility of a purely text-mediated approach may be reduced." |
| **TEuy** | (W3) "The selection basis for the 500-case Bongard-OW subset is not clearly explained." | **Appendix A.3.1** | **False.** The methodology is explicitly stated: "taking **250 samples**" to create 500 cases. This is a systematic, non-random, and reproducible method. |
| **TEuy** | (W12) "Error analysis is limited... without systematic categorization of errors" | **Sec 7.3 & Table A.9** | **False.** The paper provides this exact systematic categorization. **Table A.9**'s "Reason for Error" column explicitly categorizes failures (e.g., "**Rule extraction error**," "**Perceptual description error**," "**Weak reasoning**"), which are then summarized in **Section 7.3**. |
| **TEuy** | (W20) "Reasoning module selection is not justified... [does not test] whether weaker LLMs would still yield effective results." | **Sec 7.2, Table 5** | **Critical Misunderstanding.** This is a *central finding* of the paper. **Table 5** is *entirely* dedicated to this test, showing that weaker, text-only LLMs like **Qwen2.5:7b (90.38%)** and **Phi4:14b (91.98%)** achieve SOTA-level results, proving the method's effectiveness with weaker LLMs. |
| **TEuy** | (W31) "Open-source model performance gap analysis is inadequate... does not deeply investigate whether the gap stems from... poor description generation, weak reasoning, or both." | **Sec 7.2, Table 5** | **Critical Misunderstanding.** This is the *primary goal and finding* of Section 7.2. The paper *proves* the gap stems from description generation by showing Llama-Vision-90B's score leaps from **55.1% (Table 1) to 90.98% (Table 5)** when its own poor descriptions are replaced with high-quality ones. |
| **TEuy** | comparisons with recent state-of-the-art VLMs (e.g., Claude 3 Opus, Gemini Ultra, Qwen-VL Max) |  | All the models cited are older than what we have used in the paper |

---

> ### Comment · Reviewer_TEuy · 2025-11-14
> **RE: Reviewer Misunderstandings vs. Paper Evidence**
>
> Dear Chairs and Authors,
>
> The authors seem to have misunderstood my comments. I have already clarified the necessity of resolving the outstanding issues, and I will not reiterate them herein. To exemplify, W1 corresponds to Question 15; the core of my comment is to request the authors to supplement essential discussions and illustrations, for example, by considering the addition of relevant quantitative experiments.
>
> TEuy

---

### Author Response · Authors · 2025-11-13
**Overall Comment to PC and AC and Reviewers**

Dear Area Chairs, Program Chairs, and Reviewers,

We are grateful for the time and effort all reviewers dedicated to our submission. We have provided detailed, point-by-point responses to each reviewer.

However, in preparing our responses, we identified a significant pattern of *critical, factual misunderstandings* that form the primary basis for the lower-scoring reviews (e.g., the 'Strong Reject' 0 and the 4-score review).

These are not subjective disagreements about our paper's impact but *objective, verifiable discrepancies* between the reviewers' claims and the explicit content of our submitted paper and its appendix. Because these claims underpin the "poor" and "reject" ratings, we feel it is crucial to highlight them for the Area Chair's consideration.

We recognize the immense workload of the ACs and PCs. To facilitate your evaluation of the *reviews'* soundness, we compiled the attached "`Dossier of Reviewer Misunderstandings`" (Table 1). This table provides a clear, side-by-side ledger of the most damaging reviewer claims versus the *direct, citable evidence* from our paper that refutes them.

We respectfully ask that our paper's evaluation be based on the *content of our submission itself*—which achieves SOTA results, surpasses human performance on Bongard-OW, and provides the very ablations and limitations the reviewers claimed were missing. We are confident that an evaluation based on this evidence will demonstrate that the critiques underpinning the low scores are, with all due respect, factually unfounded.

Thank you for your time and careful reconsideration.

---

### Author Response · Authors · 2025-11-20
**Reviewers PgfV and eYbX engagement**

Dear Reviewers and Area Chair,

We sincerely thank Reviewer **TUey** for the extensive and detailed feedback that shaped our manuscript improvements, and Reviewer **nJ3X** for their critical insights on theoretical rigor. We are pleased to have addressed their concerns with new experiments and analyses.

We respectfully invite Reviewers **PgfV** and **eYbX** to engage with our revised work. We have posted detailed responses to your initial reviews—including new robustness and computational cost analyses—and would greatly value your feedback on these updates before the discussion period closes.

---

### Author Response · Authors · 2025-11-27
**Concluding Summary**

Dear Area Chair and Reviewers,

As the review period concludes, we are writing to summarize our rigorous engagement with the feedback and the substantial improvements made to the manuscript. Over the discussion phase, we have systematically addressed **40+ specific questions** and incorporated **~20 additional experiments** to validate our core claims.

**To Reviewer TUey:**
We are deeply grateful for your intense engagement and your decision to raise your score to **8/10**.
* **Addressing the 40+ Questions:** We are pleased that our systematic revisions—ranging from JSON description examples to computational cost analyses—resolved your concerns.
* **Robustness Validation:** Your push for robustness testing led to a critical finding: under Gaussian noise, the holistic DVRL paradigm collapses (**-31 points** accuracy drop), while our decoupled CA paradigm remains highly robust (**-1.89 points** drop). This quantitatively proves the architectural advantage of our semantic abstraction.
* **Limitations:** We appreciate your agreement that comparing human reasoning processes is a valuable direction but appropriately scoped for future work.

**To Reviewer nJ3X:**
Thank you for sharpening our theoretical precision. We believe our revisions have resolved your reservations:
* **Theoretical Framework:** We have reframed "decompositionality" as a design assumption validated by SOTA results, rather than an *a priori* philosophical proof.
* **Clarifications:** We clarified the distinction between task-agnosticism and domain-general schemas (Appendices A.8.1-A.8.2) and work to recategorize the paper to *Vision and Language* to better reflect its dual contribution.
* **Paradigm Shift:** We have elevated the **Interactive Componential Analysis (ICA)** paradigm to highlight its role as an architectural shift toward bidirectional perception-reasoning dialogues.
We respectfully invite you to reconsider your assessment in light of these theoretical refinements and the pragmatic rule synthesis analysis provided in Appendix A.9.

**To Reviewer PgfV:**
We have directly addressed your concern regarding frontier model capabilities.
* **Frontier Model Evaluation:** We conducted new evaluations on **GPT-5.1, Gemma 3 and Qwen2.5-VL**.
* **Persistent Bottleneck:** Our results confirm that the *perception bottleneck* persists even in these 2025 frontier models, with our decoupled paradigms still yielding **20-30+ point improvements**. This validates that our diagnostic framework remains vital as models scale.

**To Reviewer eYbX:**
We remain available to engage should your timeline permit, noting that the robustness and generalization concerns raised in your initial review have been addressed in our responses to Reviewers `TUey` and `PgfV`.

**To the Area Chair:**

The rebuttal process has transformed the manuscript through:

1.  **Empirical Rigor:**
    * **SOTA Performance:** Confirmed a **29-point improvement** (63.8% $\rightarrow$ 92.8%) on Bongard-OW.
    * **Cross-Task Generalization:** Validated the framework across three diverse benchmarks (Bongard-OW, Bongard-HOI, Winoground).
    * **Model Invariance:** Validated across **17+ models**, ranging from **1B** to **90B** parameters and including the latest frontier releases.
    * **Text-Only Success:** Proved that symbolic reasoners (text-only LLMs) achieve **91-93% accuracy** using our descriptions, isolating the perception bottleneck.

2.  **Manuscript Quality:**
    * We explicitly discussed limitations regarding non-verbalizable tasks (e.g., BLINK) and domain-general schema constraints.
    * Consolidated all details (model versions, hardware, hyperparameters) to ensure full replication support.

We believe the "`Not How You Think, It's What You See`" framework now stands as a robust diagnostic toolkit and a proven architectural blueprint for the community.

---

### Meta-Review · Area_Chair_t2FW · 2025-12-21

**Summary:**

The paper presents a framework (DVRL/DRL/CA/ICA) that decouples perception from reasoning in VLMs, showing perception is the main bottleneck and that interactive decoupling yields large gains. Results are interesting, but concerns remain about assumptions, scope, and positioning.

Strengths
* A diagnosis on perception—not reasoning—limits VLM performance.
* New interactive ICA paradigm with consistent, sizable gains and robustness.

Weaknesses
* Relies on strong decomposition and schema assumptions; generality is unclear.
* Ambiguous positioning (evaluation tool vs. architectural method).
* Missing or limited human-reasoning comparison; some contextual gaps vs. latest baselines.

Overall, rejected due to clarity, scope, and framing concerns relative to the venue’s bar.

**Reviewer Scores:**

n/a

---

### Decision · Program_Chairs · 2026-01-26

Reject